# ST2/IL-33 axis blockade inhibits regulatory T cell cytotoxicity towards CD8 T cells in the leukemic niche

Hua Jiang[1,2,9], Denggang Fu[1,2,9], Santhosh Kumar Pasupuleti [2], Baskar Ramdas [2], Alan Long [3], Abdulraouf M. Ramadan[2], Jinfeng Yang[2], Ramesh Kumar [2], Jessica H. Hartman[4], B. Jacob Kendrick [5], Ed Simpson [6], Hongyu Gao[6], Yunlong Liu [6], Drew Moore [7], Suganya Subramanian [7], Stefano Berto [7], Anilkumar Gopalakrishnapillai[8], Sonali P. Barwe [8], Hongfen Guo[3], Nai-Kong V. Cheung [3], Reuben Kapur [2] & Sophie Paczesny [1,2] ✉

Acute myeloid leukemia (AML) patients present with CD8 exhaustion signatures, and pharmacologic inhibition of checkpoints can have therapeutic benefit. The alarmin IL-33 and its receptor STimulation-2 (ST2) promote activation of tissue-regulatory T cells (T_reg cells) and accelerate malignant progression in solid tumors, but their role in leukemia remains unclear. Here, we show that ST2$^+$ T_reg cells are enriched in bone marrow (BM) of humans and mice with AML and promote CD8$^+$ T cells depletion and exhaustion. ST2 deficiency in T_reg cells restores CD8$^+$ T cell function, decreasing AML growth via retention of ST2$^+$ T_reg cells precursors in lymph nodes. AML-activated ST2$^+$ T_reg cells lack T-bet, IFN-γ and Bcl-6, and kill intratumoral CD8$^+$ T cells by amplified granzyme B-mediated cytotoxicity compared to non-AML primed T_reg cells. Engineered anti-ST2 antibodies induce ST2$^+$ T_reg cells apoptosis to extend survival in AML models. Together, our findings suggest that ST2 is a potential checkpoint target for AML immunotherapy.

Therapies for acute myeloid leukemia (AML) have not changed markedly over 30 years, while remarkable advancements have been made in treatments for other blood cancers. Recent advances in genomics have enabled the development of molecular targeted therapies that can extend survival in AML patients, but most patients still succumb[1]. Therefore, the development of more effective, less toxic immune-based therapies for AML is an urgent unmet need. Targeting the inhibitory immune microenvironment to overcome T cell exhaustion with anti-CTLA4 and anti-PD-1/PD-L1 checkpoint inhibitors has proven effective for solid tumors[2,3] but barely for leukemias, although AML patients present with CD8 exhaustion signatures[4–6].

The immunosuppressive action of tumor-infiltrating regulatory T cells (T_reg cells) is a crucial tumor immune-evasion mechanism that is associated with poor prognosis in cancer[7]. T_reg cells encompass diverse transcriptional states distinct from those of peripheral T_reg cells that may modulate tumors[8,9]. Further, T_reg cells have been shown

[1]Department of Microbiology and Immunology and Department of Pediatrics, Medical University of South Carolina, Charleston, SC, USA. [2]Department of Pediatrics, Indiana University School of Medicine, Indianapolis, IN, USA. [3]Department of Pediatrics, Memorial Sloan Kettering Cancer Center, New York, NY, USA. [4]Department of Biochemistry and Molecular Biology, Medical University of South Carolina, Charleston, SC, USA. [5]Flow Cytometry & Cell Sorting Shared Resource, Medical University of South Carolina, Charleston, SC, USA. [6]Center for Computational Biology & Bioinformatics, Indiana University School of Medicine, Indianapolis, IN, USA. [7]Bioinformatics Shared Resource, Medical University of South Carolina, Charleston, SC, USA. [8]Nemours Children's Hospital, Lisa Dean Moseley Foundation Institute of Cancer and Blood Disorders, Wilmington, DE, USA. [9]These authors contributed equally: Hua Jiang, Denggang Fu. ✉e-mail: paczesns@musc.edu

to facilitate disease progression in persistent infections by inducing T cell exhaustion[10,11]. Whether $T_{reg}$ cells directly affect the exhaustion of antigen-specific CD8 T cells in the tumor microenvironment (TME) remains to be verified. Remarkably, the alarmin interleukin (IL)-33 and its receptor, Stimulation-2 (ST2, *Il1rl1* gene), promote functional activation of resident $T_{reg}$ cells[12–14]. Moreover, IL-33/ST2 signaling promotes accumulation and function of $T_{reg}$ cells in inflamed tissues and accelerates malignant progression in solid tumors[12,15–17], but its contribution to AML development remains unknown. The transcription factor Tbet (*Tbx21* gene) drives IFN γ-associated type 1 responses and Tbet deficiency in $T_{reg}$ cells results in defective immune control in models of infection and autoimmunity, particularly through the loss of IFNγ as well as many properties of $T_{reg}$ cells, including their proliferation, maintenance, and suppressive activity[9,18–21]. This leads us to hypothesize that $T_{reg}$ cell-specific deletion of Tbet will restrain ST2+ $T_{reg}$ cell activation and pro-cytotoxicity phenotype.

Here, we investigate the role of ST2+ $T_{reg}$ cells in regulating the activity of tumor-infiltrating effector T cells in several AML models. Our results reveal that the frequencies of ST2+ $T_{reg}$ cells in malignant bone marrow (BM) niches increase over time and correlate with the reduction and exhaustion of CD8 T cells. Specific deletion of ST2 in $T_{reg}$ cells enables CD8-mediated killing of AML cells by sequestering ST2+ $T_{reg}$ cells precursors in the lymph nodes and regulating their transcriptional signatures, and inhibiting $T_{reg}$ cells' cytotoxicity towards CD8 T cells. We next engineered anti-ST2 antibodies with Fc silencing that therapeutically neutralize leukemia-infiltrating ST2+ $T_{reg}$ cells, resulting in prolonged survival in AML models. Thus, ST2 signaling in $T_{reg}$ cells induces a distinctive molecular interplay between host immune cells and leukemia cells that is targetable with engineered anti-ST2 antibodies.

## Results

### Activated ST2+ $T_{reg}$ cells are specifically enriched in malignant leukemic niches

To assess the frequencies of $T_{reg}$ cells, particularly ST2+ $T_{reg}$ cells, we first investigated their presence at the steady state in the BM of healthy humans and mice. We find that the human BM contained high frequency of both total $T_{reg}$ cells and ST2+ $T_{reg}$ cells as compared to peripheral blood mononuclear cells (Supplementary Fig. 1A). Frequencies of ST2 expression in $T_{reg}$ cells is also the highest as compared to other immune cells in the BM niche of healthy donors (HDs) (Supplementary Fig. 1B). Murine BM shows the highest frequency of both total $T_{reg}$ cells and ST2+ $T_{reg}$ cells compared to secondary lymphoid organs (spleen, liver) and peripheral blood where ST2+ $T_{reg}$ cells are lower than BM ST2+ $T_{reg}$ cells (Supplementary Fig. 1C). Frequencies of ST2 expression in $T_{reg}$ cells is high as compared to other immune cells in the BM niche of healthy mice (Supplementary Fig. 1D). These findings confirm that ST2+ $T_{reg}$ cells are present in tissues including in the BM niche in human and in mice as reported formerly[22,23]. To identify if $T_{reg}$ cells in the malignant BM niche differentially express ST2, we analyzed bulk RNA sequencing data from patients with AML in The Cancer Genome Atlas and TARGET databases (TCGA/TARGET, *n* = 360), comparing BM from AML patients at diagnosis to BM from HD data from the GTEX database (*n* = 514). AML patients BM was enriched in ST2+(*IL1RL1*+) FOXP3+ $T_{reg}$ cells as compared to HD (Fig. 1A). These data are validated in 2 other databases: AML_CG1999 (*n* = 585), and GSE6891-AML (*n* = 493) (Fig. 1A). We further analyzed data from AML patients in these databases using the Cell type Identification by Estimating Relative Subsets of RNA Transcripts (CIBERSORT) method[24]. Using the TCGA-AML database, the patients were divided into high and low-ST2 expression groups according to the median ST2 expression level. Fractions of seven types of immune cells were analyzed for high (>median) and low (<median) ST2 expression in AML tumor infiltrate, and $T_{reg}$ cells with high ST2 expression are significantly increased while ST2 expression on the other immune subsets

is not significantly changed (Supplementary Fig. 2). These data were validated in three other databases: TARGET-AML (*n* = 187), GSE6891-AML (*n* = 493), and Beat-AML (*n* = 477) (Supplementary Fig. 2). Notably, using single-cell RNA sequencing with 10X genomics method on AML patients' samples at diagnosis, an increase of total $T_{reg}$ cells was not visible but it was when using immunohistochemistry[25]. To be able to identify rare immune populations in BM samples between AML complete responders (CR) versus refractory patients (Ref.), we selected patients with BM samples after chemotherapy induction with no more than 20% blasts (characteristics in Supplementary Table 1), and performed both single-cell RNA sequencing with 10X genomics method and flow cytometry. Single-cell RNA sequencing showed on the Uniform Manifold Approximation and Projection (UMAP), several subsets of T cells including FOXP3+ $T_{reg}$ cells (Supplementary Fig. 3A) with an overrepresentation of FOXP3+ $T_{reg}$ cells, exhausted CD8 T cells and dysfunctional CD8 T cells in the Ref group versus CR group (Supplementary Fig. 3B). However, a distinct subset of ST2(*IL1RL1*)+ $T_{reg}$ cells was not seen. Inflammatory cytokines such as the alarmin IL-33 and exposure to antigen can induce KLRG1 expression on tissue-resident $T_{reg}$ cells, and recent studies about the role of specialized $T_{reg}$ cells in tissues revealed that the non-lymphoid tissue $T_{reg}$ cells abundantly express ST2 and GATA3, and a large percentage of those ST2+ $T_{reg}$ cells express KLRG1[12,14,26,27]. Therefore, we then compared, by flow cytometry, the frequencies of total Foxp3+ $T_{reg}$ cells, ST2+ $T_{reg}$ cells, and activated KLRG1+ST2+ $T_{reg}$ cells in BM samples of the patients aforementioned (characteristics in Supplementary Table 1, gating strategy in Supplementary Fig. 4A) and found that total $T_{reg}$ cells, ST2+ $T_{reg}$ cells and activated KLRG1+ST2+ $T_{reg}$ cells are increased in refractory patients as compared to patients in CR (Fig. 1B; Supplementary Fig. 5A, B).

To verify that an increase of ST2+ $T_{reg}$ cells and KLRG1+ST2+ $T_{reg}$ cells occur in leukemic samples from murine immuno-competent models of AML as well, a DNMT3A/FLT3^ITD leukemic model[28] was employed and examined in three different stages, pre-leukemia (-1 month), intermediate leukemia (-3 months), and advanced leukemia (-5 months) (Fig. 1C). The frequency and counts of KLRG1+ST2+Foxp3+ $T_{reg}$ cells in the malignant spleen and BM niches increase with leukemia progression (Fig. 1C, gating strategy in Supplementary Fig. 4B, C). To further study these $T_{reg}$ cells, we developed two aggressive leukemia models with adoptive transfer of retrovirally induced MLL-AF9[29] leukemia cells or DNMT3A/FLT3^ITD[28,30,31] leukemia cells from the animal above (schema Fig. 1D). Similar to patients and healthy mice, frequencies of ST2 expression in $T_{reg}$ cells is also the highest as compared to other immune cells in the leukemic BM niches (Fig. 1E). We observed that BM ST2+ $T_{reg}$ cell, KLRG1+ST2+ $T_{reg}$ cell percentages and numbers rose over time after leukemia challenge as compared to the no tumor condition in both models (Fig. 2A, Supplementary Fig. 6). Since MLL-AF9 and DNMT3A/FLT3^ITD-driven leukemia may display the formation of extramedullary tumors, including in the spleen and liver[32], we measured KLRG1+ST2+ $T_{reg}$ cells from other leukemic sites such as the spleen and liver and found similar phenotype as what was observed in the BM niche (Fig. 2A). These findings confirmed that ST2+ $T_{reg}$ cells are present in tissues including in the BM niche both in human and in mice[22,23]. We have also shown that in leukemia, as shown for other diseases[12,14,26,27], a large percentage of tissue-resident ST2+ $T_{reg}$ cells express KLRG1 and that frequencies of KLRG1+ST2+ $T_{reg}$ cells are correlated with total ST2+ $T_{reg}$ cells, although slightly smaller than total ST2+ $T_{reg}$ cells. Further analysis will be performed on total ST2+ $T_{reg}$ cells.

ST2+ $T_{reg}$ cells in the leukemic niche secrete large amounts of key regulatory cytokines such as TGFβ and IL-10 compared to ST2+ $T_{reg}$ cells in mice without tumor (Fig. 2B, MLL-AF9 leukemia and Fig. 2C, DNMT3A/FLT3^ITD leukemia). Importantly, malignant BM-derived ST2+

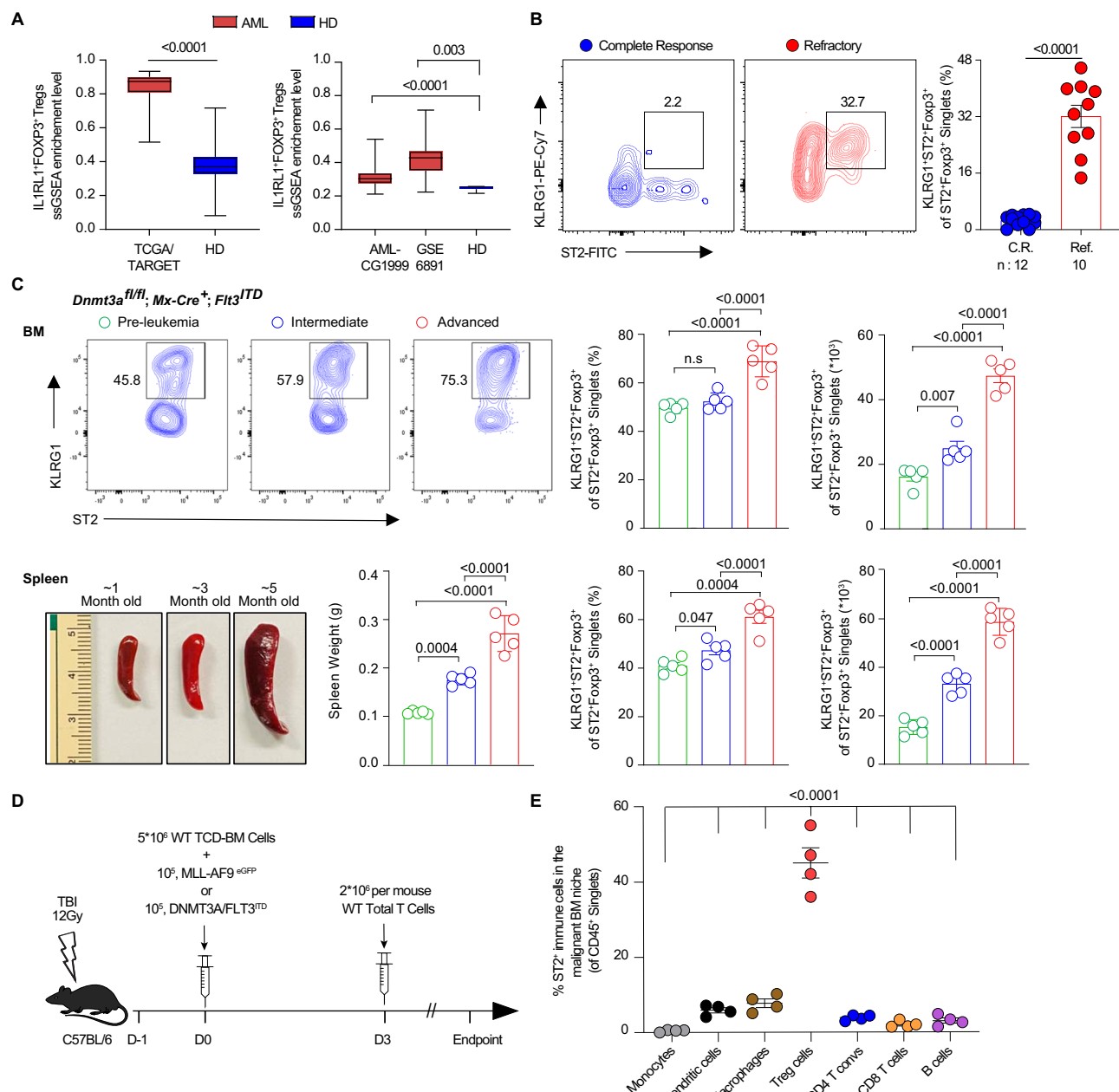

**Fig. 1 | Activated ST2⁺ T_reg cells are specifically enriched in malignant leukemic niches in refractory patients with AML and DNMT3A/Flt3^ITD leukemic mice.**
**A** Tumor-infiltrating *IL1RL1*(ST2)⁺FOXP3⁺T_reg cells enrichment levels in human AML and healthy donors using ssGSEA analysis. Left: TCGA/TARGET-AML (*n* = 360) versus GTEX normal (*n* = 514) samples. Right: Microarray datasets AML_CG1999 (*n* = 585) and GSE6891 (*n* = 493) AML and healthy donors (*n* = 38). In box plots, the center line represents the median, the box limits represent the 25th and 75th percentiles, and the whiskers extend to the minimum and maximum values of IL1RL1⁺FOXP3⁺T_regs ssGSEA enrichment level. **B** Representative plots show the frequencies of T_reg cells positive for ST2 and KLRG1 (KLRG1⁺ST2⁺ T_reg cells) in BM samples from both complete response (CR, *n* = 12) and refractory (Ref., *n* = 10) patients with AML, and statistical quantification. **C** Frequencies of activated KLRG1⁺ST2⁺Foxp3⁺ T_reg cells from three different leukemic stages in primary leukemic mice. Primary leukemic mice were generated by breeding mice to combine Flt3^ITD knock-in and homozygous Dnmt3a floxed alleles (Dnmt3a^fl/fl) with Mx1-Cre (MxCre). There are three different leukemic stages: pre-leukemia (1 month post birth), intermediate leukemia (3 months post birth), and advanced leukemia

(5 months post birth). Representative plots and statistically analyzed activated KLRG1⁺ST2⁺Foxp3⁺ T_reg cells (gated on CD4⁺Foxp3⁺ T cells) in the malignant BM niche from the three different leukemic stages. Spleen size and weight and statistically analyzed to determine activated KLRG1⁺ST2⁺Foxp3⁺ T_reg cells (gated on CD4⁺Foxp3⁺ T cells) in the malignant spleen niche from the three different leukemic stages (*n* = 5 mice per stage). **D** Schema of the adoptive transfer of two leukemic models used hereafter. Total body irradiation (TBI) at 12 Gy on day −1. On day 0, T cell-depleted BM cells (TCD-BM, 5 × 10⁶ per mouse) mixed with retrovirally induced MLL-AF9^eGFP or DNMT3A/FLT3^IDT-mutant leukemic cells (10⁵ per mouse) were injected intravenously. Then, 2 million syngeneic T cells were further transplanted intravenously on day 3. **E** Frequencies of ST2 expression on immune cells in the malignant BM niche. Frequencies of ST2 expression on immune cells in the malignant BM niche (MLL-AF9 model) at day 14 post-AML transfer. Graphed data represent three independent experiments (*n* = 4). The data are presented as means ± s.e.m. (error bar), and compared using two-sided unpaired *t*-test or ANOVA with post-hoc Bonferroni *t*-test for 3 groups or more. *P* values are presented in the figure. Source data are provided as a file.

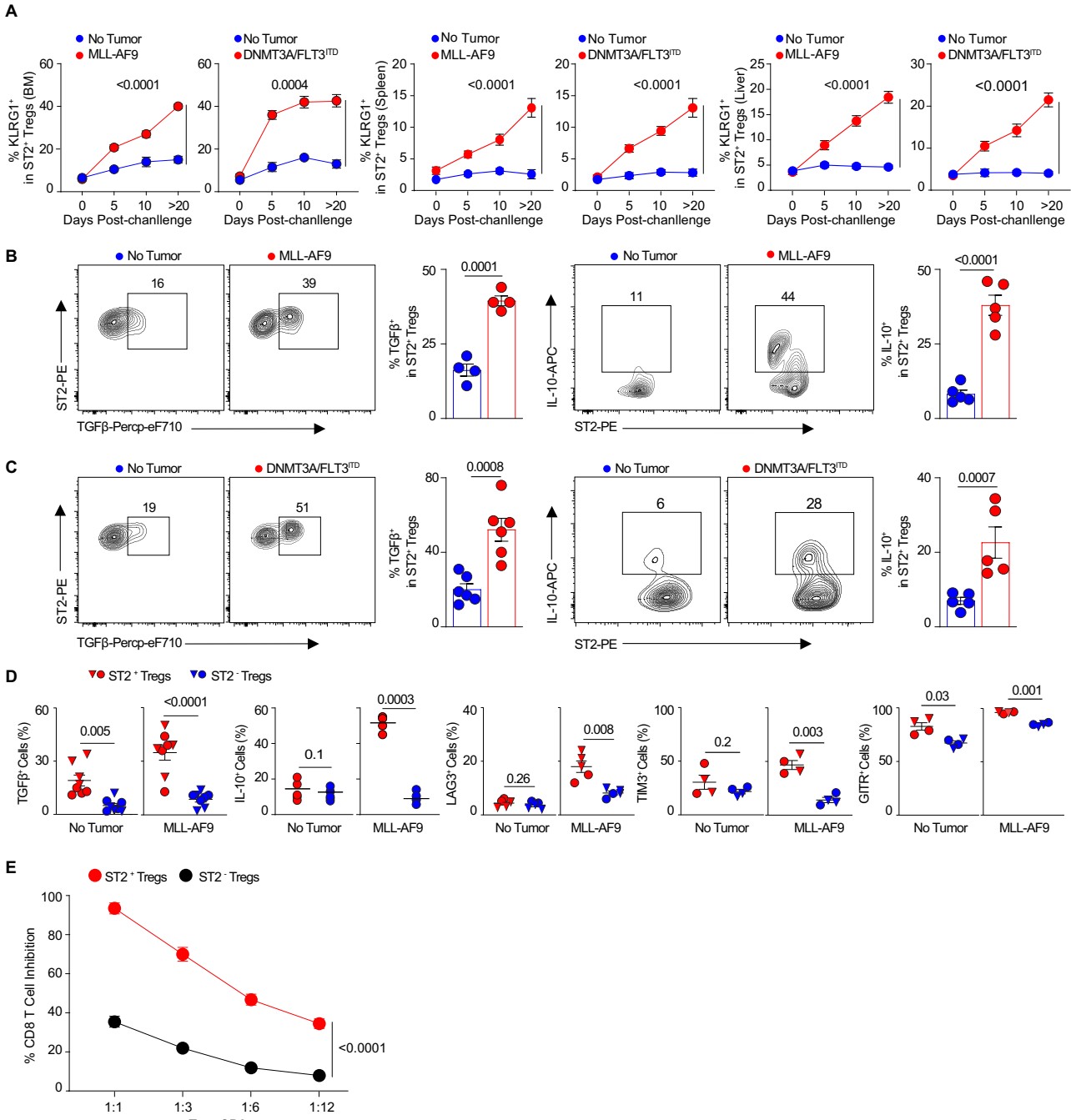

**Fig. 2 | In two AML models, the frequency, cytokine output, and immunosuppressive activity of activated ST2+ T_reg cells in malignant leukemic niches—bone marrow, spleen, and liver—intensify progressively over time. A** Kinetics of KLRG1+ST2+ T_reg cells in the malignant BM, spleen, and liver niches post leukemic cell challenge (10^5 per mouse). Frequencies of KLRG1+ST2+ T_reg cells in the malignant BM niches on the indicated days post MLL-AF9 leukemic cell challenge (*n* = 4 on days 0, 5 and 10, and *n* = 10 on day >20), or post DNMT3A/FLT3ITD-mutant leukemic cell challenge (*n* = 4 on days 0 and 5, *n* = 5 on day 10, and *n* = 10 on day >20). **B** Representative flow plots and frequencies of TGFβ+ST2+ T_reg cells (*n* = 4) and IL-10+ST2+ T_reg cells (*n* = 5) in the malignant BM niches on day 10 post MLL-AF9 leukemic cell challenge or (10^5 per mouse). **C** Representative flow plots and frequencies of TGFβ+ST2+ T_reg cells (*n* = 6) and IL-10+ST2+ T_reg cells (*n* = 5) in the malignant BM niches on day 10 post DNMT3A/FLT3ITD leukemic cell challenge or

(10^5 per mouse). **D** Percentages of T_reg cells positive for cytokines (TGFβ, IL-10) and functional markers (LAG3, TIM3, and GITR) among paired ST2+ T_reg cells versus ST2- T_reg cells on day 10 post MLL-AF9 cell challenge (10^5 per mouse, *n* = 5). Data are from two different replicates (shown as different shapes). The control group had no tumor cell transfer. A paired *t*-test was used to compare ST2+ T_reg cells versus ST2- T_reg cells. **E** Immunosuppression assay of ST2+ T_reg cells versus ST2- T_reg cells. Percent suppression of ex vivo T cell proliferation at various ratios of T_reg: CD8 T (1:1, 1:3, 1:6, 1:12) for cells taken on day 10 post MLL-AF9 leukemic cell challenge and cultured for 5 days (*n* = 2). The data are presented as means ± s.e.m. (error bar), and compared using two-sided unpaired *t*-test or ANOVA with post-hoc Bonferroni *t*-test for three groups or more. *P* values are presented in the figure. Source data are provided as a file.

$T_{reg}$ cells produce significantly more TGFβ and IL-10 than ST2$^-$ $T_{reg}$ cells (Fig. 2D, MLL-AF9 leukemia; Supplementary Fig. 7A, B, DNMT3A/FLT3$^{ITD}$ leukemia). Drivers of functional changes in $T_{reg}$ cells (LAG3, TIM3, and GITR) are also expressed at higher levels in ST2$^+$ $T_{reg}$ cells versus ST2$^-$ $T_{reg}$ cells from the malignant BM niches (Fig. 2D, MLL-AF9 leukemia; Supplementary Fig. 7C, E, DNMT3A/FLT3$^{ITD}$ leukemia). We further assessed the immune suppressive capacity of TME BM ST2$^+$ $T_{reg}$ cells as compared to ST2$^-$ $T_{reg}$ cells with CD8 T cells immune suppression assays and showed that ST2$^+$ $T_{reg}$ cells are significantly more suppressive than ST2$^-$ $T_{reg}$ cells (Fig. 2E). Collectively, ST2$^+$ $T_{reg}$ cells in the BM, spleen and liver leukemic niches display an activated phenotype (KLRG1$^+$, LAG3$^+$, TIM3$^+$, GITR$^+$), secrete high levels of TGFβ and IL-10, and significantly enhance CD8 T cells immune suppression.

## Intratumoral leukemic ST2$^+$ $T_{reg}$ cells accelerate CD8 T cells depletion, exhaustion, and Tbet/IFNγ loss

CD8 T cells serve as a central player in antitumor immunity in solid tumors, but their role is not yet established in liquid tumors. Therefore, we correlated BM leukemic ST2$^+$ $T_{reg}$ cell frequencies with TME-derived BM leukemic CD8 T cells parameters. Our data revealed that ST2$^+$ $T_{reg}$ cell frequencies and numbers are inversely correlated with those of CD8 T cells and proliferating CD8 T cells in both models of leukemia (Fig. 3A, B). To explore the role of CD8 T cells in inducing immunity in AML, we used our MLL-AF9 leukemia model with or without CD8 depletion (Fig. 3C), and found that survival was markedly shortened ($P < 0.0001$) and MLL-AF9 leukemic cells increased ($P = 0.006$, gating strategy in Supplementary Fig. 4D) in the group receiving anti-CD8 versus anti-IgG1 antibodies (Fig. 3C), thus revealing that CD8 T cells play an indispensable role in anti-AML immunity. Further, leukemic BM ST2$^+$ $T_{reg}$ cell numbers were positively correlated with expression levels of CD8 T cell exhaustion markers (PD-1 (gating strategy in Supplementary Fig. 4B), TIM3, PD-1 and TIM3, CTLA4, LAG3, and TIGIT) (Fig. 3D) and negatively correlated with Tbet, IFNγ (gating strategy in Supplementary Fig. 4B), CD107a (T cell degranulation marker), and GZMB/perforin cytolytic molecules in both models (Fig. 3E, MLL-AF9 leukemia and Supplementary Fig. 8A, B, DNMT3A/FLT3$^{ITD}$ leukemia). CD4$^+$ effector T cells are also decreased, while a more moderate correlation was seen with NK cells (Supplementary Fig. 9A, B). Functional assays showed that TME BM CD8 T cells produce less IL-2, TNF (Fig. 3F), and IFNγ (Fig. 3G) in response to the stimulation of PMA/ionomycin (or even at baseline for IL-2) and have decreased cytotoxicity towards MLL-AF9 leukemic cells compared to the non-TME CD8 T cells (Fig. 3H). These data reveal that with the progression of AML, leukemic ST2$^+$ $T_{reg}$ cells are increasingly enriched in the T cell pool of the leukemic niche (BM, spleen, or liver), which correlates with depletion of total leukemic CD8 as well as increased exhaustion resulting in a diminished type 1 phenotype.

## ST2 deficiency specifically in $T_{reg}$ cells increases CD8 T cell counts, Tbet/IFNγ expression, and Wilms Tumor 1 (WT1) tumoral specificity while decreasing CD8 T cell exhaustion to decrease AML growth

We next modeled in our MLL-AF9$^{eGFP}$ leukemic mice an adoptive transfer with 20% $T_{reg}$ cells from Foxp3$^{eGFP}$ reporter mice crossed with ST2$^{-/-}$ mice (20 generations) and 80% conventional T cells in lethally irradiated mice (Fig. 4A). We found that mice received ST2$^{-/-}$ Foxp3$^{eGFP+}$ $T_{reg}$ cells survived longer than mice given wild-type (WT) Foxp3$^{eGFP+}$ $T_{reg}$ cells (Fig. 4B). Tumor-infiltrating total CD3$^+$ gated Foxp3$^{eGFP+}$ $T_{reg}$ cells are not altered, whereas activated KLRG1$^+$ $T_{reg}$ cells are significantly decreased in the ST2$^{-/-}$Foxp3$^{eGFP+}$ $T_{reg}$ cell group versus the WT Foxp3$^{eGFP+}$ $T_{reg}$ cell group (Supplementary Fig. 10A, B), which paralleled decreases in CD3$^-$ gated MLL-AF9$^{eGFP+}$ leukemic cells at serial timepoints of Day 5, 10 and >20 in leukemic models with doses of $10^5$ and $10^6$ MLL-AF9$^{eGFP}$ cells (Fig. 4C; Supplementary Fig. 11). We subsequently investigated whether CD8-related type 1 immunity was altered. As hypothesized, the frequencies and numbers of total CD8 T cells, frequencies of proliferating Ki67$^+$CD8 T cells are higher in the ST2$^{-/-}$ Foxp3$^{eGFP+}$ $T_{reg}$ cell transplantation group (Fig. 4D; Supplementary Fig. 12A, B), while CD8$^+$PD-1$^+$ T cells are decreased (Fig. 4E). Furthermore, the frequencies of both CD8$^+$Tbet$^+$ and CD8$^+$IFNγ$^+$ T cells are consistently higher in the group with ST2$^{-/-}$Foxp3$^{eGFP+}$ $T_{reg}$ cell transplantation than in the group with WT Foxp3$^{eGFP+}$ $T_{reg}$ cell transplantation (Fig. 4F).

Since the model above is lethally irradiated, there are no persistent immune cells, and only the transferred cells are studied. However, other immune cells in the leukemic BM niche may potentially express ST2, such as macrophages, monocytes, granulocytes, and B cells[33–35]. Therefore, to further establish the weight of leukemic ST2$^+$ $T_{reg}$ cells in the overall role of ST2 in the BM niche, we used a non-irradiated leukemic model and inoculated MLL-AF9 leukemia cells into WT or ST2$^{-/-}$ mice and additionally depleted $T_{reg}$ cells with anti-CD25 antibody (Clone No.: PC61) on day 1 and day 7 at the total dose of 500 µg/mouse or used isotype-Ab control (Rat IgG1) (Fig. 4G). Survival of Treg-depleted ST2$^{-/-}$ mice is significantly extended as compared to non-$T_{reg}$-depleted (isotype-treated) WT mice. Approximately half of the survival increase is due to $T_{reg}$ cells (anti-CD25 in WT mice) and half to other non-$T_{reg}$ ST2-expressing cells (isotype in ST2$^{-/-}$ mice). Consistent with the survival data, the frequencies of eGFP-positive MLL-AF9 are significantly decreased in the BM in both isotype-Ab and CD25-Ab-treated ST2$^{-/-}$ mice. To verify that the additional ST2 effect is independent of $T_{reg}$ depletion, we detected Foxp3$^+$ $T_{reg}$ cells and showed that in both CD25-Ab-treated groups, $T_{reg}$ cells were successfully removed when compared to the control groups (Fig. 4G). This suggests that there is an additional role of ST2 beyond the tumor-infiltrating ST2$^+$ $T_{reg}$ cells that account for approximately half of the effect.

To confirm the specific and unique role of ST2 in leukemic $T_{reg}$ cells, we crossed mice with loxP-flanked ST2 alleles ($ST2^{fl/fl}$) that we generated (see Methods) with Foxp3$^{YFP-Cre}$ ($Foxp3^{Cre}$) mice[36] to conditionally deplete ST2 in Foxp3$^+$ $T_{reg}$ cells (designated $Foxp3^{Cre}ST2^{fl/fl}$) and used these mice with the non-irradiated MLL-AF9 leukemic model (Fig. 5A). For each leukemic experiment we verified by flow cytometry that ST2 was specifically deleted in $T_{reg}$ cells and no other immune cells in $Foxp3^{Cre}ST2^{fl/fl}$ mice (Fig. 5A). In this model where ST2 is deleted in all $T_{reg}$ cells, we observed that compared with $Foxp3^{Cre}$ leukemic controls, $Foxp3^{Cre}ST2^{fl/fl}$ leukemic mice had longer survival and a lower leukemia burden (Fig. 5B). The frequencies and numbers of CD8 T cells, proliferating CD8 T cells (CD8$^+$Ki67$^+$), and type 1 CD8 T cells (CD8$^+$Tbet$^+$, CD8$^+$IFNγ$^+$ T) are significantly higher in the malignant BM of $Foxp3^{Cre}ST2^{fl/fl}$ mice at day 14, while exhausted CD8 T cells (CD8$^+$PD-1$^+$) decrease (Fig. 5C). These data confirm the results observed with the adoptive transfer model. Per contra, the conditional model allowed us to examine changes in all immune cells, including CD4$^+$ T effector cells, NK cells (NK1.1$^+$), myeloid cells, and B cells. The frequencies of CD4$^+$ effector T cells are also increased in $Foxp3^{Cre}ST2^{fl/fl}$ mice versus $Foxp3^{Cre}$ control mice (Fig. 5D). NK cell frequencies are increased in the BM of Foxp3$^{Cre}$ST2$^{fl/fl}$ mice compared to controls (Fig. 5E). The frequencies of either myeloid cells or macrophages, granulocytes, monocytes, and B cells do not differ between groups in the conditional model (Fig. 5F).

Because MLL-AF9 leukemic cells express the WT1 antigen (Supplementary Fig. 13), we further investigated the antigen specificity of CD8 T cells towards MLL-AF9 leukemic cells using tetramers recognizing the WT1 antigen[37,38]. With the development of AML, the frequency of WT1-tetramer$^+$ CD8 T cells progressively increases in the BM of both Foxp3$^{Cre}$ and Foxp3$^{Cre}$ST2$^{fl/fl}$ mice, but much higher numbers are found in the BM of Foxp3$^{Cre}$ST2$^{fl/fl}$ mice (Fig. 5G). Additionally, in the BM of Foxp3$^{Cre}$ST2$^{fl/fl}$ mice, the frequency of PD-1$^+$ exhausted CD8 T cells is significantly decreased within the WT1-tetramer$^+$CD8$^+$ subset (Fig. 5H). Since antigen-specific central memory T cells provide a better antitumoral immunity compared to effector memory T cells due to

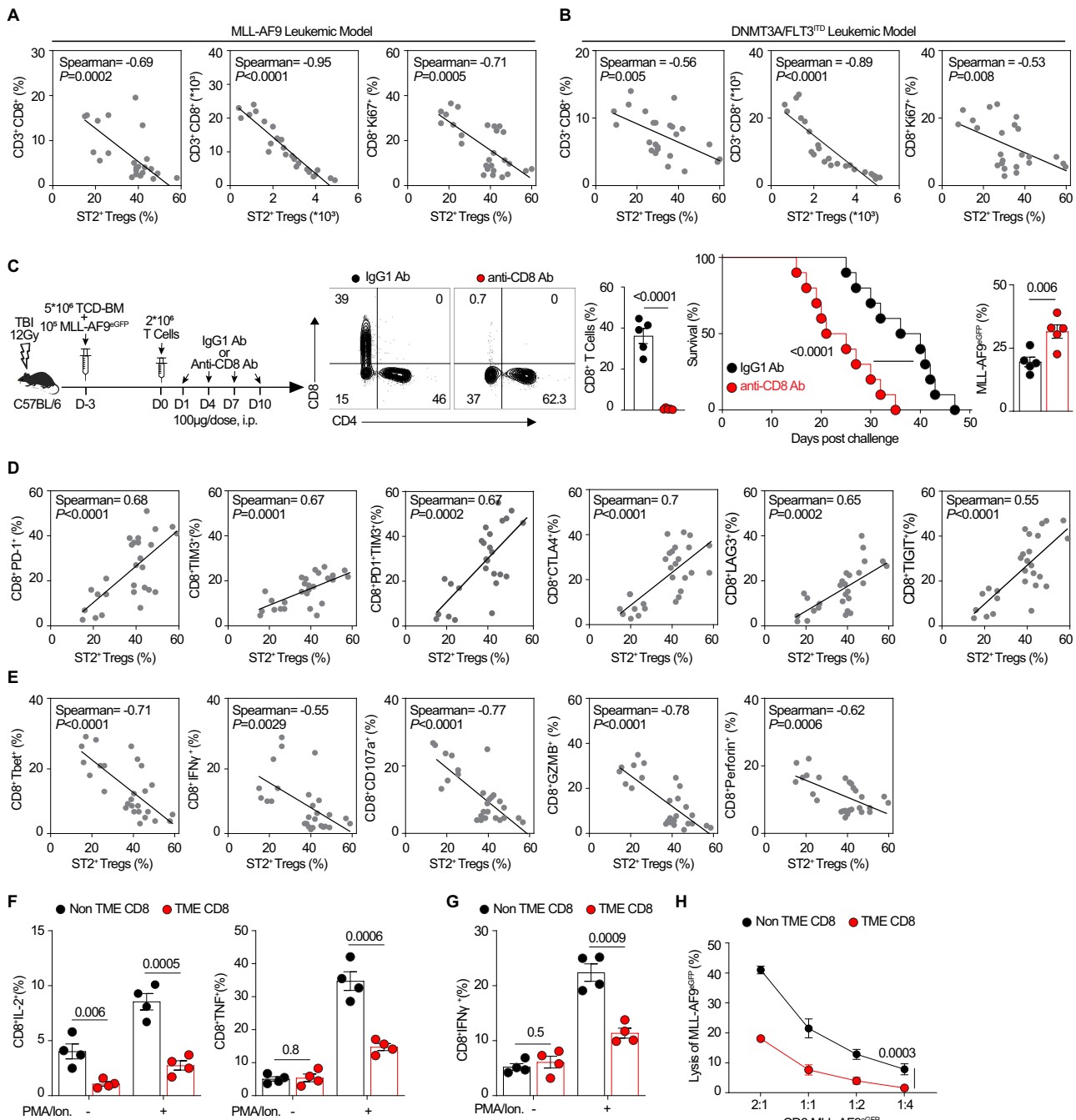

**Fig. 3 | Activated ST2⁺ T_reg cells in the leukemic niches correlate with CD8 T cell depletion and exhaustion. A**, **B** Correlation analysis of CD3⁺CD8⁺ T cells ($n = 24$, samples taken on day 5, day 10 and out of day 20 post leukemic challenge) and CD8⁺Ki67⁺ T cells ($n = 27$, samples taken on day 5, day 10 and out of day 20 post leukemic challenge) with ST2⁺ T_reg cells from the malignant BM niches post MLL-AF9 or DNMT3A/FLT3^ITD-mutant leukemic cell challenge ($10^5$ per mouse), using Spearman's correlation analysis. From three independent experiments. **C** CD8 T cells depletion in the adoptively transferred MLL-AF9 leukemic model. CD8 T cell depletion in the adoptive transfer MLL-AF9 leukemic model ($10^5$ per mouse) was used (see schematic and representative flow plots and frequencies of CD8 T cells in the malignant BM niches upon anti-CD8 neutralizing Ab treatment). Data represent three independent experiments ($n = 5$); Kaplan–Meier survival curve post MLL-AF9 leukemic cell challenge ($10^5$ per mouse) upon CD8 T cell depletion ($n = 10$). Frequencies of MLL-AF9 leukemic cells in the malignant BM niches on day 14 post-challenge ($n = 5$). Log-rank test was used for survival analysis. **D** Correlation analysis of CD8⁺PD-1⁺ T cells, CD8⁺TIM3⁺ T cells, and CD8⁺PD-1⁺TIM3⁺ T cells, CD8⁺CTLA4⁺ T cells, CD8⁺LAG3⁺ T cells, and CD8⁺TIGIT⁺ T cells with ST2⁺ T_reg cells from the malignant BM niches post MLL-AF9 leukemic cell challenge ($10^5$ per mouse), using

Spearman's correlation analysis ($n = 27$, samples taken on day 5, day 10 and out of day 20 post leukemic challenge). From three independent experiments. **E** Correlation analysis of CD8⁺Tbet⁺ T cells, CD8⁺IFNγ⁺ T cells, CD8⁺CD107a⁺ T cells, CD8⁺GZMB⁺ T cells, and CD8⁺Perforin⁺ T cells with ST2⁺ T_reg cells from the malignant BM niches post MLL-AF9 leukemic cell challenge ($10^5$ per mouse), using Spearman's correlation analysis ($n = 27$). From three independant experiments. **F** IL-2 and TNF production by non-TME and TME CD8 T cells taken on day 10 post MLL-AF9 leukemic cell challenge with and without PMA/Ionomycin stimulation ($n = 4$). **G** IFNγ production by non-TME and TME CD8 T cells taken on day 10 post MLL-AF9 leukemic cell challenge with and without PMA/Ionomycin stimulation ($n = 4$). **H** Functional killing towards MLL-AF9 leukemia of non-TME and TME CD8 T cells taken on day 10 post MLL-AF9 leukemic cell challenge at different CD8/MLL-AF9 ratios. Cytolytic effect evaluated with SYTOX staining ($n = 4$). The data are presented as Spearman correlations and *p* value for the rank correlation (**A**, **B**, **D**, **E**), means ± s.e.m. (error bar) compared using two-sided unpaired *t*-test (**C**, **H**), and Kaplan–Meier curves compared using log-rank test (**C**). *P* values are presented in the figure. Source data are provided as a file.

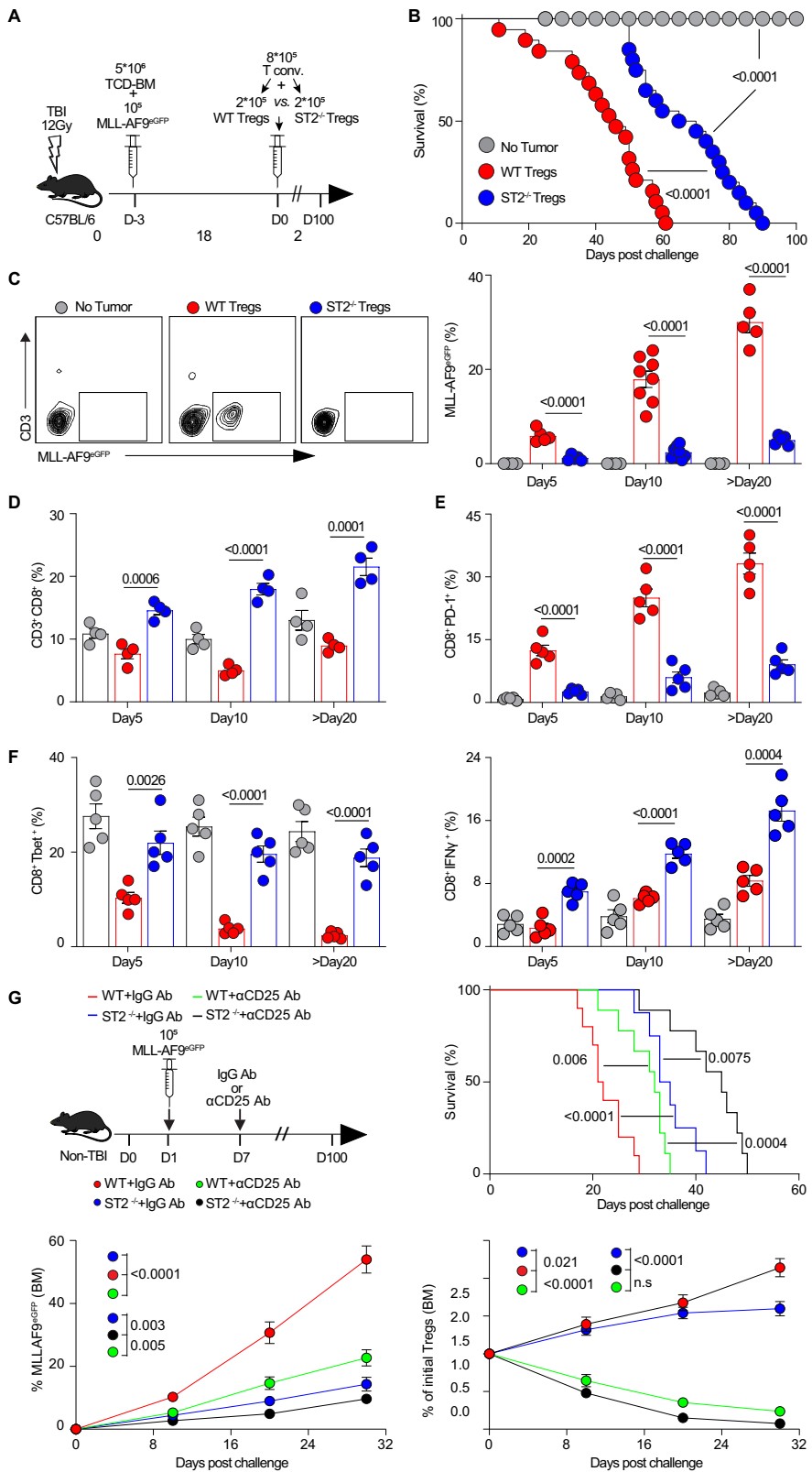

their ability to rapidly recirculate through secondary lymphoid organs, allowing for sustained antigen presentation and a more robust immune response against cancer cells when re-exposed to tumor antigens[39], we explored the frequency of central memory CD8 T cells within WT1-tetramer[+] CD8 T cells. In the BM of Foxp3[Cre]ST2[fl/fl] mice, the frequency of central memory CD8 T cells (CD45RA[–]CCR7[+]) is higher in WT1-tetramer[+] CD8 T cells compared to Foxp3[Cre] control mice but not

in WT1-tetramer[–] CD8 T cells (Fig. 5I). Accordingly, the percentages of central memory CD62L[high]CD44[high] T cells are higher within the WT1-tetramer[+] subset but not in the WT1-tetramer[–]CD8[+] subset in the malignant BM of Foxp3[Cre]ST2[fl/fl] mice versus Foxp3[Cre] mice (Fig. 5J). Taken together, conditional deficiency of ST2 under the Foxp3 promoter altered the activity of T$_{reg}$ cells from the malignant leukemic niche, increasing the frequencies of central memory WT1-specific CD8

**Fig. 4 | Adoptive transfer of ST2-deficient T$_{reg}$ cells increases CD8 T cells count, Tbet/IFNγ expression, and WT1 tumoral specificity while decreasing CD8 T cells exhaustion to decrease AML growth. A** Schematic of the MLL-AF9 leukemic model with the adoptive transfer of 20% WT or ST2$^{-/-}$ T$_{reg}$ cells among total T cells. **B** Kaplan–Meier survival analysis of the adoptive transfer leukemic model as described in (**A**) (10$^5$ MLL-AF9 $^{eGFP}$ leukemic cells per mouse, $n = 20$). Log-rank test was used to compare groups. **C** Representative flow plots showing the frequencies of MLL-AF9$^{eGFP}$ in the malignant BM niches on day 10 post leukemic cell challenge (10$^5$ per mouse), and the statistical analysis results for the indicated days ($n = 5$ on day 5, $n = 8$ on day 10, and $n = 5$ on day >20). Statistically analyzed frequencies of CD3$^+$CD8$^+$ ($n = 4$) (**D**), CD8$^+$PD-1$^+$ ($n = 5$) (**E**), CD8$^+$Tbet$^+$ and CD8$^+$IFNγ$^+$ ($n = 5$) (**F**) T cells in the malignant BM niches on the indicated days post MLL-AF9 leukemic cell challenge (10$^5$ per mouse). **G** Schema of non-irradiated MLL-AF9 leukemia model in

either WT or ST2$^{-/-}$ mice followed by in vivo depletion of T$_{regs}$ with an anti-CD25-Ab (Clone PC61) on day 1, and day 7 for a total dose of 500 μg/mouse; the control group received an isotype-Ab (Rat IgG1). Kaplan–Meier curves for survival in the four groups: both survival of T$_{reg}$-depleted WT mice (green curve) and ST2$^{-/-}$ mice (blue curve) are significantly extended in comparison to the WT mice receiving control isotype-Ab (red curve), but both differ between them. T$_{reg}$-depleted ST2$^{-/-}$ mice (black curve) displayed significantly improved survival as compared to T$_{reg}$-depleted WT mice (green curve). Frequencies of eGFP$^+$ MLL-AF9 cells in the BM and of T$_{reg}$ cells in the BM in the 4 groups over time ($n = 4$ per time point). The data are presented as means ± s.e.m. (error bar) compared using two-sided unpaired $t$-test or ANOVA with post-hoc Bonferroni $t$-test for three groups or more (**C–G**), and Kaplan–Meier curves compared using log-rank test (**B**, **G**). $P$ values are presented in the figure. Source data are provided as a file.

T cells while decreasing their exhaustion. Conditional deficiency of ST2 in T$_{reg}$ cells resulted in a significant survival benefit in leukemic mice.

## Deficiency of ST2 in T$_{reg}$ cells maintains their numbers in LNs at a precursor stage while reducing their numbers in BM, and BM leukemic ST2$^+$ T$_{reg}$ cells have a high migratory potential

Functional ST2$^+$ T$_{reg}$ cells have been shown to reside in non-lymphoid tissues, and a subpopulation of ST2$^+$ T$_{reg}$ cells at a precursor stage has been identified in lymphoid organs. These precursors express the transcription factors nuclear factor interleukin 3 regulated (Nfil3), basic leucine zipper ATF (Batf), and GATA-binding protein 3 (Gata3)[40]. To explore whether ST2 modulates these transcription factors in the context of the leukemia niche, we performed RNA sequencing of non-tumoral and tumoral infiltrating WT(ST2$^+$) versus ST2$^{-/-}$ T$_{reg}$ cells. We found that Nfil3, Batf and Gata3 expression is higher in ST2$^+$ T$_{reg}$ cells than in ST2$^{-/-}$ T$_{reg}$ cells, and ST2$^+$ T$_{reg}$ cells have higher expression of activation markers such as Ccr8, Pparg, Areg, Il10, Klrg1 and Ebi3 (Fig. 6A). To investigate if this is due to a differential localization of the mature and precursor stages in the LNs versus BM, we measured the protein levels of three representative markers for tissue T$_{reg}$ cells that are already expressed on their precursors: Nfil3, Batf and Gata3 in both malignant LN-infiltrating and BM-infiltrating T$_{reg}$ cells from Foxp3$^{Cre}$ control mice versus Foxp3$^{Cre}$ST2$^{fl/fl}$ mice. The results show that LN-derived T$_{reg}$ cells from leukemic Foxp3$^{Cre}$ST2$^{fl/fl}$ mice have higher protein expression of Nfil3, Batf, and Gata3 than those LN-T$_{reg}$ cells from leukemic Foxp3$^{Cre}$ mice, and the expression of these proteins in LN-T$_{reg}$ cells is inversely correlated with their expression in BM T$_{reg}$ cells for each respective group (Fig. 6B). This data suggests that ST2 deficiency in T$_{reg}$ cells limits the migration of precursor ST2$^+$ T$_{reg}$ cells from the LNs to the BM, consequently limiting the presence of functional mature ST2$^+$ T$_{reg}$ cells in the malignant BM. Since our model suggested ST2 expression determines the trafficking of ST2$^+$ T$_{reg}$ cells from LNs into the BM, we next assessed the expression of corresponding migration markers on ST2$^+$ T$_{reg}$ cells compared to ST2$^-$ T$_{reg}$ cells. There is an increased expression of CCR2, CCR5, CCR6, CCR7, CCR8, CCR9, CXCR3, and CXCR6 on ST2$^+$ T$_{reg}$ cells versus ST2$^-$ T$_{reg}$ cells in both the healthy and leukemic BM niche (Fig. 6C). This chemokine receptor analysis infers that the ST2$^+$ T$_{reg}$ cells have a higher migratory potential. Importantly, CXCR4 expression is significantly increased in ST2$^+$ T$_{reg}$ cells compared to ST2$^-$ T$_{reg}$ cells mainly in the leukemic niche (Fig. 6C). This data suggests that CXCR4$^+$ST2$^+$ T$_{reg}$ cells are majorly found in the BM during leukemia progression and in agreement with the report that blocking migration of T$_{reg}$ cells with AMD3100 (a CXCR4 inhibitor) delays disease progression in the MLL-AF9 induced leukemia model[41].

## Tbet restrains activation and pro-cytotoxicity of ST2$^+$ T$_{reg}$ cells while Bcl6 reciprocally regulates ST2$^+$ T$_{reg}$ cells in the leukemic niche

The GATA3 transcription factor binds to the ST2 promoter, enhancing expression of ST2 in T$_{reg}$ cells and a Th2-like phenotype[42,43]. T$_{reg}$ cells

can express either the transcription factor Tbet or the transcription factor GATA3, which are transiently upregulated to maintain T cell homeostasis[9,18]. IFN-γ in T$_{reg}$ cells has been shown to promote anti-tumor immunity[44], and Tbet (encoded by the *Tbx21* gene) is required for optimal IFN-γ production in CD4$^+$ T cells[45]. While Tbet can bind to the ST2 promoter in the context of Th1 cells[46], the relationship of Tbet and ST2 in T$_{reg}$ cells is still unknown. Therefore, we investigated whether ST2$^+$ T$_{reg}$ cells are correlated with Tbet expression in both steady and pathogenic states. We hypothesized that BM-WT (ST2$^+$) T$_{reg}$ cells would be inversely correlated with Tbet expression in healthy and leukemic mice. To this purpose, we first performed a nanostring analysis of sorted BM-Tbet$^{-/-}$ T$_{reg}$ cells compared to sorted BM-WT (ST2$^+$) T$_{reg}$ cells in naive mice. The results showed increased transcript levels of key molecules needed for T$_{reg}$ cell function [*Pparg, Il10, Il1rl1 (ST2), Ebi3*] and pro-cytotoxicity molecules such as Gzmb and Gzma in Tbet$^{-/-}$ T$_{reg}$ cells versus WT(ST2$^+$) T$_{reg}$ cells (Supplementary Fig. 14A, Supplementary Table 2, and Supplementary Data 1). Il1rl1 increased expression was confirmed at the protein level with flow cytometry, where the frequency of ST2$^+$ T$_{reg}$ cells is higher in the BM of Tbet$^{-/-}$ mice without a significant change in the total number of T$_{reg}$ cells at steady state (Supplementary Fig. 14B).

We next investigated the role of Tbet in TME BM T$_{reg}$ cells using the MLL-AF9 leukemic model with adoptive transfer of 20% of 3 types of T$_{reg}$ cells: WT (ST2$^+$) versus Tbet$^{-/-}$ versus ST2$^{-/-}$ T$_{reg}$ cells and compared it to the no tumor model (Fig. 7A). Transfer of Tbet$^{-/-}$ T$_{reg}$ cells to leukemic mice results in significantly shortened survival compared with transfer of WT T$_{reg}$ cells ($P = 0.014$), and those mice have shorter survival and higher frequencies of BM-infiltrating MLL-AF9 cells than mice receiving ST2$^{-/-}$ T$_{reg}$ cells ($P < 0.0001$) (Fig. 7B, C). RNA sequencing of day 14 sorted Foxp3$^{eGFP+}$ T$_{reg}$ cells from WT without tumor, WT with tumor, ST2 knockout tumor, and Tbet knockout tumor samples was performed. Transcript expression was represented on a heatmap comparing the 4 groups; it reveals that transcript levels of *Il1rl1, Gzmb, Gzma, Gzmk, Prf1, Eomes, Pparg, Ebi3*, and *Il10* are increased in TME WT (ST2$^+$) T$_{reg}$ cells and even more in TME Tbet$^{-/-}$ T$_{reg}$ cells, while they are decreased in TME ST2$^{-/-}$ T$_{reg}$ cells and non-TME T$_{reg}$ cells (Fig. 7D, Supplementary Table 3). Transfer of Tbet$^{-/-}$ T$_{reg}$ cells to leukemic mice results in significantly higher frequencies of BM-infiltrating ST2$^+$ T$_{reg}$ cells than mice that receive WT T$_{reg}$ cells or ST2$^{-/-}$ T$_{reg}$ cells or no tumor ($P = 0.005$, $P < 0.0001$ and $P < 0.0001$, respectively) (Fig. 5E). Notably, the frequency and cytokines production of the ST2-expressing cells versus ST2-negative cells among all the WT Foxp3$^{eGFP+}$ T$_{reg}$ cells are increased over time in the ST2$^+$ (WT) TME (Supplementary Fig. 15A–D). In the groups that received ST2$^{-/-}$ T$_{reg}$ cells versus WT T$_{reg}$ cells versus Tbet$^{-/-}$ T$_{reg}$ cells, the numbers of proliferating, degranulating, and type 1 activated CD8 T cells are higher while the frequency of exhausted CD8 T cells is lower (Supplementary Fig. 16A–F). Flow cytometry analysis of these T$_{reg}$ cells subsets confirmed increases in frequencies of IL-10 and perforin T$_{reg}$ cells, and frequencies and mean fluorescence intensity (MFI) of GZMA,

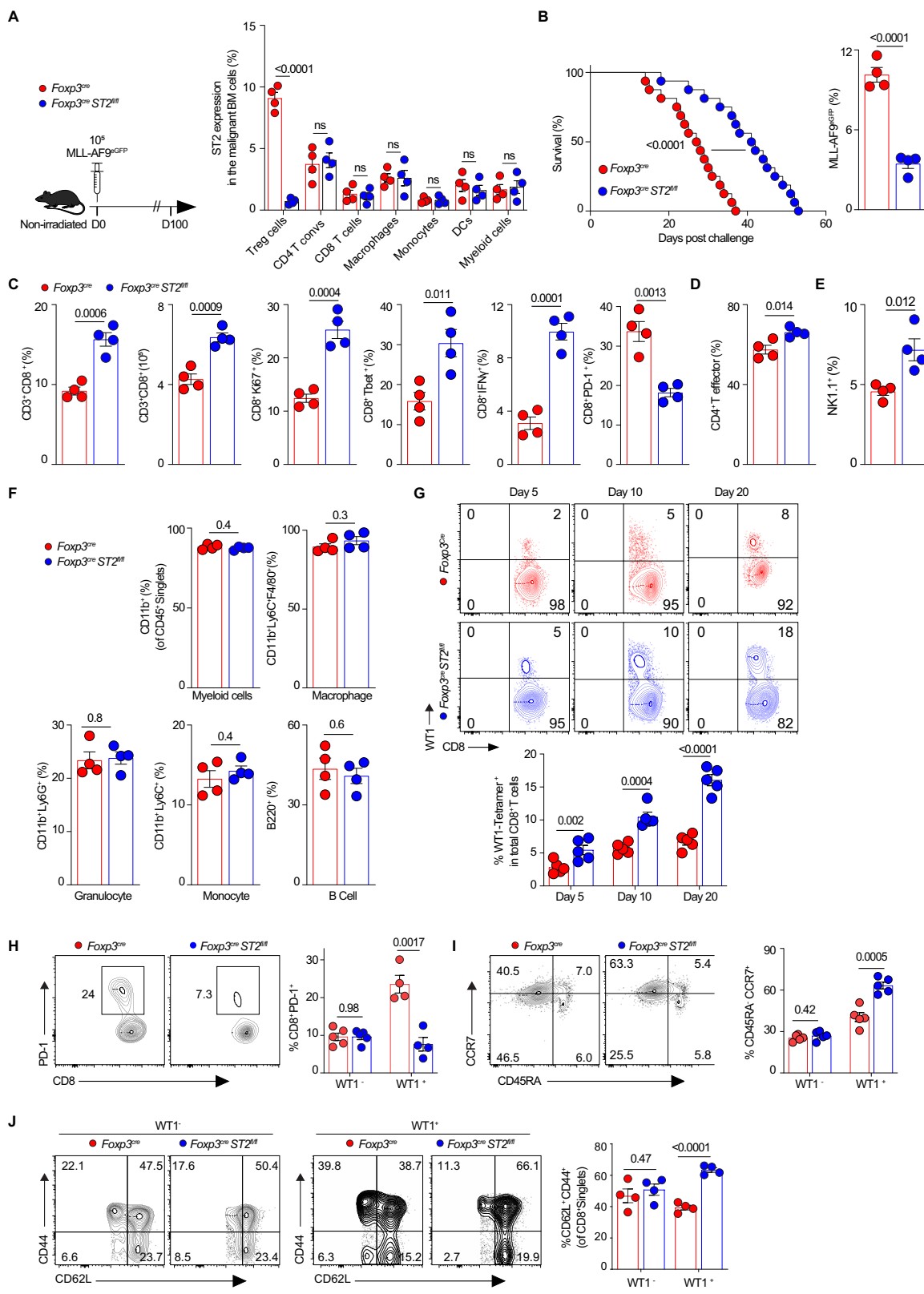

and GZMB expression in both Tbet$^{-/-}$ T$_{reg}$ and WT T$_{reg}$ cells (Fig. 7F, G). Additionally, GZMB expression progressively increases in ST2$^+$ T$_{reg}$ cells from day 5 to day 20 with leukemia development in two leukemia models (Fig. 7G, H). Since we found a negative correlation between Tbet and ST2, we measured Tbet expression in ST2$^-$ T$_{reg}$ cells as compared to ST2$^+$ T$_{reg}$ cells in the malignant BM of mice at different timepoints post-AML challenge in both MLL-AF9 and DNMT3A/FLT3$^{ITD}$

leukemic models and found that Tbet was increased in ST2$^-$ T$_{reg}$ cells (Fig. 7I). This suggests that intrinsic Tbet restrains activation of ST2$^+$ T$_{reg}$ cells. We further quantified ST2 DNA concentrations in real time after chromatin immunoprecipitation (ChiP-qPCR) using anti-Tbet antibody (clone: 39D, Invitrogen) versus IgG1 and IgG2 controls. No ST2 (*Il1rl1*) cDNA in ST2$^+$ T$_{reg}$ cells was detected after ChIP-qPCR with Tbet (Fig. 7J). Thus, we showed here that ST2 expression in T$_{reg}$ cells is

**Fig. 5 | Conditional knockout of ST2 in T_reg cells unleashes potent antitumor immunity, amplifies WT1-specific CD8 T cells responses, and curbs CD8 T cells exhaustion—effectively halting AML progression. A** Schema of non-irradiated MLL-AF9 leukemia model in either *Foxp3^Cre^ST2^fl/fl^* or *Foxp3^Cre^* mice and specific depletion of ST2 in T_reg cells in *Foxp3^Cre^ST2^fl/fl^* mice. Representative experiment verifying ST2-specific deletion in T_reg cells and no other immune cells in *Foxp3^Cre^ST2^fl/fl^* mice (*n* = 4). **B** Survival analysis of naïve *Foxp3^Cre^ST2^fl/fl^* and *Foxp3^Cre^* mice challenged with MLL-AF9^eGFP^ leukemic cells (10⁵ per mouse, *n* = 20) and frequencies of MLL-AF9^eGFP^ on day 14 after leukemic cell challenge (*n* = 4). **C** Statistically analyzed frequencies of CD3⁺CD8⁺, numbers of CD3⁺CD8⁺, frequencies of CD8⁺Ki67⁺, CD8⁺Tbet⁺, CD8⁺IFNγ⁺, and CD8⁺PD-1⁺ among GFP⁻CD3⁺ total T cells in the malignant BM niches on day 14 post MLL-AF9^eGFP^ leukemic cell challenge (10⁵ per mouse) (*n* = 4). Statistically analyzed frequencies of CD4⁺ T effector cells (CD4⁺Foxp3⁻) of CD3⁺CD4⁺ T cells (**D**), NK cells (NK1.1⁺) of CD3⁻ cells (**E**) and myeloid cells (CD11b⁺) including macrophages, granulocytes, monocytes (**F**), and B cells in the malignant BM niches on day 14 post MLL-AF9^eGFP^ leukemic cell challenge (10⁵ per mouse) (*n* = 4). **G** Representative flow plots showing the frequencies of WT1-tetramer positive and negative CD8 T cells in the malignant BM

niches on day 10 post leukemic cell challenge (10⁵ per mouse), and the statistical analysis results for the indicated days (*n* = 5). **H** Representative flow plots showing the frequencies of CD8⁺PD-1⁺ T cells from WT1-tetramer positive CD8 T cells in the malignant BM niches on day 10 post leukemic cell challenge (10⁵ per mouse), and statistical analysis results for CD8⁺PD-1⁺ T cells from WT1-tetramer positive CD8 T cells (*n* = 4). **I** Representative flow plots showing the frequencies of central memory CD8 T cells (identified as CD45RA⁻ CCR7⁺) from WT1-tetramer positive CD8 T cells in the malignant BM niches on day 10 post leukemic cell challenge (10⁵ per mouse), and statistical analysis results for memory CD8 T cells from both WT1-tetramer positive and negative CD8 T cells (*n* = 5). **J** Representative flow plots showing the frequencies of central memory CD8 T cells (identified as CD62L^Hi^ CD44^Hi^ CD8 T cells) gated on tetramer WT1⁺ and WT1⁻, in the malignant BM niches on day 10 post MLL-AF9 leukemic cell challenge (10⁵ per mouse), and statistical analysis results for central memory CD8 T cells from both WT1-tetramer positive and negative CD8 T cells (*n* = 4). The data are presented as means ± s.e.m. (error bar) compared using a two-sided unpaired *t*-test (**A**–**J**), and Kaplan–Meier curves compared using the log-rank test (**B**). *P* values are presented in the figure. Source data are provided as a file.

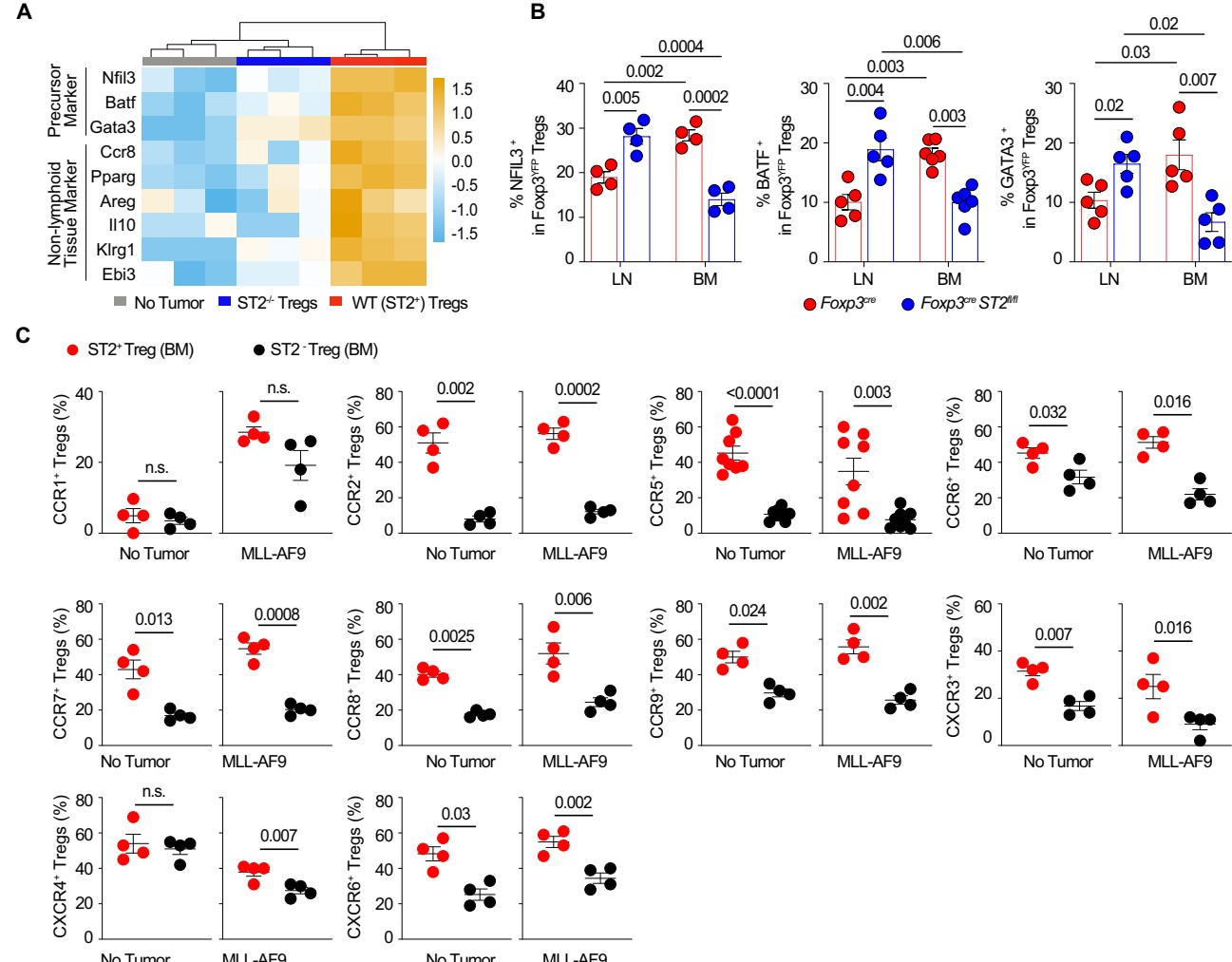

**Fig. 6 | Deficiency of ST2 in T_reg cells leads to their retention in inguinal lymph nodes (LNs) at a precursor stage while reducing their numbers in the BM. A** Bulk RNA sequencing was performed, and clustering heatmap analysis was performed using selected differentially expressed genes related to precursor markers and non-lymphoid tissue activation markers in T_reg cells derived from three different sources on day 14 post leukemic cell challenge (*n* = 3). **B** Statistically analyzed frequencies of NFIL3⁺ (*n* = 4), BATF⁺ (*n* = 5 for LN; *n* = 6 for BM), and GATA3⁺ (*n* = 5) T_reg cells from both LN and BM tissues on day 14 post MLL-AF9 leukemic cell

challenge (10⁵ per mouse). Data are mean ± s.e.m.; unpaired *t*-test was used. **C** Comparisons of percentages of T_reg cells for the indicated chemokine receptors between paired ST2⁺ T_reg cells and ST2⁻ T_reg cells. CCR1 (*n* = 4), CCR2 (*n* = 4), CCR5 (*n* = 8), CCR6 (*n* = 6), CCR7 (*n* = 4), CCR8 (*n* = 4), CCR9 (*n* = 4), CXCR3 (*n* = 4), CXCR4 (*n* = 4) and CXCR6 (*n* = 4) post MLL-AF9 leukemic cell challenge (10⁵ per mouse) on day 10. Data represents three independent replicates. The control group had no tumor cell transfer. A paired *t*-test was used.

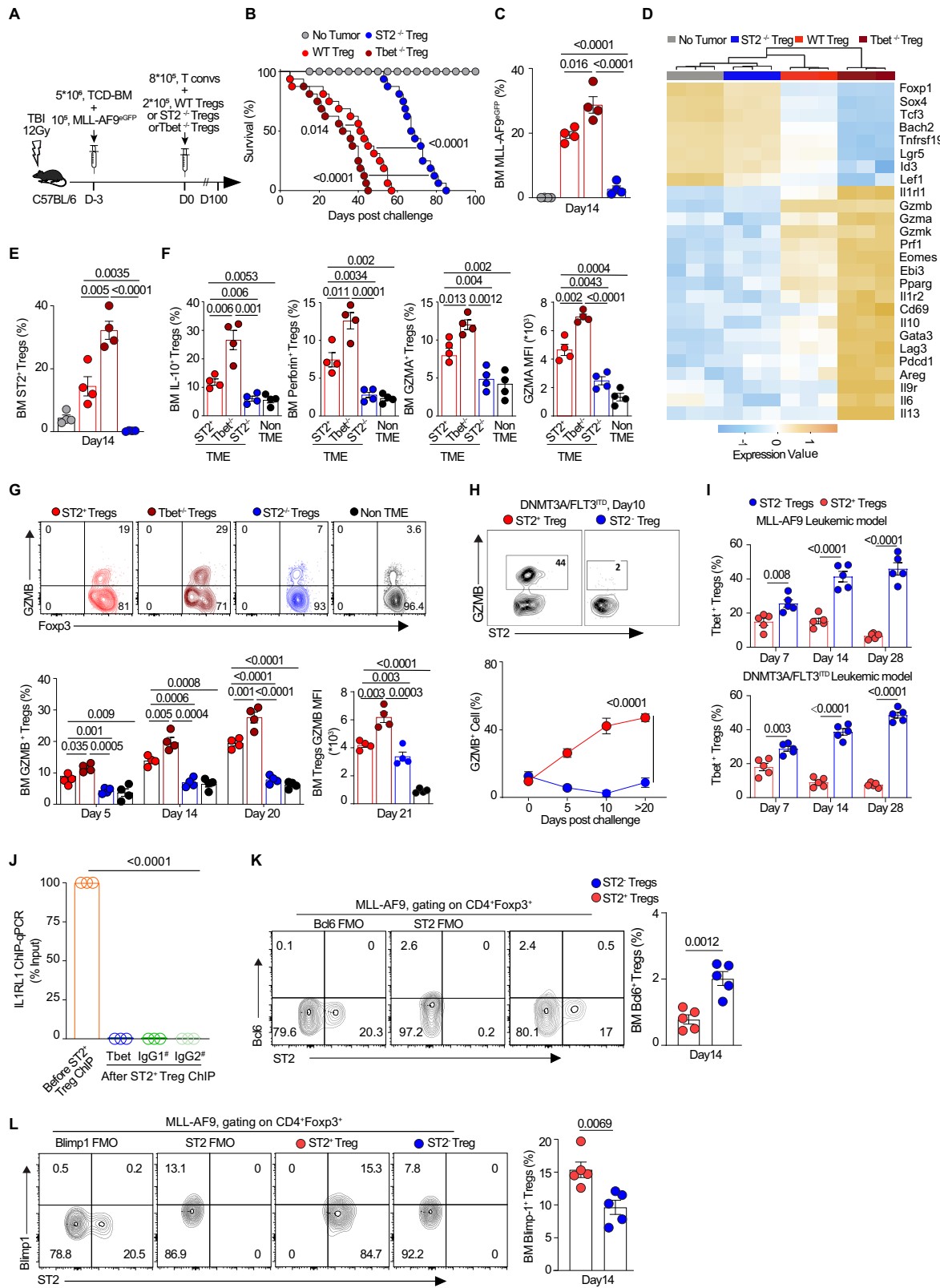

negatively correlated with Tbet without evidence showing Tbet is associated with ST2 expression in $T_{reg}$ cells.

Furthermore, Bcl6 has been shown to inhibit Th2 activity of $T_{reg}$ cells by repressing Gata3 function[47]. Additionally, the development of ST2+ $T_{reg}$ cells is also reciprocally regulated by Bcl6 and Blimp1 in the context of allergic airway inflammation[48]. Thus, we hypothesized that Bcl6 inhibits and Blimp1 promotes ST2 expression in leukemic $T_{reg}$

cells. To this end, we analyzed Bcl6 and Blimp1 transcription factors by flow cytometry in ST2+ $T_{reg}$ cells from leukemic BM, 14 days post-AML challenge, and found that the percentage of Bcl6-expressing $T_{reg}$ cells is decreased in ST2+ $T_{reg}$ cells compared to ST2- $T_{reg}$ cells (Fig. 7K). In contrast, Blimp1 expression in ST2+ $T_{reg}$ cells is decreased as compared to ST2- $T_{reg}$ cells (Fig. 7L). In summary, Bcl6 expression inversely correlates with ST2 expression in leukemic ST2+ $T_{reg}$ cells, and Blimp1

**Fig. 7 | Intrinsic Tbet restrains activation and pro-cytotoxicity of ST2$^+$ T$_{reg}$ cells while Bcl6 reciprocally regulates ST2$^+$ T$_{reg}$ cells in the leukemic niche.**
**A** Schematic of mice with or without MLL-AF9 injection and with adoptive transfer of 20% WT or ST2$^{-/-}$ or Tbet$^{-/-}$ T$_{reg}$ cells and 80% of Foxp3$^-$ T cells. T$_{reg}$ cells were sorted from single-cell suspensions of normal BM or malignant BM niches by gating CD4$^+$CD25$^+$ cells with CD127$^-$ (for Tbet$^{-/-}$ T$_{reg}$ cell isolation) or eGFP$^+$ (for WT T$_{reg}$ cell and ST2$^{-/-}$ T$_{reg}$ cell). **B** Survival analysis of leukemic mice that received adoptive transfer of WT T$_{reg}$ cells ($n = 15$) or ST2$^{-/-}$ T$_{reg}$ cells ($n = 16$) or Tbet$^{-/-}$ T$_{reg}$ cells ($n = 16$) (20% of total T cells) and MLL-AF9$^{eGFP}$ leukemic cells ($10^5$ per mouse); control group received no tumor cell transfer ($n = 18$). Kaplan–Meier curves with log-rank test to compare the groups. **C** Statistical analysis of the MLL-AF9$^{eGFP}$ leukemic cells ($n = 4$) on day 14 post leukemic cell challenge in the 4 Treg cell groups. Data are mean ± s.e.m.; unpaired $t$-test was used. **D** Heatmap of RNA sequencing analysis of T$_{reg}$ cells sorted from malignant BM of mice in which ST2$^{-/-}$ T$_{reg}$ cells versus WT (ST2$^+$) T$_{reg}$ cells versus Tbet$^{-/-}$ T$_{reg}$ cells versus no tumor cells were transferred. RNA sequencing of increased or decreased transcripts of key molecules needed for T$_{reg}$ cell activation, tissue repair, and pro-cytotoxicity function on day 14 post leukemic cell challenge. Data are visualized as hierarchical clustering of normalized log$_2$-transformed fold change ($n = 3$ per group). **E** Statistical analysis of ST2$^+$ T$_{reg}$ cells on day 14 post leukemic cell challenge ($n = 4$) on day 14 post leukemic cell challenge in the 4 T$_{reg}$ cell groups. Data are mean ± s.e.m.; unpaired $t$-test was used. **F** Statistically analyzed frequencies of BM-derived IL-10$^+$ T$_{reg}$ cells, perforin$^+$ T$_{reg}$ cells, and GZMA$^+$ T$_{reg}$ cells among the total T$_{reg}$ cells and GZMA MFI at day 14 post leukemic cell challenge in groups that received transfer of ST2$^{-/-}$ T$_{reg}$ cells versus WT (ST2$^+$) T$_{reg}$ cells versus Tbet$^{-/-}$ T$_{reg}$ cells versus no tumor cells ($n = 4$).

Data are mean ± s.e.m.; ANOVA with post-hoc Bonferroni $t$-test, unpaired $t$-test was used. **G** Representative flow plots and frequencies of BM-derived GZMB$^+$ T$_{reg}$ cells among the total T$_{reg}$ cells and GZMB MFI at day 20 post leukemic cell challenge in groups that received transfer of ST2$^{-/-}$ T$_{reg}$ cells versus WT (ST2$^+$) T$_{reg}$ cells versus Tbet$^{-/-}$ T$_{reg}$ cells versus no tumor cells; and statistical analysis results for the BM-derived GZMB$^+$ T$_{reg}$ cells on the indicated days ($n = 4$). Data are mean ± s.e.m.; ANOVA with post-hoc Bonferroni $t$-test and two-sided unpaired $t$-test was used. **H** Representative flow plots showing the GZMB$^+$ percentages among both ST2$^+$ T$_{reg}$ cells ($n = 6$) and ST2$^-$ T$_{reg}$ cells ($n = 4$) on day 10 post DNMT3A/FLT3$^{ITD}$-mutant leukemic cell challenge, and statistical analysis of data on indicated days. Data are mean ± s.e.m. An unpaired $t$-test was used. **I** Frequencies of the BM-derived Tbet$^+$ T$_{reg}$ cells on day 7, day 14, and day 28 post-AML challenge in both MLL-AF9 leukemic model and DNMT3A/FLT3$^{ITD}$-mutant leukemic model ($n = 5$/ model). Data are mean ± s.e.m.; an unpaired two-sided $t$-test was used. **J** *IL1RL1* enrichment in ST2$^+$T$_{reg}$ cells using anti-Tbet antibody (clone: 39D, Invitrogen) versus IgG1 (provided in the ChIP kit) and IgG2 (clone MG1-45, BioLegend) controls via ChIP-qPCR. Data are presented as percentages from the input of *IL1RL1* cDNA in ST2$^+$T$_{reg}$ cells before versus after ChIP, calculated using the comparative Ct method. Data are mean ± s.e.m. ($n = 3$). ANOVA with post-hoc Bonferroni $t$-test was used.
**K** Representative flow plot, FMOs, and frequency of Bcl6 expression in ST2$^+$ and ST2$^-$ T$_{reg}$ cells in the malignant BM post-AML challenge at Day 14 ($n = 5$). Data are mean ± s.e.m.; an unpaired two-sided $t$-test was used. **L** Representative flow plot, FMOs, and frequency of Blimp1$^+$ cells in ST2$^+$ and ST2$^-$ T$_{reg}$ cells in the malignant BM post-AML challenge at Day 14 ($n = 5$). Data are mean ± s.e.m.; an unpaired two-sided $t$-test was used.

positively correlates with increased ST2 expression in leukemic BM T$_{reg}$ cells.

## ST2$^+$ T$_{reg}$ cells kill intratumoral CD8 T cells via granzyme B

Since T$_{reg}$ cells derived from the TME were shown to induce CD8 T cell death via GZMB release[49], we investigated whether leukemic BM-derived ST2$^+$ T$_{reg}$ cells expressing high levels of GZMB can directly kill tumor effector lymphocytes (CD8$^+$, CD4$^+$Foxp3$^-$, and NK cells). We first performed coculture of in vitro AML-activated BM ST2$^+$ T$_{reg}$ cells or non-activated ST2$^-$ T$_{reg}$ cells with BM leukemic CD8 T cells stained with SYTOX, a high-affinity nucleic acid that penetrates cells with compromised plasma membranes, in the presence or not of IL-33 and with or without a specific inhibitor of GZMB, Z-AAD-CMK. SYTOX release by CD8 T cells is significantly higher when cocultured with AML-activated BM ST2$^+$ T$_{reg}$ cells as compared to non-activated ST2$^-$ T$_{reg}$ cells and this killing is increased in presence of Il-33 and decreased when GZMB is inhibited (Fig. 8A). We then used transwell assays of CD8 T cells with AML-activated BM ST2$^+$ T$_{reg}$ cells and showed that AML-activated BM ST2$^+$ T$_{reg}$ cells are able to kill BM leukemic CD8 T cells in a contact-dependent manner (no killing in transwell assays; Fig. 8B). Notably, this killing is not observed with either leukemic cells or other myeloid cells or B cells (Fig. 8C).

We next performed coculture of ex vivo sorted ST2$^+$ versus ST2$^-$ T$_{reg}$ cells from leukemia mice with BM leukemic CD8 T cells from the same mice and stained the CD8 T cells with SYTOX at 2:1 and 1:1 T$_{reg}$:CD8 ratios. The direct cytotoxicity of ST2$^+$ T$_{reg}$ cells through GZMB was confirmed by imaging studies in which ST2$^+$ T$_{reg}$ cells but not ST2$^-$ T$_{reg}$ cells with or without IL-33 showed SYTOX positivity among CD8 T cells, whereas Z-AAD-CMK abolished SYTOX staining. A higher coculture ratio of ST2$^+$ T$_{reg}$ cells with CD8 T cells resulted in higher killing (Fig. 8D). At a T$_{reg}$ cells to CD8 ratio of 1:1, the difference in killing between IL-33 stimulated versus non-stimulated ST2$^+$ T$_{reg}$ cells is moderate, which suggests they have already acquired the full pro-cytotoxicity machinery. GZMB-mediated killing was also detected with CD4$^+$ T conventional cell cocultures, while ST2$^+$ T$_{reg}$ cells may impair NK cell survival not only through immunosuppressive cytokines such as IL-10 and TGFβ but also potentially via a granzyme B-dependent cytotoxic mechanism, as indicated by the reversal of NK cell death upon Z-AAD-CMK treatment[50] (Supplementary Fig. 17A, B). To investigate whether ST2$^+$ T$_{reg}$ cells preferentially kill TME-derived

CD8 T cells, we cocultured ex vivo sorted ST2$^+$ T$_{reg}$ cells from leukemia mice with BM-derived CD8 T cells from naïve versus non-tumoral versus tumoral mice at increasing T$_{reg}$:CD8 ratios. ST2$^+$ T$_{reg}$ cells killed TME-derived CD8 T cells at up to 30% at a ratio of 4:1. They exclusively lysed TME-derived CD8 T cells and not non-TME- or naïve BM-derived CD8 T cells (Fig. 8E). Furthermore, ST2$^+$ T$_{reg}$ cells from other leukemic sites such as the spleen are also able to kill TME CD8 T cells suggesting the phenomenon is not specific to BM ST2$^+$ T$_{reg}$ cells but to all leukemic ST2$^+$ T$_{reg}$ cells (BM, spleen, liver) (Fig. 8F). We used the retroviral MLL-AF9 in vivo immunocompetent pre-clinical model for most of our experiments; however, we confirmed all key findings in the DNMT3A/FLT3$^{ITD}$ leukemic model and found that DNMT3A/FLT3$^{ITD}$ leukemic BM ST2$^+$ T$_{reg}$ cells display a similar killing of TME CD8 T cells (Fig. 8G).

We also tested if leukemic antigenic activation versus global TCR activation by CD3/CD8 was needed for the killing and noted that only AML-activated CD8 T cells were killed by AML-activated ST2$^+$ T$_{reg}$ cells (Fig. 8H). Next, to verify ST2$^+$ T$_{reg}$ cell killing of TME CD8 T cells in vivo, we used an antigen-specific killing assay[51]. Leukemic BM CD8 T cells taken 14 days post leukemic cells challenge are labeled with CFSE$^{high}$ mixed with CFSE$^{low}$ non-TME CD8 T cells as target and adoptively transferred with sorted leukemic BM ST2$^+$ T$_{reg}$ cells or WT naïve ST2$^-$ T$_{reg}$ cells at a ratio of 1:1 into mice carrying leukemia. At 24 h post transfer, only leukemic BM CD8 T cells (CFSE$^{high}$) in the presence of TME ST2$^+$ T$_{reg}$ cells are proportionally decreased in comparison to CFSE$^{low}$ non-TME CD8 T cells that are increased, which suggests TME CD8 T cells have been killed by TME ST2$^+$ T$_{reg}$ cells (Fig. 8I, middle panel). At 24 h post transfer, lysis of CD45.1$^+$CFSE$^{high}$ TME CD8 T cells was calculated and was six-fold higher in the presence of TME ST2$^+$ T$_{reg}$ cells than in the presence of WT naïve ST2$^-$ T$_{reg}$ cells (Fig. 8I, right panel). Collectively, these data indicate that leukemia-primed ST2$^+$ T$_{reg}$ cells kill in vitro and in vivo, via direct contact and a GZMB-mediated lysis of antigen-specific CD8 lymphocytes that have been continuously stimulated by the leukemia.

## Engineered anti-ST2 antibody promotes abatement of ST2$^+$ T$_{reg}$ cells to extend survival in AML models

For translational purposes, we used the sequences of two ST2 blocking antibodies engaging mouse or human ST2[33,52] that we engineered on the IgG[L]-scFv platform where the aspartic acid residue was substituted with alanine at amino acid position 265 for Fc silencing,

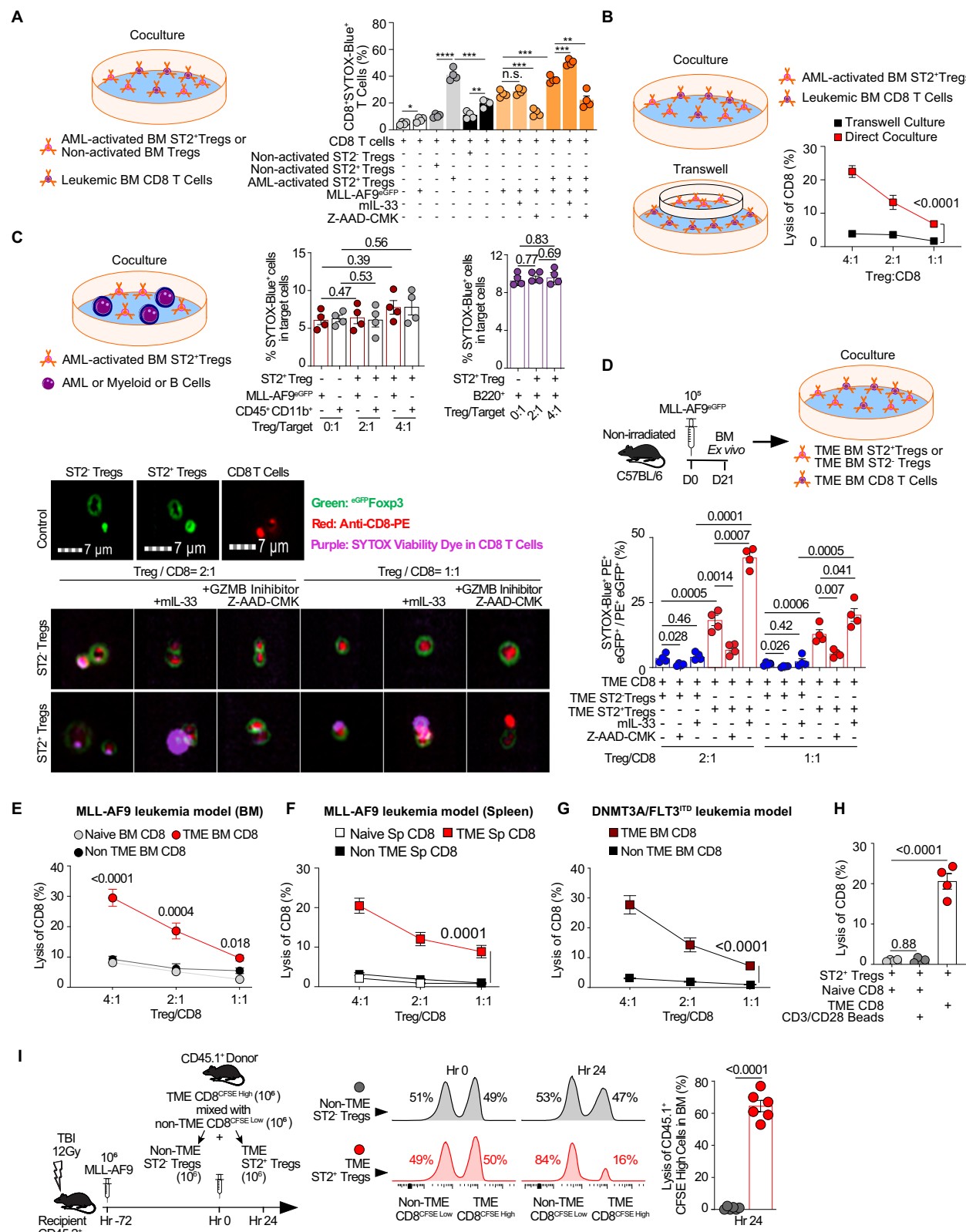

thereby eliminating non-specific FcγR binding on stromal cells and antigen-presenting cells (Fig. 9A, O)[53]. Silencing Fc domains in T cell–directed antibodies improves T cell trafficking and antitumor potency[54]. HPLC analysis verified the purity of ST2 blocking antibodies (Supplementary Fig. 18A), surface plasmon resonance (SPR) microscopy analysis revealed strong antibody affinities with slow $K_{off}$ values, and pharmacokinetics in vivo indicated long half-lives and good area

under the curve (Supplementary Fig. 18B, C). To first understand the mechanisms of action of anti-ST2 antibodies on ST2$^+$ T$_{reg}$ cells, IgG-281 was directly applied to ST2$^+$ T$_{reg}$ cells in a dose-dependent manner, and ST2$^+$ T$_{reg}$ cell apoptosis and proliferation were monitored. IgG-281 suppresses ST2$^+$ T$_{reg}$ cells via apoptosis and inhibition of proliferation but do not have the same effects on Foxp3$^+$ST2$^-$ T$_{reg}$ cells (Fig. 9B), or have any direct killing effect on CD8 T cells (Supplementary Fig. 19).

**Fig. 8 | ST2$^+$ T$_{reg}$ cells kill intratumoral (TME) CD8 T cells in the leukemic niche.** **A** In vitro direct killing assay of TME BM CD8 T cells by TME BM ST2$^+$ T$_{reg}$ cells. Coculture of AML-activated BM ST2$^+$ T$_{reg}$ cells or non-activated BM ST2$^-$ T$_{reg}$ cells with SYTOX-Blue viability dye-stained TME BM CD8 T cells at a ratio of 1:1, treated with or without IL-33 (20 ng/ml) or GZMB inhibitor Z-AAD-CMK (50 ng/ml). Data are mean ± s.e.m. ($n = 4$); an unpaired two-sided $t$-test was used. **B** In vitro direct killing versus through transwell assays. Coculture or transwell of AML-activated BM ST2$^+$ T$_{reg}$ cells or non-activated BM ST2$^-$ T$_{reg}$ cells with SYTOX-Blue viability dye-stained TME BM CD8 T cells at ratios ST2$^+$ T$_{reg}$ cells: CD8 of 4:1, 2:1, and 1:1. Data are mean ± s.e.m. ($n = 4$); unpaired two-sided $t$-test was used. **C** ST2$^+$ T$_{reg}$ cells do not directly kill leukemic cells, myeloid cells, or B cells. Killing ability of TME-derived BM ST2$^+$ T$_{reg}$ cells in coculture at the indicated ratios with MLL-AF9$^{eGFP}$ leukemic cells, myeloid cells (CD45$^+$CD11b$^+$), and B cells (B220$^+$). Data are mean ± s.e.m. ($n = 4$); an unpaired two-sided $t$-test was used. **D** Ex vivo direct killing assay of TME BM CD8 T cells by TME BM ST2$^+$ T$_{reg}$ cells. Representative flow images of Foxp3$^{eGFP}$ (Green) T$_{reg}$ cells, anti-CD8-PE T cells (Red), and SYTOX-Blue AML cells (Purple) immunofluorescence in the cocultures of TME BM-derived CD8 T cells and TME BM-derived T$_{reg}$ cells at the indicated ratios, treated with or without IL-33 (20 ng/ml) or GZMB inhibitor Z-AAD-CMK (50 ng/ml). Statistically analyzed lysis percentages of CD8 T cells killed by surrounding T$_{reg}$ cells (SYTOX-Blue$^+$ PE$^+$ eGFP$^+$) from each coculture group ($n = 4$). Data are mean ± s.e.m.; an unpaired two-sided $t$-test was used. **E** Coculture of TME BM ST2$^+$ T$_{reg}$ cells with CD45RA-sorted naïve CD8 T cells, non-TME-derived BM CD8 T cells, or TME-derived BM CD8 T cells from the MLL-AF9 leukemia model, at the indicated ratios and comparison of the cytolytic effect of these ST2$^+$ T$_{reg}$ cells on these different CD8 T cells ($n = 4$). Naïve CD8 T cells were sorted for the CD45RA marker from naïve mice; non-TME CD8 T cells were sorted from naïve mice; TME CD8 T cells were sorted from the MLL-AF9 leukemic mice on day 10. These three different CD8 T cell populations were used in killing assays. Data are mean ± s.e.m.; an unpaired two-sided $t$-test was used. **F** Splenic ST2$^+$ T$_{reg}$ cells kill leukemic spleen-derived CD8$^+$T cells in the MLL-AF9 model. Coculture of TME Spleen (Sp) ST2$^+$ T$_{reg}$ cells with CD45RA-sorted naïve CD8 T cells or non-

TME-derived Sp CD8 T cells, or TME-derived Sp CD8 T cells from the MLL-AF9 leukemia model, at the indicated ratios and comparison of the cytolytic effect of these Sp ST2$^+$ T$_{reg}$ cells on these different CD8 T cells ($n = 4$). Naïve CD8 T cells were sorted for the CD45RA marker from naïve mice; non-TME CD8 T cells were sorted from naïve mice; TME CD8 T cells were sorted from the MLL-AF9 leukemic mice on day 10. These three different CD8 T cell populations were used in killing assays. Data are mean ± s.e.m.; an unpaired two-sided $t$-test was used. **G** ST2$^+$ T$_{reg}$ cells kill DNMT3A/FLT3$^{ITD}$ leukemic BM-derived CD8$^+$T cells. Coculture of TME BM ST2$^+$ T$_{reg}$ cells with non-TME-derived BM CD8 T cells, or TME-derived Spleen CD8 T cells from the DNMT3A/FLT3$^{ITD}$ leukemia model, at the indicated ratios and comparison of the cytolytic effect of these ST2$^+$ T$_{reg}$ cells on these different CD8 T cells ($n = 4$). Non-TME CD8 T cells were sorted from naïve mice; TME CD8 T cells were sorted from the DNMT3A/FLT3$^{ITD}$ leukemic mice on day 10. These two different CD8 T cell populations were used in killing assays. Data are mean ± s.e.m.; an unpaired two-sided $t$-test was used. **H** Coculture of ST2$^+$ T$_{reg}$ cells with CD45RA-sorted naïve CD8 T cells, TME-derived CD8 T cells, and CD3/CD8 activated CD8 T cells (beads to cell ratio of 1:1) at T$_{reg}$/CD8 ratio of 1:1 and comparison of the cytolytic effect of ST2$^+$ T$_{reg}$ cells on these CD8 T cells ($n = 4$). Non-TME or TME cell populations were sorted from naïve or leukemic mice on day 10 and used in killing assays. Data are mean ± s.e.m.; ANOVA with post-hoc Bonferroni $t$-test. **I** In vivo killing assay of TME BM CD8 T cells by TME BM ST2$^+$ T$_{reg}$ cells. Schema and data of an assay to evaluate T$_{reg}$ cells' in vivo cytotoxicity of TME and non-TME CD8 T cells. Representative histograms showing the frequencies of donor-derived TME CD8 T cells (CFSE$^{high}$) and non-TME CD8 T cells (CFSE$^{low}$) in the BM at 24 h post-adaptive transfer with WT naïve ST2$^-$ T$_{reg}$ cells or TME AML ST2$^+$ T$_{reg}$ cells. The red histograms represent the in vivo proportion of non-TME CD8 T cells (CFSE$^{low}$) and TME CD8 T cells (CFSE$^{high}$) after adoptive transfer at baseline (0 h; equal proportion) and at 24 h (decreased proportion in the population that has been killed). Statistically analyzed lysis percentages of donor-derived TME CD8 T cells (CD45.1$^+$CFSE$^{high}$) in the malignant BM niches after 24 h of coculture with WT naïve ST2$^-$ T$_{reg}$ cells or TME BM ST2$^+$ T$_{reg}$ cells ($n = 6$). Data are mean ± s.e.m.; an unpaired two-sided $t$-test was used.

Based on the pharmacokinetics data for anti-murine ST2 (IgG-281) and anti-human ST2 (IgG-282) neutralizing antibodies aforementioned, a dose of 50 μg per injection was administered every 3 days for a total of 3–4 doses in the immunocompetent MLL-AF9 leukemia mice and humanized MOLM-14 NSG mice (Fig. 9C, M–O). The in vivo data further confirm that IgG-281 induces remarkable apoptosis (Annexin+) of intra-leukemic ST2$^+$ T$_{reg}$ cells at days 7 and 14, decreasing their frequencies and GZMB expression (Fig. 9D). In immunocompetent MLL-AF9 mice, anti-murine ST2 treatment with IgG-281 causes marked extension of leukemia survival as compared with the IgG1-treated control group (Fig. 9E). Consistently, the frequencies of leukemic cells are decreased (Fig. 9F) while both the numbers and frequencies of CD3$^+$CD8$^+$ T cells, and frequencies of CD8$^+$Ki67$^+$ are markedly increased (Fig. 9G). Frequencies of CD8$^+$IFNγ$^+$, and CD8$^+$Tbet$^+$ T cells are also increased (Fig. 9H). Finally, frequencies of WT1$^+$CD8$^+$ T cells are noticeably increased (Fig. 9I). As hypothesized, exhausted CD8 T cells (PD-1$^+$, TIM3$^+$, and PD-1$^+$TIM3$^+$) are decreased in the IgG-281-treated group (Fig. 9J). NK cells are also increased in the IgG-281-treated group (Fig. 9K).

Since in normal mice ST2$^+$ T$_{reg}$ cells are frequent in the gastrointestinal (GI) tract, we monitored potential GI toxicity by following body weight, colon tract length, the frequencies of GI total Foxp3$^+$ T$_{reg}$ cells and ST2$^+$ T$_{reg}$ cells, and the presence of infiltrating GI CD8 T cells, GI CD4 and IL-17–secreting CD4 T cells. Body weight and colon length are not different between the anti-ST2 versus isotype control groups (Fig. 9L). While GI T$_{reg}$ cells and ST2$^+$ T$_{reg}$ cells are slightly decreased on day 7, the GI T$_{reg}$ cells and ST2$^+$ T$_{reg}$ cells frequencies are recovered on day 14 in the anti-ST2 treated group (Fig. 9L). Additionally, no significant differences in the infiltrating GI CD8 T cells, GI CD4, and IL-17–secreting CD4 T cells are observed between the two groups (Fig. 9L).

We then questioned if the addition of an anti-PD-1 checkpoint inhibitor would further delay leukemia growth and prolong survival. The combination of IgG-281 and anti-PD-1 treatment in MLL-AF9

leukemic cell-bearing mice further statistically prolongs survival to up to 55 days as compared to IgG-281 and control treatments with significant decrease of frequencies of MLL-AF9 leukemic cells in the malignant BM niches as well as remodeling of TME immune cells (Fig. 9M; Supplementary Fig. 20A–C). MLL-AF9 at 10$^4$ leukemic cells is an aggressive model explaining why anti-ST2 antibody only delay leukemia progression without full rescue of the mice. Thus, we posited that rescue would be possible if a lower leukemia burden were inoculated. Using a dose of 10$^3$ leukemic cells, we observed that anti-ST2 IgG-281 is able to eliminate all leukemic cells and salvage all mice. In this low MLL-AF9 leukemia burden, anti-murine ST2 treatment with IgG-281 causes no death for a follow-up of 180 days compared to the IgG1-treated control group with all the death happening before day 60 post-challenge (Fig. 9N) and leukemic cells are undetectable on day 35 in the IgG-281 treated mice, suggesting elimination of all leukemia and full rescue of the mice (Fig. 9N). As expected, BM ST2$^+$ T$_{reg}$ cells are significantly decreased while total BM T$_{reg}$ cells are preserved, showing specific in vivo inhibition of ST2$^+$ T$_{reg}$ cells in the BM (Fig. 9N). Moreover, frequencies of Tbet$^+$ and Bcl6$^+$ T$_{reg}$ cells are increased in IgG-281-treated mice, whereas Blimp1$^+$ T$_{reg}$ cells are decreased (Fig. 9N). Study of BM CD8 T cells shows that frequencies of CD3$^+$CD8$^+$, CD8$^+$Ki67$^+$, and CD8$^+$Tbet$^+$ T cells are significantly increased while frequencies of CD8$^+$PD-1$^+$ T cells are decreased in the IgG-281-treated mice (Fig. 9N).

We next modeled human leukemia by grafting MOLM-14$^{eGFP}$ cells in NOD.Cg-Prkdc$^{scid}$ Il2rg$^{tm1wjl}$/SzJ (NSG) mice. Furthermore, in order to humanize our model and to make it immunologically relevant to our findings, we adoptively transferred both 10$^6$ human CD8 T cells and 2 × 10$^5$ human ST2$^+$ T$_{reg}$ cells. We then used a similar regimen for the anti-human ST2 (IgG-282) antibody as in the immunocompetent model (Fig. 9O). MOLM-14$^{eGFP}$ levels in the peripheral blood at day 7, 14, 21, and the BM at day 21 are decreased (Fig. 9P), leading to an extended overall survival of the IgG-282-treated group versus the IgG1 control group (Fig. 9P). As in the murine model, the frequencies of BM apoptotic AnnexinV$^+$ST2$^+$Foxp3$^+$ T$_{reg}$ cells, total ST2$^+$Foxp3$^+$ T$_{reg}$ cells and

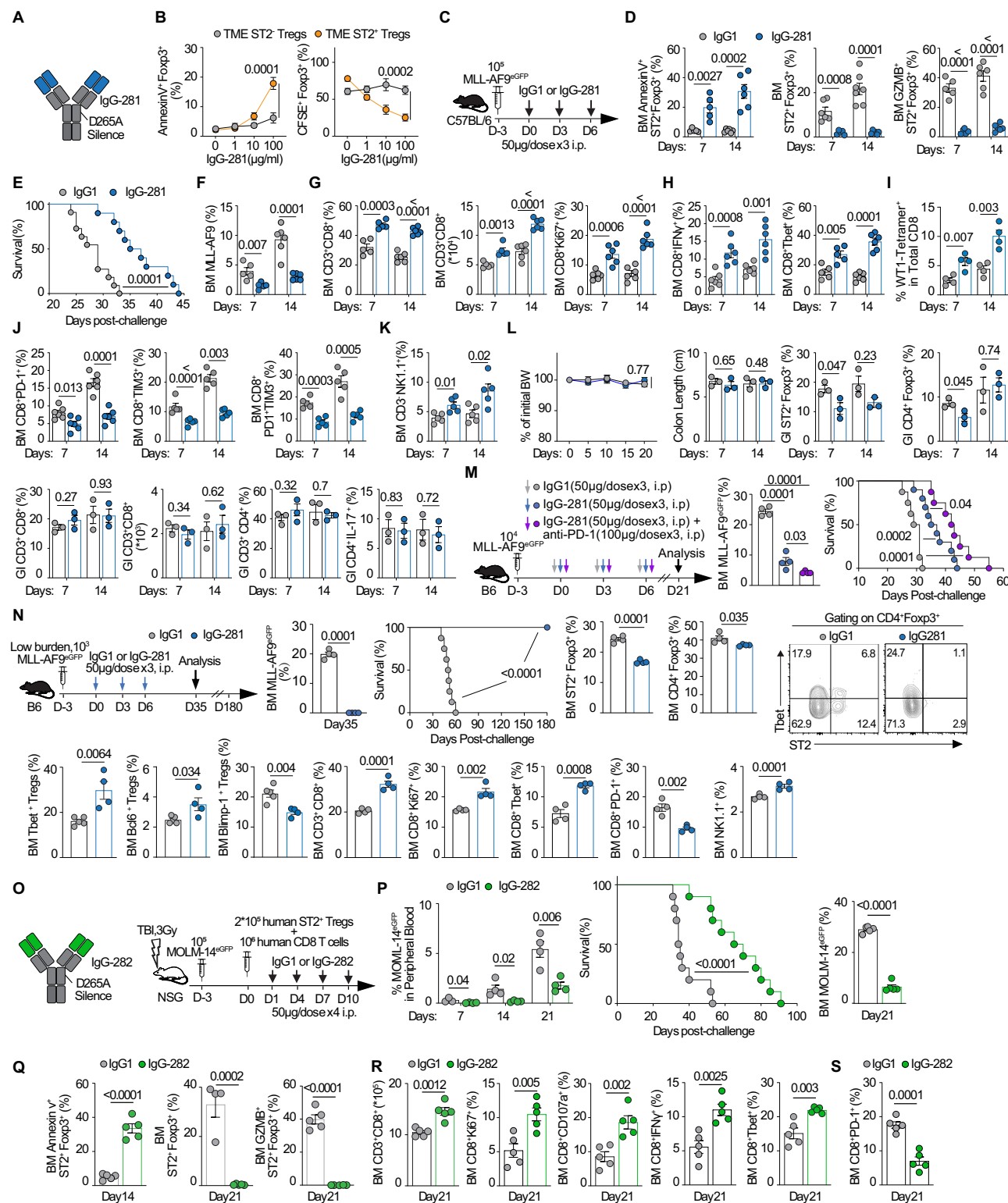

GZMB$^+$ST2$^+$ Foxp3$^+$ T$_{reg}$ cells are decreased (Fig. 9Q), while the number of BM CD8 T cells, frequencies of CD8$^+$Ki67$^+$, CD8$^+$CD107a$^+$, CD8$^+$IFNγ$^+$, and CD8$^+$Tbet$^+$ T cells are increased in the IgG-282-treated group (Fig. 9R). As expected, the frequency of exhausted human CD8 T cells is significantly decreased in the IgG-282-treated humanized leukemic model (Fig. 9S). Since there is a graft-versus-leukemia (GVL) effect with MOLM-14 in NSG mice[55], we further sought to HLA-match the T cells injected and the MOLM-14 leukemia. We first tested MOLM-14 for HLA-A2, which was positive although as expected at a lower level than the

control due to downregulation of the expression of HLA class I (Supplementary Fig. 21A). We then modeled MOLM-14$^{eGFP}$ leukemia in NSG mice using the same design as the model above with HLA-A2 matching of all T cells that diminish a potential GVL effect (Supplementary Fig. 21B). Almost identical survival, tumor burden and frequencies of BM-infiltrating immune cells are observed as compared to the non-HLA matched model (Supplementary Fig. 21C–J). Overall, these data suggest that anti-ST2-neutralizing antibodies limit the progression of aggressive leukemia and rescue it at a lower burden of leukemia by

**Fig. 9 | Engineered anti-murine and human ST2 antibodies promote abatement of ST2$^+$ T$_{reg}$ cells to extend survival in immunocompetent and humanized AML models. A** Structural schematic of murine ST2-neutralizing antibody (IgG-281). **B** In vitro apoptosis and proliferation of TME-derived ST2$^+$ T$_{reg}$ cells and ST2$^-$ T$_{reg}$ cells treated with increasing IgG-281 concentrations ($n = 5$). **C** In vivo schematic of administration of IgG control or IgG-281 in leukemia mice. **D** In vivo apoptosis of ST2$^+$ T$_{reg}$ cells driving abatement of ST2$^+$ T$_{reg}$ cells and GZMB$^+$ST2$^+$ T$_{reg}$ cells in the malignant BM niches on days 7 and 14 after the first administration of IgG1 control or IgG-281. For AnnexinV$^+$ST2$^+$Foxp3$^+$ T$_{reg}$ cells, $n = 5$ on day 7, $n = 6$ on day 14; for ST2$^+$Foxp3$^+$ T$_{reg}$ cells, $n = 6$ on day 7, $n = 7$ on day 14; for GZMB$^+$ST2$^+$ T$_{reg}$ cells, $n = 5$ on day 7, $n = 6$ on day 14. Data represents three independent replicates. **E** Survival analysis of MLL-AF9 leukemic cell-bearing mice treated with IgG1 control ($n = 9$) or IgG-281 ($n = 10$). **F** Frequencies of MLL-AF9 leukemic cells in the malignant BM niches on days 7 ($n = 5$) and 14 ($n = 6$) after the first administration of IgG1 control or IgG-28. **G** Frequencies and absolute numbers of CD3$^+$CD8$^+$ T cells on days 7 ($n = 5$) and 14 ($n = 6$), and frequencies of proliferating Ki67$^+$CD8 T cells ($n = 6$). Cells collected from the malignant BM niches after the first administration of IgG1 control or IgG-281. **H** Frequencies of CD8$^+$IFNγ$^+$ ($n = 6$) and CD8$^+$Tbet$^+$ in the malignant BM niches on days 7 ($n = 5$) and 14 ($n = 6$) after the first administration of IgG1 control or IgG-281. **I** Frequencies of antigen-specific CD8$^+$WT1$^+$ T cells. Cells collected from the malignant BM niches on days 7 and 14 after the first administration of IgG1 control or IgG-281 ($n = 4$). **J** Frequencies of exhausted CD8$^+$PD-1$^+$ ($n = 6$), CD8$^+$TIM3$^+$ ($n = 5$), and CD8$^+$PD-1$^+$TIM3$^+$ ($n = 5$) T cells in the malignant BM niches on days 7 and 14 after the first administration of IgG1 control or IgG-281. **K** Frequencies of CD3$^-$NK1.1$^+$ NK cells in the malignant BM niches on days 7 and 14 after the first administration of IgG1 control or IgG-281 ($n = 5$). **L** Evaluation of the absence of gastrointestinal (GI) colitis by measurement of body weight (BW) changes ($n = 10$), colon tract length on days 7 and 14 ($n = 3$); frequencies of GI-infiltrating ST2$^+$ T$_{reg}$ cells and total T$_{reg}$ cells on days 7 and 14 after the first administration of IgG1 control or IgG-281 ($n = 3$); frequencies and number of GI-infiltrating CD3$^+$CD8$^+$ T cells, frequencies of GI-infiltrating CD3$^+$CD4$^+$ and IL-17-producing CD4$^+$ T cells in the GI tract on days 7 and 14 after the first administration of IgG1 control or IgG-281 ($n = 3$). **M** In vivo schematic of administration of IgG control or IgG-281 or IgG-281+anti-PD-1 in leukemia mice; frequencies of MLL-AF9 leukemic cells in the malignant BM niches on day 21 after the first administration of the antibodies ($n = 4$), and survival analysis of MLL-AF9 leukemic cell-bearing mice treated with the three treatments ($n = 10$ /group). **N** In vivo schematic of administration of IgG1 control or IgG-281 in MLL-AF9 leukemic cell-bearing mice with low burden leukemia (dose of $10^3$ MLL-AF9 leukemic cell); frequencies of MLL-AF9 leukemic cells in the malignant BM niches on day 35 after the first administration of IgG1 control or IgG-281 ($n = 4$/group), ND non detectable; survival analysis by Kaplan–Meier in these low burden leukemia mice treated with IgG1 control or IgG-281 ($n = 10$/group); ablation of ST2$^+$ T$_{reg}$ cells while total T$_{reg}$ cells are preserved; Tbet$^+$ T$_{reg}$ cells and Bcl6$^+$ T$_{reg}$ cells were increased and Blimp1$^+$ T$_{reg}$ cells were decreased, increased frequencies of CD3$^+$CD8$^+$ T cells, proliferating Ki67$^+$CD8 T cells, CD8$^+$Tbet$^+$ T cells, and decrease of CD8$^+$PD-1$^+$ T cells; increase of NK cells in the IgG-281 treated mice versus IgG1 mice ($n = 4$/group). **O** Structural schematic of humanized ST2-neutralizing antibody (IgG-282) and in vivo schematic of administration of IgG1 control or IgG-282 to the humanized NSG leukemic model. **P** Frequencies of human MOLM-14$^{eGFP}$ leukemic cells in the peripheral blood on day, 7, 14 and 21, and in the malignant BM niches on day 21 after the first administration of IgG1 control or anti-human ST2 IgG-282 ($n = 5$); survival analysis of human MOLM-14 leukemic cell-bearing NSG mice treated with IgG1 control ($n = 9$) or IgG-282 ($n = 10$). **Q** In vivo apoptosis of ST2$^+$ T$_{reg}$ cells driving abatement of ST2$^+$ T$_{reg}$ cells and GZMB$^+$ST2$^+$ T$_{reg}$ cells in the malignant BM niches on day 21 after the first administration of IgG-282 compared to IgG1 control ($n = 5$). **R** Counts of human CD3$^+$CD8$^+$ T cells and frequencies of human CD8$^+$Ki67$^+$, CD8$^+$CD107a$^+$, CD8$^+$IFNγ$^+$, and CD8$^+$Tbet$^+$ in the malignant BM niches on day 21 after the first administration of IgG1 control or IgG-282 ($n = 5$). **S** Frequencies of human CD8$^+$PD-1$^+$ T cells in the malignant BM niches on day 21 after the first administration of IgG1 control or IgG-282 ($n = 5$). Data are mean ± s.e.m compared with unpaired two-sided $t$-test (**B, D, F–L, N, P–S**) or ANOVA with post-hoc Bonferroni $t$-test (**M**), and Kaplan–Meier curves compared using log-rank test (**E, M, N, P**).

protecting CD8 T cells from cytolysis by ST2$^+$ T$_{reg}$ cells in both immunocompetent and humanized pre-clinical models.

## Colocalization of ST2$^+$ T$_{reg}$ cells and CD8 T cells is disrupted by engineered anti-ST2 antibody in the splenic and liver leukemic niche of epigenetically mutated immunocompetent myeloid DNMT3A/FLT3$^{ITD}$ mice

To test ST2 neutralization in another immunocompetent clinically relevant model of leukemia, we used the epigenetically mutated DNMT3A/FLT3$^{ITD}$ model[28] that developed frequent extramedullary leukemia in the spleen and liver. Furthermore, decalcification methods of BM biopsy specimens are complex, and imaging of the leukemic spleen could address the colocalization of ST2$^+$ T$_{reg}$ cells and CD8 T cells. Three weeks after transplantation, we started therapy with anti-murine ST2 or control Abs at 25 μg/mice for the first three doses and then 50 μg/mice for the fourth to the tenth dose. Mice were euthanized after receiving ten doses of anti-murine ST2 Ab or control, and spleen and liver tumoral and immune cells were analyzed. As shown in Fig. 10A, B, a significant reduction in spleen and liver sizes and weights is observed on treatment with anti-murine ST2 Ab compared to the control group, indicating a significant decrease in leukemic burden, emphasizing the suppressive ability of anti-murine ST2 Ab in this aggressive epigenetic AML model. Furthermore, a significant increase in spleen CD3$^+$CD8$^+$ and CD3$^+$CD4$^+$ T cells (Fig. 10C) was observed. Frequencies of myeloid-derived suppressor cells/ monocytes (CD11b$^+$Gr1$^{int}$F4/80$^+$) and neutrophils (Gr1$^+$CD11b$^+$) expressing ST2 are also decreased (Fig. 10D).

We next showed in an extramedullary leukemic niche, the spleen, of the DNMT3A/FLT3$^{ITD}$ model using confocal imaging, the colocalization of ST2$^+$ Foxp3$^+$ T$_{reg}$ cells and CD8 T cells in the control-treated mice. For quantification, we have analyzed the distance between ST2$^+$ T$_{reg}$ cells and CD8a$^+$ T cells and observed some notable findings. In the control-Ab-treated group, the average distance between ST2$^+$ T$_{reg}$ cells and CD8a$^+$ T cells is on average 0.208 μm, while in the anti-ST2-Ab-treated group, the average distance is 19.25 μm. These results suggest that in controls, there is a close contact of these two cells populations in the leukemia niche and that anti-ST2 Ab treatment dissociates CD8 T cells from ST2$^+$ T$_{reg}$ cells (Fig. 10E). Furthermore, ST2$^+$ Foxp3$^+$ T$_{reg}$ cells per field almost vanished while a substantial increase of CD8 T cells per field was seen in the spleen of anti-ST2 Ab-treated mice (Fig. 10E). Together, these data suggest that ST2 neutralization dissociate the interaction between ST2$^+$ Foxp3$^+$ T$_{reg}$ cells and CD8 T cells allowing a recovery of CD8 T cells numbers.

## Discussion

AML is a clonal disease characterized by the rapid expansion of immature myeloid cells with impaired differentiation in the BM[56]. Therapies for several blood cancers of lymphoid origin (i.e., ALL, NHL) have made remarkable leaps forward, while AML therapies have barely changed over the past 30 years. Although some advances in AML therapy and higher rates of complete remission have been obtained with molecular targeted therapies, many patients still experience relapse and die from AML, which remains the 6$^{th}$ most common cause of cancer death[57]. In solid tumors, checkpoint inhibitors have transformed patients' outcomes. They work by inhibiting the exhaustion brake, such as PD-1 or CTLA4, on effector T cells[2,3]. Little is known about the role of checkpoint inhibitors in AML, and no specific checkpoint inhibitor has been developed for AML. However, evidence of dysfunctional and exhausted CD8 T cells in AML patients has been reported[4–6]. The results of the present study underscore a critical role for CD8 T cells in antileukemic effects that are perturbed by the presence of intratumoral ST2$^+$ T$_{reg}$ cells in the malignant BM niche. Herein, we demonstrated this mechanism in aggressive myeloid leukemia models. In comparison to those observed in solid tumor models, the frequencies of T$_{reg}$ cells in the BM, even at steady state, are much higher relative to those of other immune cells, and upregulated expression of the IL-33 receptor ST2 is observed. The special role of T$_{reg}$ cells in the maintenance and development of the healthy

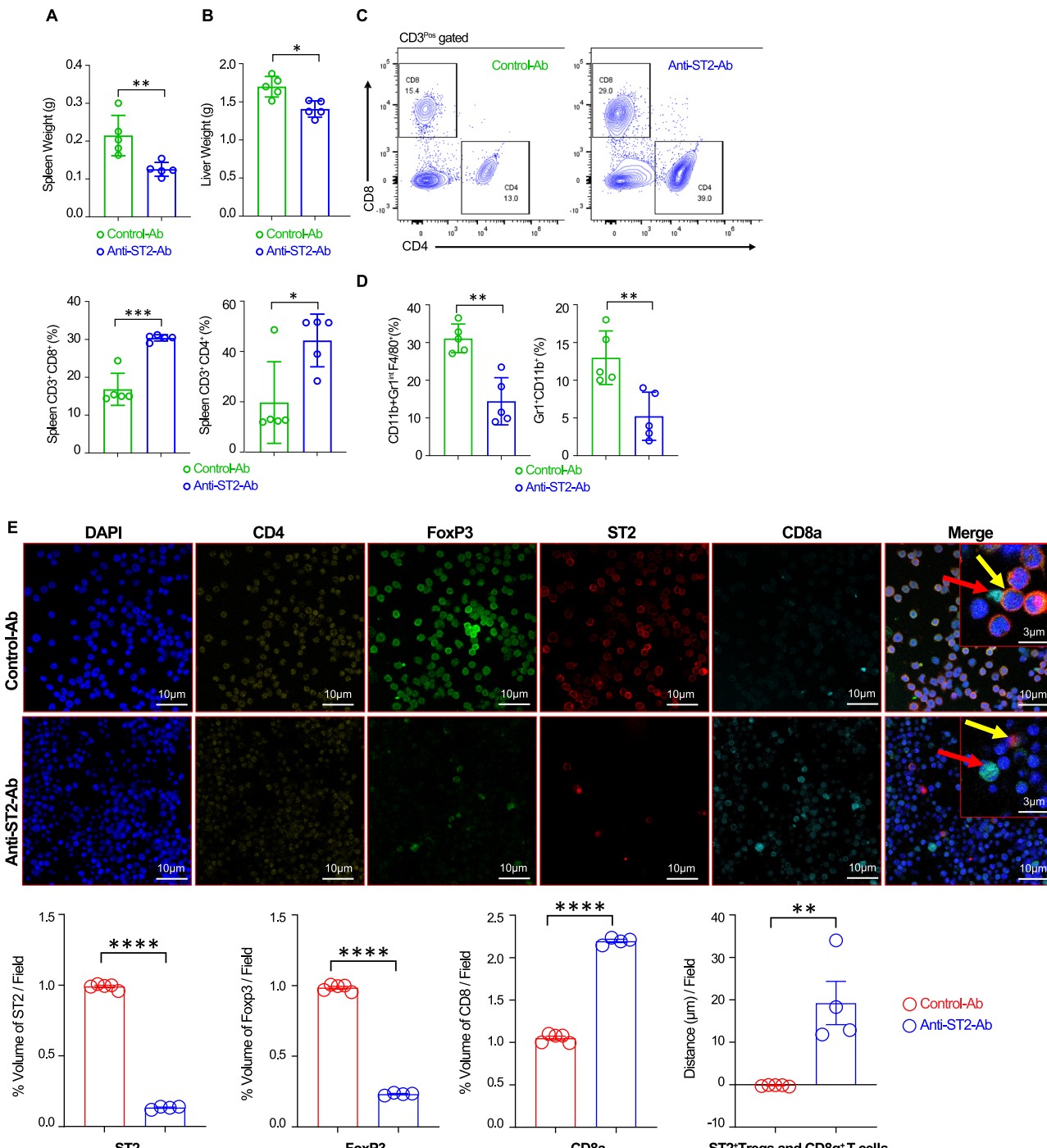

**Fig. 10 | Colocalization of ST2⁺ T_reg cells and CD8 T cells is disrupted by engineered anti-ST2 antibody in the splenic leukemic niche of epigenetically mutated immunocompetent myeloid DNMT3A/FLT3^ITD mice.**
**A**, **B** Representative sizes and weights of spleens and livers from DNMT3A/FLT3^ITD transplanted mice after the 10th administration of control Ab or anti-ST2 Ab ($n = 5$ in each group). **C** Representative plots and frequencies of spleen CD3⁺CD8⁺ T cells and frequencies of spleen CD3⁺CD4⁺ T cells following the 10th administration of control Ab or anti-ST2 Ab ($n = 5$ in each group). **D** Frequencies of spleen myeloid-derived suppressor cells/ monocytes (CD11b⁺Gr1^int F4/80⁺) and spleen neutrophils (Gr1⁺CD11b⁺) following the 10th administration of control Ab or anti-ST2 Ab ($n = 5$ in each group). **E** Representative confocal images showing DAPI (blue), anti-CD4

(yellow), anti-FoxP3 (green), anti-ST2 (red), anti-CD8a (cyan), and merged immunofluorescence staining in the spleens of DNMT3A/FLT3^ITD AML mice treated with either control antibody (Control-Ab) or anti-ST2 antibody (Anti-ST2-Ab). In the spleens of Anti-ST2-Ab-treated AML mice, ST2⁺ T_reg cells (CD4⁺FoxP3⁺ST2⁺) are nearly absent, while the number of CD8 T cells is significantly increased compared to the control-Ab-treated group. The shortest distance between ST2⁺ T_reg cells (insert in merged image, highlighted with a yellow arrow) and CD8 T cells (insert in merged image, highlighted with a red arrow) was quantified using Imaris (Elmo) software. Data are mean ± s.e.m. ($n = 5$ in each experiment). A two-sided unpaired $t$-test was used. *$P < 0.05$; **$P < 0.01$; ***$P < 0.001$, ****$P < 0.0001$. Source data are provided as a Source Data file.

hematopoietic stem-cell niche has been described[58]. ST2[+] T[reg] cells are enriched in tissues, including the BM niche[22,23], and we confirm here that human HDs BM have a high frequency of ST2[+] T[reg] cells. The mode of action of IL-33/ST2 in solid tumors is predominantly driven by type-2 tumor-associated macrophages (TAM), as has been shown in colorectal cancers, squamous cell carcinoma, and ovarian cancers[59–61]. Conversely, in the malignant BM niche, ST2[+] T[reg] cells frequencies are amplified in several AML datasets and murine leukemia models, and these cells represent the predominant subset of regulatory cells, while ST2[+] TAMs are not significantly increased in these AML patients' datasets and murine leukemia models. Cell compositions of TME in solid cancers and leukemia vis-à-vis TAMs are different, considering TAMs are derived from monocytes egressing from the BM and migrating to tumor sites rather than originating from tissue-resident macrophages[62]. Indeed, bulk RNA sequencing analysis showed ST2[+]FOXP3[+]T[reg] cells are increased in AML patients as compared to HDs. It is important to note that in a single-cell RNA sequencing analysis using the 10X genomics platform with a focus on T cells[25], FOXP3[+] T[reg] cells were not identified, probably due to the low number of captured T[reg] cells proportionally to the number of leukemic and myeloid cells and immunohistochemistry was required to detect T[reg] cells. Our approach to identify rare immune populations was to use BM samples after chemotherapy induction with no more than 20% blasts and flow cytometry measurement, which facilitates the quantification of T[reg] cells. Single-cell RNA sequencing showed on the UMAP several subsets of T cells including FOXP3[+] T[reg] cells (Supplementary Fig. 3A) with an overrepresentation of FOXP3[+] T[reg] cells, exhausted CD8 T cells and dysfunctional CD8 T cells in the Ref. group versus CR group (Supplementary Fig. 3B). However, a distinct subset of ST2(IL1RL1)+ T[reg] cells was not seen. However, analysis at the protein level of BM samples from refractory patients versus CRs by flow cytometry revealed that the total numbers of both ST2[+] T[reg] cells and KLRG1[+] activated ST2[+] T[reg] cells were significantly higher in the refractory patients' samples. Primary leukemic models of DNMT3A/FLT3[ITD]-mutant mice[28] confirmed that in advanced leukemic mice, the frequencies of activated ST2[+] T[reg] cells in the malignant BM were much higher than those found in pre-leukemia or intermediate leukemia mice. To dynamically monitor ST2[+] T[reg] cells in malignant BM niches, we developed two aggressive leukemic models: MLL-AF9[eGFP] and DNMT3A/FLT3[ITD], and confirmed in both the increases of KLRG1[+]ST2[+] T[reg] cells in the BM TME with the progression of leukemia. The role of KLRG1 on T[reg] cells remains to be elucidated[63], and our data in AML provide some clues, given that up to 50% of KLGR1[+] T[reg] cells stained positively for ST2, suggesting some overlapping role for KLRG1 and ST2. Further, Batf controls the activation program of T[reg] cells in tumors[64] and was a key transcript correlating with ST2 expression on AML TME T[reg] cells.

ST2 is the unique receptor for IL-33, and the role of ST2 in T[reg] cell function has been explored by several groups in host homeostasis, particularly in the gut, inflammatory diseases, and solid tumors[12–17]. As seen in these conditions, ST2[+] T[reg] cells from the malignant leukemic niches (BM, spleen, liver) upregulate critical inhibitory cytokines and activation markers, leading to an increased suppressive capacity as compared to ST2[−] T[reg] cells. While ST2[+] T[reg] cells inhabit tissues, a precursor population of ST2[+] T[reg] cells expressing the transcription factor nuclear factor Nfil3 exists in lymphoid organs[40]. We have shown that these precursors are increased in the LNs following ST2 blockade in T[reg] cells in leukemia models and decreased in the BM. Furthermore, chemokine receptor analysis showed that ST2[+] T[reg] cells are a subset with high BM homing potential in both healthy and leukemic mice, and CXCR4 expression is preferentially increased in ST2[+] T[reg] cells from the leukemic niche. Previous literature has shown that increased CXCR4 on leukemia cells promotes migration of T[reg] cells toward the leukemic niche, and blocking the CXCR4 axis decreased T[reg] cells activation and leukemia growth[41]. Here, we showed that activated ST2[+] T[reg] cells can directly express CXCR4, favoring their migration to the leukemic BM niche.

The heterogeneity of T[reg] cells can be, in part, attributed to differential expression of transcription factors that shape their trafficking, survival, and functional properties, some of which are niche-specific. Classically, GATA3 binds directly to the ST2 promoter to increase ST2 expression through a positive feedback loop in T[reg] cells[42,43] but in the context of a tumor, IFN-γ in T[reg] cells enhances tumor-killing immune responses[44], and the transcription factor Tbet controls type 1 immune responses in a subset of T[reg] cells[9,18]. Tbet has been shown to bind to the ST2 promoter and regulate ST2 through STAT4 in Th1 cells[46]. However, if Tbet can bind to the ST2 promoter in T[reg] cells and the role of Tbet in T[reg] cells in the framework of the tumor has not been studied. We hypothesized that, similar to IFN-γ in intratumoral T[reg] cells, Tbet expression in T[reg] cells would better kill the leukemia and be negatively correlated with ST2[+] T[reg] cells frequencies in the leukemic niche. Using bulk RNA sequencing, we found, as hypothesized, that Tbet deficiency in T[reg] cells in leukemic mice resulted in a significantly increased tumor burden and shortened survival at the opposite spectrum of ST2 deficiency in T[reg] cells. To test the possible direct interaction between ST2 and Tbet transcription factor, we used ChIP-qPCR, which did not show an association, which does not preclude binding up or downstream promoter regions, which may be explored in future studies. Interestingly, a recent study discovered a new alternative type 1 immunity-restricted promoter of ST2 located far upstream of the classical ST2-coding region, and it drove anti-viral Th1 cells and CTLs function[65]. Since Bcl6 controls Th2 activity of T[reg] cells by inhibiting Gata3[47], and represses ST2 expression in T[reg] cells in a cell-intrinsic manner[48], we explored this lineage-defining transcription factor and associated accessory Blimp1 transcription factor in the liquid tumor context. We found, in the leukemic niche, that Bcl6 percentage is decreased in ST2[+] T[reg] cells while the percentage of Blimp1-expressing T[reg] cells is increased in ST2[+] T[reg] cells and correlated with GZMB cytolytic molecule. Furthermore, anti-ST2 Ab treatment reverted the Bcl6/Blimp1 balance of these signaling pathways in ST2[+] T[reg] cells. Together, ST2 expression in T[reg] cells may be regulated by multiple lineage-defining and accessory transcription factors such as Gata3, Tbet, Bcl6, and Blimp1. In leukemia, where T[reg] cells display a Th2-like phenotype including expression of ST2, Tbet, and Bcl6 are repressed, and Blimp1 is increased, whereas ST2 neutralization inversely affects the Tbet, Bcl6, and Blimp1 transcription factors program.

While the major role of T[reg] cells and ST2[+] T[reg] cells is their direct immunosuppression of effector CD4[+] and CD8 T cells[13,22,23,29], it has been shown that T[reg] cells and recently engineered Chimeric Antigen Receptor (CAR) T[reg] cells can clear tumors through a contact-dependent granzyme- and perforin-mediated process[49,66]. Here, we showed that ST2[+] T[reg] cells that are non-engineered and non-professional cytotoxic T cells are equipped with high levels of GZMB (up to 40%) and can directly kill leukemia-primed effector T cells at a low ratio of 1:1 to promote tumor progression, adding a second mechanism of suppression. Indeed, we observed the direct killing of surrounding CD8 T cells (from BM or spleen) by TME ST2[+] T[reg] cells by imaging studies. IL-33 stimulation enhanced this ST2[+] T[reg] cells killing, whereas adding a GZMB-specific inhibitor, Z-AAD-CMK, abolished the cytolytic effect. Though the exact mechanism still needs to be elucidated, the present study indicated that leukemia-derived ST2[+] T[reg] cells from BM or spleen preferentially kill leukemia-primed (TME) CD8 T cells and not CD45RA-sorted naïve or non-TME CD8 T cells. Since TME CD8 T cells are AML antigen-specific (WT1 as a representative antigen), it is important to preserve them in the BM niche. Also important, ST2 deficiency in T[regs] or treatment with neutralizing anti-ST2 Abs early during leukemia restores WT1[+]CD8[+] T cells in the TME. The hypothesis that treatment when only a lower burden of tumor is present is attractive because the TME ratio of CD8 T cells to ST2[+] T[reg]

cells, and to MLL-AF9 cells, will be favorable for antigen-specific WT1⁺CD8⁺ T cells to be able to kill the leukemia. Indeed, an adoptive transfer with only $10^3$ leukemic cells and treatment with anti-ST2 IgG-281 resulted in a complete salvage of the leukemic mice.

How ST2⁺ T$_{reg}$ cells recognize tumor-infiltrating effector T lymphocytes in the TME to kill them remains incompletely understood. Since TME CD4 effector T cells could also be directly killed through the GZMB-mediated process by ST2⁺ T$_{reg}$ cells, but modest depletion of NK cells was achieved, probably through an indirect mechanism, and leukemic cells, other myeloid cells, and B cells could not be killed or suppressed by ST2⁺ T$_{reg}$ cells, it suggests a MHC-TCR recognition between ST2⁺ T$_{reg}$ cells and effector T cells. In vitro direct killing by ST2⁺ T$_{reg}$ cells of TME CD8 T cells could be mediated through trogocytosis, as both cells' subtypes are primed using coculture with AML cells[67]. Deciphering the TCR-antigen recognition mechanism through TCR sequencing is beyond the scope of this study.

Increasing evidence indicates that targeting T$_{reg}$ cells could improve the function of effector lymphocytes in the TME of solid tumors, thereby strengthening antitumoral immune responses[68,69]. Use of an anti-ST2-neutralizing antibody has shown therapeutic efficacy in graft-versus-host disease and maintained the GVL response[29]. Here, we used Fc-silenced engineered antibodies towards murine and human ST2 with Fc silencing to eliminate non-specific Fc binding[53]. After verifying these antibodies depleted ST2⁺ T$_{reg}$ cells by inducing apoptosis and inhibiting proliferation, we showed in multiple models of leukemia that increased antitumoral activity occurred with a decrease in CD8⁺ T cell exhaustion and restoration of type 1 immunity, extending survival. Upon conditional knockout of ST2 in T$_{reg}$ cells, we further demonstrated that antigen-specific WT1⁺CD8⁺ T cells were significantly increased, particularly those of the effector memory phenotype. Because ST2⁺ T$_{reg}$ cells are tissue-specific, these data reveal a new way to deplete T$_{reg}$ cells within the local TME without impacting the total and circulating T$_{reg}$ cells, which is a potentially safer approach. To this point, we verified the absence of toxicity, specifically the absence of colitis, by demonstrating that ST2⁺ T$_{reg}$ cells are present in the healthy gut and no changes in mouse body weight, colon length, and long-term T$_{reg}$ cell frequencies occurred with anti-ST2 antibody treatment. These results suggest that targeting ST2⁺ T$_{reg}$ cells is an efficacious and safe immunotherapeutic treatment for AML.

ST2 may be expressed by other immune cells in the leukemic niche. Therefore, we addressed the role and weight of ST2⁺ T$_{reg}$ cells versus non-T$_{reg}$ immune ST2⁺ cells in AML development with several experiments. First, an adoptive transfer in a lethally irradiated mice that contained only AML cells, supportive BM cells, T cells and T$_{reg}$ cells and does not contain other immune cells or mature myeloid cells showed that mice transplanted with ST2⁻/⁻Foxp3$^{eGFP+}$ T$_{reg}$ cells versus WT Foxp3$^{eGFP+}$ T$_{reg}$ cells had an extended survival, less AML growth, and an increase in CD8⁺Tbet⁺ and CD8⁺IFNγ⁺ T cells and less CD8⁺ PD-1⁺ exhausted T cells. (Fig. 4A–F). Second, to further establish the weight of leukemic ST2⁺ T$_{reg}$ cells in the overall role of ST2 in the BM niche, we used a non-irradiated leukemic model and injected MLL-AF9 leukemia cells into WT or ST2⁻/⁻ mice and additionally depleted T$_{reg}$ cells with anti-CD25 antibody. Approximately half of the survival increase was due to T$_{reg}$ cells (anti-CD25 in WT mice) and half to other non-T$_{reg}$ ST2-expressing cells (isotype in ST2⁻/⁻ mice) (Fig. 4G). Third, to confirm the specific and unique role of ST2 in leukemic T$_{reg}$ cells, we generated *Foxp3$^{Cre}$ST2$^{fl/fl}$* and used these mice with the non-irradiated MLL-AF9 leukemic model (Fig. 5A). In this model where ST2 is deleted in all T$_{reg}$ cells, we observed that compared with *Foxp3$^{Cre}$* leukemic controls, *Foxp3$^{Cre}$ST2$^{fl/fl}$* leukemic mice had longer survival and a lower leukemia burden (Fig. 5B). In this model, the frequencies of either myeloid cells or macrophages, granulocytes, monocytes, and B cells did not differ between groups. Therefore, the sole deficiency of ST2 in T$_{reg}$ cells is sufficient to inhibit MLL-AF9 AML growth and prolong survival. Furthermore, we

used the epigenetic *Dnmt3a$^{fl/+}$ Cre+/Flt3$^{ITD/+}$* AML model. In the context of this *Dnmt3a$^{fl/+}$ Cre+/Flt3$^{ITD/+}$* AML model, monocytes and neutrophils are well-recognized for their roles in AML progression. In this model, the treatment with anti-murine ST2 Ab led to a significant reduction in the tumor burden in the spleen and liver (Fig. 10A, B). These findings reveal that non-T$_{reg}$ ST2⁺ cells contribute to the *Dnmt3a$^{fl/+}$ Cre+/Flt3$^{ITD/+}$* AML.

Additionally, nonhematopoietic stromal cells in the BM may engage ST2 to support hematopoiesis[70]. In our AML models (MLL-AF9 and DNMT3A/FLT3$^{ITD}$), as was seen in myeloproliferative neoplasm models, the main source of IL-33 is stromal cells. As aforementioned, we have shown that half of the ST2 effect in AML is driven by T$_{reg}$ cells (Fig. 4G). Furthermore, all the outcomes and phenotypes seen in primary leukemia are reversed in the *Foxp3$^{Cre}$ST2$^{fl/fl}$* mice (Fig. 5A). Moreover, AML hematopoietic stem and progenitor cells have been shown to upregulate ST2[71,72]. Therefore, these three types of BM niche cell subsets that all accelerate leukemia growth may represent additional targets for ST2 neutralization.

In summary, our results identified a new function for the ST2/IL-33 pathway as a key linker between inflammation-driven leukemic remodeling and the local BM T$_{reg}$ cell response. Indeed, our data indicated that BM-derived leukemic T$_{reg}$ cells respond to IL-33 upon tumor growth through increased expression of ST2 and that signaling has a crucial role in their augmented cytolytic function. Increased intratumoral ST2⁺ T$_{reg}$ cell numbers correlated with an unfavorable outcome in AML as well as CD8⁺ T cell exhaustion and depletion. Remarkably, the intrinsic Tbet transcription factor limits this amplifying pro-tumoral response via inhibition of ST2⁺ T$_{reg}$ cell cytotoxicity towards effector lymphocytes and preservation of ST2⁺ T$_{reg}$ cell precursor status. Mechanistically, ST2⁺ T$_{reg}$ cells kill, through a GZMB-mediated cytotoxicity, CD8 T cells in the TME. This suggests that the IL-33/ST2 axis represents an attractive potential immune checkpoint target for AML, and blockade of ST2⁺ T$_{reg}$ cells either genetically or with engineered neutralizing antibodies showed promising inhibition of AML growth to improve survival in pre-clinical murine and humanized models. The ability of intratumoral ST2⁺ T$_{reg}$ cells to kill exhausted CD8 T cells in response to leukemia progression may represent a broader mechanism for cancer treatment. We harnessed this information to develop a new checkpoint therapy that significantly improved survival even as a single agent with no toxicity. These results strongly suggest that ST2 neutralization would benefit patients with aggressive leukemia.

## Methods

### Human research participants
AML patients' BM samples and information were collected after obtaining consent in accordance with institutional review board-approved studies at Fred Hutchinson Cancer Research Center (FHCRC), and the demographic data are presented in Supplementary Table 1. Healthy human BM frozen cells (Cat: 2S-101D) were purchased from Lonza.

### Immune cell subtype deconvolution of AML samples
RNA-seq datasets of 151 TCGA-AML and 187 TARGET-AML samples were downloaded from TCGA. The Beat-AML dataset was downloaded from Vizome (http://vizome.org/aml/). Microarray dataset GSE6891 ($n = 493$)[73] was downloaded from the Gene Expression Omnibus (GEO) database. AML patients were divided into high- and low-*IL1RL1* (ST2-coding gene) groups based on the median value of ST2 expression. Cell-type Identification By Estimating Relative Subsets Of RNA Transcripts (CIBERSORT)[74] was used to quantify the fraction of 7 immune cell types including CD4⁺, CD8⁺, γδ, regulatory T cells, NK, Macrophage, and B cells, which is an analytical tool to provide the deconvolution of the abundances of human distinct immune cell types in a mixed cell population using microarray and sequencing expression

matrices through linear support vector regression. The fractions of these immune cell subtypes in ST2 high- and low-ST2 groups were compared using the Wilcox test.

### ST2$^+$T$_{reg}$ cells enrichment analysis in human AML datasets and healthy donors

TCGA/TARGET-AML and normal GTEX RNA sequencing datasets[75] were downloaded from TCGA. Microarray datasets including GSE6891 ($n = 493$), AML_CG1999 trial (GSE12417 ($n = 163$) and GSE37642 ($n = 422$))[76] and normal HDs GSE9476 ($n = 10$)[77] from the GEO database were also downloaded. To calculate the enrichment levels of ST2$^+$T$_{regs}$ on AML and HD, TCGA/TARGET-AML and normal GTEX datasets were merged, and batch effects were removed using the "sva" R package, as well as the datasets and normal datasets from the GEO datasets. Then, we quantified the enrichment levels of the ST2$^+$T$_{reg}$ cells using *IL1RL1* and FOXP3 as the signatures in these RNA sequencing and microarray datasets by the single-sample gene-set enrichment analysis (ssGSEA)[78,79]. The enrichment levels (ssGSEA scores) of the ST2$^+$T$_{reg}$ cells in AML versus HD were compared using a two-sided unpaired *t*-test or ANOVA.

### Cell lines and reagents

The murine malignant leukemic cell line MLL-AF9$^{eGFP}$ (mixed-lineage leukemia and AF9 fusion protein) was initially generated by former members of the Paczesny Lab and maintained in C57BL/6 mice[29]. These cells were sorted by gating the enhanced green fluorescence protein (eGFP)-positive cells from the malignant BM niche on a BD FACS Aria (BD Bioscience). Murine *FLT3$^{ITD}$/DNMT3A*-mutant leukemic cells were isolated from the malignant BM niches of *Flt3$^{ITD/ITD}$[28]; Dnmt3a$^{fl/fl}$ Mx-Cre* mice as previously reported[28] and provided by Dr. Reuben Kapur. The human leukemic cell line MOLM-14 was acquired from the ATCC maintained in RPMI media supplemented with 10% fetal calf serum (Hyclone, Cat: SH30073.03), 100 U/ml penicillin, and 100 mg/ml streptomycin (Gibco, Cat:15140122), then reconstituted with a luc2-gfp-luciferase lentiviral construct and sorted by flow cytometry to gfp$^+$ >99% purity. ST2 antibodies were dissolved with endotoxin-free PBS (Millipore, Cat: 6C1766) and filtered through 0.22-μm filters (Thermo Scientific, Cat: 09-720-03). Prior to use in animal experiments, a LAL chromogenic endotoxin quantification kit (Thermo Scientific, Cat: 88282) was used to validate the endotoxin levels in the cell cultures.

### Mouse strains

C57BL/6 (B6, H-2b, CD45.2$^+$, Stock: 000664) mice or C57BL/6 BoyJ (CD45.1$^+$, Stock: 002014) (6–7 weeks old, female) were purchased from The Jackson Laboratory. Two pairs of NOD.Cg-Prkdc$^{scid}$ Il2rg$^{tm1Wjl}$/SzJ (NSG, Stock:005557) mice were initially purchased from The Jackson Laboratory and maintained and bred by the Division of Laboratory Animal Resources of the Medical University of South Carolina. C57BL/6 background *Foxp3$^{eGFP}$* reporter mice were provided by Matthew J. Turner (Indiana University)[80]; C57BL/6 ST*2$^{-/-}$* (CD45.2$^+$) mice were initially provided by Andrew McKenzie (University of Cambridge, Cambridge, United Kingdom)[81] and then were bred to C57BL/6 Foxp*3$^{eGFP}$* mice. The homozygous *ST2$^{-/-}$Foxp3$^{GFP}$* genotype was confirmed by PCR sequencing. C57BL/6 ST2 conditional knockout mice were generated by crossing C57BL/6 Foxp3$^{YFP/Cre}$ mice (Foxp3$^{Cre}$, Jackson Laboratory, Stock: 016959)[36] with C57BL/6 ST2$^{fl/fl}$ mice (generated by us and Cyagen Biosciences, Santa Clara, CA, USA. Quote: TKC-190220-AHL-01-tac), and conditional knockout mice were further verified by genotyping (TransnetYX, Cordova, TN, USA). Prior to each experiment, verified by flow cytometry that ST2 was specifically deleted in T$_{reg}$ cells was performed in *Foxp3$^{Cre}$ST2$^{fl/fl}$* mice (Supplementary Fig. 17). All mice mentioned above were housed and bred in specific-pathogen-free facilities, under a 12-h reverse light-dark cycle condition with ad libitum access to water and food. Both experimental and control mice were co-housed and bred together before separating. Mice are euthanized by carbon dioxide (CO$_2$) inhalation using the euthanasia chamber, and double verified the death by ascertaining cardiac and respiratory arrest. We define the criteria for humane endpoint needing euthanasia of the leukemic mice based on weight loss >20% of original body weight and/or percentage of leukemic cells (CD3$^-$eGFP$^+$) in the peripheral blood >10%, which in our MLL-AF9$^{Egfp}$ leukemic model represents a tumor burden of 30–50% of leukemic cells in the BM and/or other signs of leukemic infiltration such as uncontrolled pain, increased facial or abdominal swelling, and limb paralysis. Both male and female animals were included in the animal models used in this study. However, for certain experiments, only female mice were used to avoid aggressive behaviors observed in male mice, such as fighting, that may cause wounds and potentially affect our scoring and thus experimental outcomes. Animal protocols were, respectively, reviewed and approved by the Institutional Animal Care and Use Committee (IACUC, No.:18033) at Indiana University School of Medicine and the IACUC (No.:20-0977) at Medical University of South Carolina. All animal experiments were strictly carried out following the guidelines for experimental pain in conscious animals to minimize their suffering and improve their welfare.

### Flow cytometry

In the current study, unless otherwise stated, all antibodies and reagents used for flow cytometry were purchased from eBioscience (San Diego, CA), BD Bioscience (San Jose, CA) or Biolegend (San Diego, CA), except mouse ST2-PE (Clone DJ8; Cat: 101001PE) and human ST2-FITC (Clone B4E6; Cat: 101002F) antibodies, which were from MD Biosciences Bioproducts (Oakdale, MN) (Supplementary Table 4). For staining of multiple surface and intracellular markers, to prevent non-specific binding of the antibodies, samples of single-cell suspensions were preincubated with anti-mouse CD16/CD32 monoclonal antibody for 10–20 min at 4 °C. Surface marker staining was carried out at 4 °C for 30 min. Then, the fixable viability dye (FVD, eBioscience; Cat: 65-0866-14) was added at 1:1000 to distinguish live cells from dead cells. The Foxp3/Transcription Factor Staining Buffer Set (eBioscience; Cat: 00-5523-00) and the Fixation/Permeabilization Kit (BD Biosciences; Cat: 555028) were used in combination to stain for intracellular transcription factors and cytokines. For cytokine staining, single-cell samples were generally stimulated with PMA (50 ng/ml; Sigma-Aldrich; Cat: 16561-29-8), ionomycin (1 μg/ml; Sigma-Aldrich; Cat: 56092-82-1), and GolgiStop monensin (1 μg/ml; eBioscience; Cat: 00-4505-51) for 5–6 h at 37 °C.

For Wilms' tumor 1 (WT1) staining on C57BL/6 MLL-AF9$^{eGFP}$ leukemia cells, MLL-AF9$^{eGFP}$ leukemia cells were sorted twice, and eGFP$^+$ >99% purity on BD Aria sorter was obtained. Surface marker CD45 (clone 2D1) staining was described as above. Then, anti-mouse WT1 polyclonal antibody conjugated Alexa Fluor 594 (Bioss, Cat: bs-6983R-A594) and IgG control were used at a 1:100 dilution following fixation by the Foxp3/Transcription Factor Staining Buffer Set and the Fixation/Permeabilization Kit.

Both the surface staining samples and the intracellular staining samples were analyzed via flow cytometry using BD LSRII and BD LSR Fortessa (X-20) or ThermoFisher Attune NxT systems. All flow cytometric analyses were performed using FlowJo 10.7.0 software (TreeStar). The gating strategies used to analyze BM samples of AML patients and leukemic mice are presented in Supplementary Fig. 4. To calculate the absolute numbers of a cell population, we first count the total number of viable cells from the tissues of interest, for example for BM, all harvested bones (tibias and femurs) or for a spleen for each mouse. Then, the absolute number of a specific population (i.e., KLRG1$^+$ST2$^+$T$_{regs}$) was calculated by multiplying the total number of

viable cells per tissue (BM or spleen) by the population frequency (%) of parent cells. Each absolute number represents one mouse.

## T_reg cell sorting and effector lymphocyte isolation

T_reg cells used for Nanostring analysis and bulk mRNA sequencing were sorted from single-cell suspensions of normal BM or malignant BM niches by gating of CD4⁺CD25⁺ cells with CD127⁻ (for Tbet⁻/⁻ T_reg cell isolation) or eGFP⁺ (for WT T_reg cell and ST2⁻/⁻ T_reg cell isolation). The purity of sorted T_reg cells was required to be at least 95% before the next step. Cell sorting was performed via flow cytometry with a BD FACS Aria (BD Bioscience). The T_reg cells used for adoptive transfer or in vitro and ex vivo studies were isolated and purified from single-cell suspensions of normal spleen and BM or malignant BM niches using the immunomagnetic beads according to the manufacturer's protocol (Miltenyi Biotec; Cat: 130-091-041). For purification of ST2⁺ T_reg cells and ST2⁻ T_reg cells, T_reg cells were isolated using both anti-mouse or anti-human CD25-APC and the Anti-APC MultiSort Kit (Miltenyi Biotec; Cat: 130-091-255). ST2⁺ T_reg cells were further purified using anti-mouse ST2-PE or anti-human ST2-FITC and anti-PE microbeads (Miltenyi Biotec; Cat: 130-048-801) or anti-FITC microbeads (Miltenyi Biotec; Cat: 130-048-701).

Normal CD8 T cells were isolated from single-cell suspensions of normal spleen and BM using the indicated immunomagnetic beads (Miltenyi Biotec; Cat: 130-104-075 or Cat: 130-116-478) according to the manufacturer's protocol. Tumor-infiltrating (TIL) CD8 T cells were specifically purified via positive selection (Miltenyi Biotec; Cat: 130-116-478) in an LS magnetic cell-sorting column (Miltenyi Biotec; Cat: 130-042-401). For use in in vitro and ex vivo cell coculture experiments, selected cell fractions containing target cells were separated over a second column to increase the purity of isolated T_reg cells and effector lymphocytes. To obtain T cell-depleted BM cells (TCD-BM), an immunomagnetic bead-based T cell depletion kit (Miltenyi Biotec; Cat: 130-049-101) was used to deplete CD90.2⁺ T cells in the single-cell suspensions of donor BM. To isolate conventional T cells from the single-cell suspensions of donor spleen, an immunomagnetic bead-based negative selection kit (Miltenyi Biotec; Cat: 130-095-130) was employed to pool purified total T cells, and CD25⁺ T cells were then depleted from the total T cells (Miltenyi Biotec; Cat: 130-091-072). The remaining T cells were used as conventional T cells.

Human peripheral blood mononuclear cells (PBMCs) were enriched from HD peripheral blood via density gradient centrifugation by using Ficoll-Paque PREMIUM (GE Healthcare; Cat:17-5442-02). CD8 T cells and T_reg cells were further purified via positive selection using immunomagnetic beads (Miltenyi Biotec, Cat: 130-045-201) and (Miltenyi Biotec; Cat: 130-091-301) according to the manufacturer's protocol and passed through the LS magnetic cell-sorting column.

## Immunosuppression assays

The ex vivo inhibitory capacity of cultured ST2-positive or ST2-negative T_reg cells was assessed with a CFSE inhibition assay as described before[82]. Briefly, tumor effector T cells (CD8a⁺ T cells) labeled with CFSE (Invitrogen; Cat: C34554) after purifying from the leukemic BM single cells or Splenocytes, and cocultured with non-malignant or malignant BM or Spleen-derived ST2⁺ T_reg cells at indicated ratios in the presence of anti-CD3 antibody (3 µg/ml) (Clone 145-2C11; Biolegend; Cat: 100301) and anti-CD28 antibody (5 µg/ml) (Clone 37.51; Biolegend; Cat: 102101). After 4 or 5 days of coculture, proliferation of CD8a⁺ T cells was measured by flow cytometry, data were analyzed with FlowJo, and suppression was determined by the cell division.

## Killing assays of leukemic cells by CD8 T cells

CD8a-positive T cells were sorted from normal or non-malignant or malignant BM niches, or sometimes from the same mice's spleens, and cocultured with background-matched MLL-AF9^eGFP leukemic cells at indicated ratios for 12–18 h in 96 round-bottom wells. Killing assays were then performed as follows: cocultured cells were taken out from the incubator and washed with PBS twice, and stained with viability dye SYTOX-Blue (Invitrogen; Cat: S34857) at 1:1000 for 30 min before flow analysis[83]. Data was analyzed by Flowjo, and lysed cells were defined as eGFP and SYTOX-Blue double-positive ones.

## Cytotoxicity of T_reg cells

The indicated T_reg cells were pre-stimulated with or without recombinant mouse IL-33 (R&D Systems; Cat: 3626-ML) at 20 ng/ml or GZMB inhibitor Z-AAD-CMK (Millipore-Sigma; Cat: 368050) at 50 ng/ml for 15 h before plating in round-bottom, 96-well tissue culture plates ($2 \times 10^5$/well). CD8 T cells were added to the wells at the indicated ratios relative to T_reg cells. After thorough mixing, the plate was centrifuged for 1 min at 2000 rpm to force interaction of the cells, which were then cocultured at 37 °C for 6 h with or without transwell before analysis of CD107a and GZMB expression in T_reg cells and SYTOX-Blue positivity among cells in the coculture wells via flow cytometry.

In vivo T_reg cell cytotoxicity. Both TME CD8 T cells and non-TME CD8 T cells were labeled with CFSE (Invitrogen; Cat: S34554), for TME CD8 T cells labeled with a higher dose of CFSE (5 µM, CFSE^high) and for non-TME CD8 T cells labeled with a lower dose (2 µM, CFSE^low). They were then transferred in lethally irradiated recipient B6 female mice (H2^b, CD45.2⁺) that were inoculated with MLL-AF9^eGFP leukemic cells 72 h before adoptive transfer T_reg cells and CD8 T cells at a ratio of 1:1. Transferred CD8 T cells present in the BM of recipient mice were analyzed by flow cytometry at times 0 and 24 h post transfer of CD8 T cells and T_reg cells. Cells were gated on CD45.1⁺ before further analysis. Lysis of CFSE^high-labeled cells was calculated as: Lysis Percent = (Percent of non-TME CFSE^high − Percent of TME CFSE^low)/Percent of non-TME CFSE^high × 100%.

## Imaging flow cytometry

To directly monitor the cytotoxicity of T_reg cells to tumor effector lymphocytes, imaging flow cytometric analysis was performed on an Amnis ImageStream Mk II Imaging Flow Cytometer (Luminex, USA) according to the manufacturer's instructions. Briefly, mouse TME-derived ST2⁺ T_reg cells, ST2⁻ T_reg cells, and effector lymphocytes were separately isolated from malignant BM niches and purified using immunomagnetic beads. T_reg cells and effector lymphocytes were cocultured at the indicated ratios in round-bottom, 96-well tissue culture plates for 6 h before staining with the indicated labeled antibodies and SYTOX-Blue. The frequencies of T_reg cells were calculated based on the detection of eGFP-positive cells. T_reg cell cytotoxicity was calculated based on the number of SYTOX-Blue−positive effector lymphocytes that colocalized with eGFP-positive T_reg cells, and the data were analyzed using IDEAS Software (Luminex, USA).

## Confocal microscopy for colocalization of ST2⁺ T_reg cells and CD8 T cells in AML mice

To investigate the colocalization of ST2⁺ T_reg cells and CD8 T cells in the spleens of Dnmt3a⁺/⁻; Flt3^ITD/+ AML mice treated with either control antibody (Control-Ab) or anti-ST2 antibody (Anti-ST2-Ab), we performed confocal microscopy as follows: Sample Preparation: Spleen cells were isolated from AML mice treated with either Control-Ab or Anti-ST2-Ab. These cells were then seeded into ibidi µ-slide 8-well™ chambers (Ibidi; Cat: 50-305-795) for subsequent imaging. To preserve cellular structure and protein localization, cells were fixed with 4% paraformaldehyde (Thermo Scientific, Cat: J61899.AK) at room temperature for 15 min. Following fixation, cells were blocked using 5% UltraCruz blocking reagent (Santa Cruz Biotechnology; Cat: sc-516214) for 30 min at room temperature to minimize non-specific binding. Antibody Incubation: After blocking, the cells were incubated overnight at 4 °C with the following, fluorescent-conjugated antibodies to identify the target populations: ST2 (PE), Foxp3 (FITC), CD8a (Alexa

Fluor 647), and CD4 (Brilliant Violet 750). Washing and Staining: Following antibody incubation, cells were washed three times with phosphate-buffered saline (PBS) for 5 min each to remove unbound antibodies. The nuclear stain DAPI (4′,6-diamidino-2-phenylindole, Invitrogen, Cat: D1306) was applied for 1 min to visualize cell nuclei, and the slides were subsequently washed once more with PBS. Confocal Imaging: Fluorescent images of the stained cells were acquired using a Leica SP8 resonant-scanning confocal/multiphoton microscope system located at the Indiana University Microscope Core Facility. Imaging was performed at multiple z-planes to capture high-resolution images of the cellular interactions. A minimum of 4–5 different fields per condition were imaged and analyzed to ensure robust data collection and minimize bias. Colocalization Analysis: The colocalization of ST2$^+$ $T_{reg}$ cells and CD8 T cells was quantitatively assessed using Imaris (Elmo) software. The software was used to measure the shortest distance between ST2$^+$ $T_{reg}$ cells and CD8 T cells within the imaged fields. This analysis provided insight into the spatial relationship between these two immune cell populations in response to antibody treatment.

### Nanostring and bulk RNA sequencing

$T_{reg}$ cells were sorted from the indicated tissues by flow cytometry with a BD FACS Aria. A minimum of three replicates per group were analyzed. RNA was extracted using the PureLink RNA Micro Kit (Invitrogen; Cat: 12183-016). The NanoString method was applied[84], and the nCounter Mouse Immunology kit, which includes 561 immunology-related genes, was used (nanoString; Cat: XT-CSO-MIM1-12). The data analysis was performed with the nCounter Analysis System at NanoString Technologies.

Bulk RNA sequencing was performed in the Core of Molecular Genomics of Indiana University School of Medicine. A minimum of three replicates per group were analyzed. Briefly, RNA quality was evaluated using an Agilent 2100 Bioanalyzer. The resulting library was prepared using the KAPA Hyperprep Kit (Roche; Cat: 07962363001) and QIAseq FastSelect rRNA Remove Kit (QIAGEN; Cat: 333390) in combination. The library was sequenced using a custom program for 28-bp plus 91-bp paired-end sequencing on an Illumina NovaSeq 6000. The sequence reads were mapped to the designated reference genome using Spliced Transcripts Alignment to a Reference (STAR)[85]. To evaluate the quality of the RNA sequencing data, the number of reads that fell into different annotated regions (exonic, intronic, splicing junction, intergenic, promoter, UTR, etc.) of the reference genes was determined with bamUtils[86]. The edgeR package[87] was employed to identify differentially expressed genes between WT no tumor and WT tumor, WT tumor and ST2 knockout tumor, WT tumor and Tbet knockout tumor, and ST2 knockout tumor and Tbet knockout tumor samples. P values were adjusted for multiple comparisons by limiting the false discovery rate (FDR) via the Benjamini-Hochberg method. Only if the log$_2$ fold change was >1 or ≤1 and the adjusted $p$ value was <0.05 was a gene considered differentially expressed. Pathway enrichment analysis was performed using clusterProfiler[88]. For pathway enrichment analysis, genes with a log$_2$ fold change >1 and adjusted $p$ value < 0.01 were used as input.

### Single-cell RNA sequencing of lymphoid subsets in the AML bone marrow microenvironment following chemotherapy induction

Using scRNA-seq, BM aspirates, collected post-induction, from 12 CRs and 10 refractory patients, were compared for lymphoid cells subpopulations. AML patients' BM samples and information were collected after obtaining consent in accordance with institutional review board-approved studies at Fred Hutchinson Cancer Center. One vial of BM from each of the 22 patients was thawed with the standard protocol. Then, cells were counted, and viability was evaluated with Trypan blue staining. 50,000 cells were directly sent to the Center for

Medical Genomics of the Indiana University School of Medicine to be processed on the 10x Genomics, while the leftover cells were stained for multicolor flow cytometry. Extract protocol: After evaluating for cell number, cell viability, and cell size, the appropriate number of cells was loaded on a multiple-channel microfluidics chip of the Chromium Single-Cell Instrument (10x Genomics, Pleasanton, CA, USA) with a targeted cell recovery of 10,000. Single-cell gel beads in emulsion containing barcoded oligonucleotides and reverse transcriptase reagents were generated with the v2. Next GEM Single Cell 3′ reagent kit (10X Genomics, Pleasanton, CA, USA). Following cell capture and cell lysis, cDNA was synthesized and amplified. Library construction protocol: An Illumina sequencing library was then prepared with the amplified cDNA with the Chromium Next GEM Single-Cell 3′ GEM, Library & Gel Bead Kit v2 (10X Genomics, Pleasanton, CA, USA). The resulting library was sequenced, including cell barcode and unique molecular identifier (UMI) sequences, and the read lengths were 26 bp read 1 (cell barcode and UMI) and 91 bp read 2 (RNA read) were generated with Illumina NovaSeq 6000 (Illumina, San Diego, CA, USA) at the Center for Medical Genomics of the Indiana University School of Medicine. ScRNA-seq data processing and visualization: Raw base call (BCL) files were analyzed using CellRanger (v7.0.0)[89]. The "mkfastq" command was used to generate FASTQ files, and the "count" command was used to generate raw gene–cell expression matrices. Ambient RNA contamination was inferred and removed using CellBender (v0.2.0) with standard parameters. Human Genome hg38 was used for the alignment, and gencode.v42 was used for gene annotation and coordinates[90]. Samples from 22 patients were combined in R using the Read10X function from the Seurat package (v4.3.0)[91], and an integrated Seurat object was generated. Filtering was conducted by retaining cells that had UMIs less than 20,000 and had mitochondrial content less than 20 percent. Cell cycle analysis was conducted using the CellCycleScoring with a list of cell cycle markers[92]. Doublets were removed using scDblFinder (v1.12.0)[93]. The SCTransform workflow was used for count normalization, initial integration, and to identify highly variable genes[94] using 30 principal components and resolution of 0.4 for Louvain clustering and UMAP. Batch correction was performed using Harmony (v0.1.1)[95]. Cluster marker genes were identified using FindAllMarkers using the Wilcoxon Rank Sum test with the standard parameters. Cell-type annotation was performed using two different approaches: (1) Seurat Reference Mapping and Label Transfer approach, Bone Marrow Atlas was downloaded from DISCO[96] and was used as a reference to transfer labels to these data; (2) Manual curation was done by gene markers to reflect the prediction results. Lymphoid cells subset Analysis: T cells and NK cell clusters were subset and reclustered, and T cell subtypes were identified based on the canonical markers. CD4 Naive by CD4 and LEF1, CD4 $T_{regs}$ by FOXP3, IL2RA and IKZF2, Exhausted CD8 by CD8A, TOX and HAVCR2, Cytotoxic CD8 by GZMH, PRF1, NKG7, NK Cells by NCR3 and NCAM1, we identified some unique cell clusters like CD8_GZMK+, TNF_TCells and KLRB1+ TCells, IFN signaling-associated gene expressing TCell (ISAG_TCells) by OASL and ISG15, Proliferative gene expressing CD8 Dysfunctional cell cluster by MKI67, TOP2A, STMN1 and TUBB.

### Anti-ST2 antibodies, chromatography, and surface plasmon resonance (SPR)

Anti-mouse ST2 (IgG-281), anti-human ST2 (IgG-282), and control (IgG-283) VH and VL sequences were inserted into a mouse IgG1 Fc cassette carrying the D265 mutation for Fc silencing (U.S. patent application: 63/250,706). Each IgG was produced using the Expi293 expression system (Thermo Fisher Scientific RRID: CVCL_D615) or HEK 293T cells (RRID: CVCL_KS61). Antibodies were purified by affinity column chromatography, and stability was monitored by size exclusion chromatography-HPLC. An endotoxin level of <1 EU/mg was confirmed. The protocols for both chromatography and SPR were as follows: ST2 antibodies were dissolved with 10 µl PBS as the mobile phase

and pushed through a Superdex S200 10/300GL column (GE Healthcare) at a flow rate of 0.5 ml/min on an AKTA purifier (GE Healthcare)[97]. The percent monomer was calculated as the area of the monomeric peak divided by the total area of the monomeric plus nonmonomeric peaks at 280 nm. For SPR, recombinant murine or human ST2 was immobilized on CM5 sensor chips. The binding kinetics were monitored by flowing IgG-281 or IgG-282 over the chip for association, and dissociation from the surface was further monitored during a 5-min wash step.

### Pharmacokinetic analysis, serum ST2 measurement, and assessment of the direct effect of IgG-281 on $T_{regs}$ and CD8 T cells

Pharmacokinetic analysis was carried out as follows: plasma samples were drawn from mice of the same background after intravenous injection of ST2 antibodies at the indicated time points[97]. Serum concentrations of the ST2 antibodies were determined by enzyme-linked immunosorbent assay (ELISA) using plates coated with either murine ST2 (Sino Biologics; 51143-M08H) or human ST2 (Acro; IL1-H5229). WinNonlin software from Certara was used to perform a noncompartmental analysis of the serum concentrations.

On day 14 or 21 post-AML challenge, $ST2^+$, $ST2^-$ $T_{reg}$ cells, and CD8 T cells were isolated from malignant BM niches and seeded into a 96-well plate at a dose of $10^5$ per well. IgG-281 was added to the wells at the indicated doses. For proliferation assays, cells were labeled with CFSE, and CFSE low was measured after 18 or 24 h of coculture. For evaluation of apoptosis, cells were stained with Annexin V after 18 or 24 h of coculture and subjected to flow analysis. Data represents three independent tests.

### Chromatin immunoprecipitation coupled to quantitative PCR (ChIP-qPCR)

To explore whether ST2 binds to Tbet transcription factor in $T_{reg}$ cells could decrease its suppressive function in the malignant leukemia BM niche, ChIP-qPCR was used. $T_{reg}$ cells were isolated and purified from malignant BM niches using the immunomagnetic beads according to the manufacturer's protocol (Miltenyi Biotec, 130-091-041). $ST2^+T_{reg}$ cells were sorted from purified $T_{reg}$ cells by gating of $ST2^+CD4^+FITC$ (Foxp3$^{eGFP}$) cells with a BD FACS Aria (BD Bioscience). Chromatin immunoprecipitation (ChIP) assay of $ST2^+T_{reg}$ cells was performed using Pierce Agarose ChIP Kit (Thermo Scientific, Cat: 26156) according to the manufacturer's instructions. In brief, we crosslinked $ST2^+T_{reg}$ cells using 1% formaldehyde and quenched them with glycine solution. Then, the cell pellet was lysed using lysis buffer containing protease inhibitors and MNase digestion buffer containing micrococcal nuclease and stopped by adding MNase stop solution. We spared 10% of the DNA–protein mixture as input (before immunoprecipitation), while we incubated the remainder (after immunoprecipitation) with antibodies against mouse T-bet (3 µl; clone 39D; eBioscience), normal rabbit IgG (4 µl; kit provided) and normal IgG1 (2 µl, clone MG1-45, BioLegend) overnight at 4 °C. We enriched chromatin-protein using agarose resin, then reverse-crosslinked the protein-bound with protease K. Finally, the eluted ChIP and input DNA were purified with DNA wash buffer. We then performed real-time PCR for ChIP and input DNA using Scavenged Universal SYBR® Green Supermix (Cat: 1725270, Bio-Rad) with the published *IL1RL1* primer pairs as shown in Supplementary Table 5[29], and CHIP positive control GAPDH primers provided in the kit for quality control. Data are presented as fold enrichment of ST2 in $ST2^+T_{regs}$ before versus after T-bet IP, calculated using the comparative Ct method[98].

### Quantification and statistical analysis

Quantification methods are described in the figure legends. Unless otherwise noted, statistical analysis was performed using GraphPad Prism 10. The statistical tests applied in each experiment are described in the figure legends. Data are presented as mean ± standard error of the mean (s.e.m.). No data exclusion or outlier analysis was performed during the data acquisition and analysis. Briefly, phenotypic and functional data were compared using an unpaired *t*-test for comparison between two groups and analysis of variance (ANOVA) for comparison of three or more groups. To account for the type I error inflation due to multiple comparisons, we applied the Bonferroni correction. A log-rank test was used for survival analysis. All tests were two-sided at the significance level $P \leq 0.05$. For sample allocation, in all in vivo and in vitro experiments, animals and/or samples were randomly assigned to experimental groups. No specific method of randomization was used, but allocation was done in a manner that minimized potential bias. For patients' data, investigators were blinded to patients' outcomes during experimentation. For animal experiments, investigators were not blinded to group assignment during experimentation, but outcome assessments were performed using objective and standardized protocols. For sample size determination, no formal statistical power analysis was performed. Sample sizes were determined based on previous experience, feasibility, and consistency with standard practices in the field[99,100]. In all cases, sample sizes were sufficient to observe statistically significant and reproducible effects, and are consistent with those used in comparable published studies.

### Reporting summary

Further information on research design is available in the Nature Portfolio Reporting Summary linked to this article.

## Data availability

The RNA bulk sequencing data supporting the conclusions of the study have been deposited in the Gene Expression Omnibus (GEO) under accession identification GSE189688, and the patients' scRNA sequencing have been deposited in the Gene Expression Omnibus (GEO) under accession identification GSE279904. The following published datasets were used and appropriately cited: TCGA and TARGET-AML cohorts, Beat-AML, GTEx, GSE37642, GSE112417, GSE6891, and GSE9476. Source data are provided with this paper.

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

## Acknowledgements

This work was supported by a Cancer Moonshot U01CA232491 grant (to S.P. and N.-K.V.C). S.P. is partly supported by the Sally Abney Rose Endowed Chair in Cancer Stem Cell Biology and the Hollings Cancer Center grant (P30 CA138313). N.-K.V.C. is partly supported by the Enid A. Haupt Endowed Chair and Robert Steel Foundation. The Flow Cytometry core at Hollings Cancer Center, Medical University of South Carolina, is partially funded by a National Cancer Institute grant (P30 CA138313). We thank A. McKenzie from the Medical Research Council Laboratory of Molecular Biology, Cambridge, UK, for providing the ST2$^{-/-}$ mice. We thank Wade Clapp for helping with the generation of the ST2$^{fl/fl}$ mice. We thank Derek Stirewalt and Era Pogosova-Agagjanyan for providing patients' BM samples from the Fred Hutch/UW Hematopoietic Diseases Repository. We thank Jacob Kendrick, operation manager of the Flow Cytometry Resource Facilities at Hollings Cancer Center at the Medical University of South Carolina, for outstanding technical help with the imaging data; shared resources are supported by the Hollings Cancer Center grant (P30 CA138313).

## Author contributions

H.J., D.F., A.L., S.K.P., B.R., R.K., N.-K.V.C., and S.P. designed the research, analyzed the data, and wrote the manuscript. A.M.R., J.Y. performed analyses on leukemia patients' samples and in vivo models and interpreted the data. J.H.H. provided help with the design of the metabolic experiments. B.J.K. provided help with the design of the image flow cytometry experiments. A.L., H.G., and N.-K.V.C. designed and produced the neutralizing antibodies, including biochemical characterization and affinity evaluation. PK/PD. E.S., H.G., Y.L., D.M., S.S., and S.B. provided scientific input for bioinformatics analysis and discussion. S.K.P., B.R., and R.K. performed analyses on leukemia in vivo models, interpreted the data, provided essential materials, and provided intellectual input. A.G. and S.P.B. provided scientific input for the humanized leukemic model. S.P. conceptualized the project and supervised all aspects of this study.

## Competing interests

N.-K.V.C. reports receiving commercial research grants from Y-mAbs Therapeutics and Abpro-Labs Inc., holding ownership interest/equity in Y-mAbs Therapeutics Inc., holding ownership interest/equity in Abpro-Labs, and owning stock options in Eureka Therapeutics. N.-K.V.C. is the inventor and owner of issued patents licensed by Memorial Sloan Kettering Cancer Center (MSKCC) to Y-mAbs Therapeutics, Biotec Pharmacon, and Abpro-labs. Both MSKCC and N.-K.V.C. have financial interests in Y-mAbs. N.-K.V.C. is an advisory board member for Abpro-Labs and Eureka Therapeutics. Otherwise, the authors declare that they have no competing interests.
