## [Transparent Peer Review file · Nature Communications]

ST2/IL-33 axis blockade inhibits regulatory T cell cytotoxicity towards CD8 T cells in the leukemic niche

Corresponding Author: Professor Sophie Paczesny

Version 0:

Reviewer comments:

Reviewer #1

(Remarks to the Author)

>General comments

In this study, Hua Jiang & Denggang Fu et al. use several complementary approaches and models to address the role of IL-33/ST2 signaling on the bone marrow (BM) tissue microenvironment (TME) in the context of AML. The authors report that ST2+ regulatory T cells (Tregs) are increased in the BM of patients and mice with AML, as well as the continuous increase of these cells during AML progression. This correlated with a reduction of total CD8+ T cells, and an increased exhaustive status of these cells. BM-derived ST2+ Tregs expressed more IL-10, TGF- β , LAG3, TIM3, and GITR than ST2 negative Tregs, thus displaying enhanced suppressive capacity. Further, the authors show that genetic ablation of ST2 in Tregs prevents Tregs precursors from exiting lymph nodes, thereby resulting in improved CD8+ T cell immunity and increased numbers of activated CD8+ T cells specific for the AML WT1 antigen.

While addressing mechanisms, the authors find that ST2+ Tregs directly kill specific CD8+ lymphocytes in the AML TME. Using both immunocompetent and humanized pre-clinical murine models, the authors also show that anti-ST2 neutralizing antibodies can either limit or rescue AML progression depending on the leukemic burden, by protecting CD8+ T cells from cytotoxicity by ST2+ Treg cells.

Previous studies have already addressed the role of IL-33/ST2 signaling for AML, yet focusing mainly on the direct contribution of this pathway on leukemic cells, whereby ST2 was reported to be upregulated on AML cells compared to healthy CD34+ stem and progenitor cells (PMID: 37643244; 33324412). Others have previously used antibodies to target the IL1RAP, the co-receptor of ST2, for the selective killing of AML stem cells (PMID: 23479569). ST2 expression determines the phenotype and function of Tregs in different disease contexts (PMID: 31165767, 25043027, 31801068). Lastly, Tregs are upregulated in peripheral blood and BM of AML patients, where they have been suggested to play a regulatory role (PMID: 32456622, 32849603). However, the present findings on the role of ST2 positive or negative Tregs in AML are novel and add potential value to optimize immunotherapies for AML.

Overall, the experiments described in the manuscript are well-executed and the data are generally sound and convincing. The manuscript includes a plethora of data, some of which may potentially detract from the study's core message. The lack of precision in some instances makes the reading of the manuscript not straight forward and there are several mistakes. I provide below a detailed description of the aspects that should be improved.

>Major comments

- The reading of the manuscript is somewhat difficult: the authors shall be more precise in i) describing exactly which type of cellular populations are displayed in each FACS blot (% of cells among what population?); ii) clearly indicating which cells are ST2-/- (i.e. ST2 KO), WT, ST2+ or ST2- (among WT Tregs); iii) providing information on the origin of the cells shown in the different figures (for instance, BM would be more precise than TME as a tissue of origin, also considering my next comment on extramedullary hematopoiesis in AML). The type of AML model used should be clearly indicated in all legends.
- Is the reported effect of ST2 limited to BM-derived Tregs in the context of their AML model? The authors should test the suppressive capacity of such non-BM Tregs and compare it to the one of BM Tregs in AML mice. From the description, it is not clear whether this point has been addressed in the current manuscript. Note that models of MLL-AF9-driven leukemia may display the formation of extramedullary tumors, including in the spleen (PMID: 27344946). Although such TME is different from the BM, Treg in these locations may be similarly affected in an ST2-dependent manner.
- Lines 106 and thereafter: it should be better described in the manuscript text that the authors started from bulk RNA seq. data and applied deconvolution for these human data. The current presentation of the data suggests that they had access to single-cell data.

- Lines 210-214: the text does not correspond to the fig. that is called. The rationale for assessing central memory T cells and effector memory T cells (CD62L Low CD44 high) in their model should be indicated in the text. This description of the data should be re-checked. Line 216: CD62L Low CD44 high cells are effector memory T cells.
- Line 215: this statement is far-fetched; this is not exactly what the data show.
- Fig. 3A: since in their model ST2 expression determines the trafficking of Treg from lymph nodes to the BM, have the authors assessed the expression of corresponding homing / migration markers?
- Lines 244-246: even though ST2 and Tbet expressions are negatively correlated, it would be better to compare all combinations of ST2⁺/⁻ and Tbet⁺/⁻ as the current comparison Tbet neg. Tregs vs (Tbet pos) ST2⁺ Treg is not directly logical as currently presented and feels a bit forced. The same applies further to the data presented in Lines 264-265.
- Figure 4:
 - o Lines 247-248 / Fig. 4A: the description of the data in the text is partly incorrect. Cross-check.
 - o Fig. 4F: the PCA data are not helpful compared to Fig. 4G, as they show that the non-tumor group is closer to the WT group.
 - o Line 266 / Fig 4H: the text in the manuscript does not fit with the data shown (up versus down-regulation).
 - o Are the Treg functional phenotypes shown in Fig. 4 and SFig. 23 recapitulated when distinguishing, among WT Tregs, ST2-expressing versus ST2-negative cells (assuming that in Fig4. J,-L, the population designated as "ST2⁺ TME" represents *all* the WT FoxP3⁺ Tregs, as shown in the scheme in Fig. 4C)?
 - o At least for GZMB, MFI values should be shown for the different groups in Fig. 4J.
- Lines 281-286: rephrase: the results describe a ChIP-qPCR and not a co-immunoprecipitation assay (no testing of binding of ST2 protein to Tbet). In general, these data on the ChIP-qPCR assay should be more precisely described throughout the manuscript (e.g. also at Line 493).
- Line 350: which hypothesis?
- Fig 5A: upper left panel: why does the green staining for Treg appear either as a dot or as a ring (also indicating cells of different sizes)? Right panel: which Tregs are shown in this graph? ST2⁻ Tregs versus WT Tregs or WT Tregs that are either expressing ST2 or negative for ST2? This needs to be clearly indicated. The same applies to the different panels in this figure. What type of Tregs are used for Fig. 5B? Fig. 5D-J (mechanisms of CD8⁺ T cell killing) may appear not directly relevant to the core message of the study. The authors may consider removing these data.
- Fig S24: the data in panel A are not convincing. It is not clear why panel B does not directly show Blimp-1 versus ST2 expression. The data shown are misleading. Are there data with an isotype or FMO control available for these results?
- There should be a concluding alternative explanation (also if putative) on the mechanisms underlying the negative correlation between ST2 and Tbet expressions. Is Tbet (and Blimp-1 and Bcl-6) predicted to bind the ST2 promoter? Is ST2 expression in Treg regulated by multiple transcription factors? Note that in Th1 CD4⁺ T cells, T-bet regulates ST2 expression (acting together with STAT4 (PMID: 25829541)). In general, the discussion part on the different transcription factors studied in the frame of this study is too limited.
- AML hematopoietic stem and progenitor cells (may) upregulate ST2 (PMID 37643244; 33324412), and non-hematopoietic stromal cells in the BM may engage ST2 to support hematopoiesis (PMID: 26011644). These cell types, in addition to ST2⁺ Treg, should be discussed as further targets of anti-ST2 AML immune therapy.
- Line 495: the rationale for this statement should be better explained.

>Minor comments

- The last sentence of the abstract appears convoluted and should be rephrased.
- Wherever it applies, the fig. legends should contain information on the number of independent experiments the data represent. Or whether they include pooled data from multiple independent experiments. This information is currently not provided for all results where it may apply.
- Lines 77-78: «their AML antigen specificity»: rephrase/use more precise wording.
- Line 118: check Fig. number.
- Figure S1D/line 104: human or murine?
- The graphs in Fig. S2 are too small to allow for a proper comparison of the group.
- Fig S3: panel A: an isotype or FMO control should be provided for ST2 expression.
- Line: provide information in the text on the rationale to assess KLRG1 positivity on Tregs.
- Line 138 / Fig. 1D: KLRG1 is not indicated in this fig. or its legend. Same for Fig. S8.
- Lines 144-146: rephrase; the summarizing description of the findings should be more precise.
- Line 156: check Fig. number.
- Fig2D/line 181: these are not Ki67⁺ cells, Ki67⁺ cells are only shown in the supplementary data.
- Line 237: rephrase; the current wording is not logical.
- Line 261: what data are referred to ("as was found in naïve mice")? Indicate Fig. number.
- Lines 265-266: define precisely what is "activation" and "precursor-like" as these terms do not appear in the figures that are called.
- Lines 281/284: Fig. 4L and Fig. 4M are mixed up.
- Line 312: provide a rationale for the suggested bystander mechanism.
- Lines 315-316: provide more precision on what exactly is non-tumoral versus naïve.
- Lines 319-321: discrepancy between the description in the text versus the data in Fig. S28A. Fig. S28A: provide more precision on what exactly is normal versus non-leukemic. The tissues of origin of effector Treg and targets CD8 T cells should be indicated.
- Lines 328-329: discrepancy between the description in the text versus the data in Fig. S29. Check for the correctness of the labeling of the experimental group in the middle panel of Fig. S29 (e.g. typo in the red histogram plot: TME ST2⁺ Tregs).
- Line 368: which Fig. is called there? Fig. S31c, which is otherwise not called in the text?
- Discussion/Line 471: is IL-33 KO instead of ST2 KO meant? Otherwise, this sentence is not fully logical. The last part of

this sentence also appears to be incomplete.

- Lines 484-485: how does this previous publication of the group relate to the current findings?
- Line 513: provide more precision on what exactly are naive versus non-TME T cells. The same applies to Fig. 5B.
- Fig. S3, middle panel; the data presented are cropped along the Y axis and do not seem to entirely represent the cells gated in the first panel.
- Fig. S8, panel B; have bar graphs showing different mice per group (as in Fig. 8C) been omitted?
- Fig. S23: the AML model indicated in the figure does not match the one described in the legend.
- Fig. S31: panel A: do the peaks at 25 correspond to the diluent that was used?
- Fig. S33E is not called in the text.
- Fig. S34, legend of panel B: first or third administration of the therapeutic antibodies?
- Fig. S38: panel A: indicate the gate used for the selection of HLA-A2+ MOLM14 cells.
- Fig. S39: are similar data available for Tbet, which was also investigated?
- Lines 711/712: what was the concentration of IL-33 or GZMB inhibitor used?
- Line 738, 308, 304: hrs instead of hours as before
- Fig. 6B: is there an unspecific effect of the antibody on ST2- Tregs at higher concentrations of antibodies? The results suggest it.
- Line 389: does "normal" mean healthy/WT?

Reviewer #2

(Remarks to the Author)

In this study, Jiang et al. investigated the role of ST2+Treg cells in the tumor microenvironment (TME) of AML. The authors observed that ST2+Treg cells were notably enriched in the BM niche during AML development and exhibited an inverse correlation with CD8+ effector T cell status in TME. ST2^{-/-} Treg cells or specifically knocking out ST2 in Foxp3+ Treg cells were shown to restore CD8+ effector T cell function and mitigate AML development. Mechanistically, authors showed that ST2+Tregs-derived Granzyme B (GzmB) directly induced cell death of CD8+ effector T cells in a co-culture system. Lastly, the authors employed mouse and human ST2-specific blocking antibodies and demonstrated the anti-AML effects of ST2-blockade in both mouse AML and humanized AML models. Identification of the role of ST2+Tregs in the AML niche is new. However, the authors may need to address the following major comments:

1. ST2 is expressed by basophils, eosinophils, mast cells, and their progenitors (PMID: 27568595 and ref 42). The authors may show whether ST2 expression is higher on Tregs than those cells. What happened to ST2+ cells other than Tregs (including the mentioned granulocytes) after anti-ST2 antibody treatment? Did those non-Treg ST2+ cells contribute to the AML phenotype described in this study?
2. In vitro co-culture results showed that ST2+Treg cells can directly kill TME-derived CD8+ T cells in a GzmB-dependent manner. Did GzmB contribute to ST2+Treg cell-mediated suppression on CD8+ T cells in the in vivo system? In addition to GzmB, RNAseq data showed significant enrichment of multiple immunoregulatory molecules, such as IL-10 and Areg in ST2+ Treg cells. Did these molecules contribute to the suppression of CD8+ T cells by ST2+Treg cells? Apart from inducing cell death, did ST2+Treg cells suppress the activation or proliferation of CD8+ T cells in TME?
3. ST2+ Treg cells selectively targeted and killed TME-derived CD8+ effector T cells, suggesting that the cytotoxicity exerted by ST2+ Treg cells operates in an antigen-restricted manner. Typically, cytotoxic T cells or CD8+ T cells eliminate intracellular antigen-infected host cells by releasing GzmA/B and perforin. Therefore, ST2+ Treg cells exhibit certain characteristics of CD8+ T cells. Do ST2+ Tregs kill AML cells? What is the overall function of GzmB in the AML niche? Could GzmB inhibitor decrease AML development?
4. Do ST2+ Treg localize closely to CD8+T cells within the AML BM niche?
5. In the AML transplantation models, AML cells tend to engraft the liver and spleen besides BM. What are the roles of ST2+ Tregs in spleen and liver in which the transplanted mouse AML cells are engrafted? In particular, did ST2+ Tregs inhibit CD8+ T cells in liver, spleen, and peripheral blood in the experiment of Fig 2D? What happened to ST2+ Tregs in these organs after ST2 blockade in Fig 6?
6. In Fig 2H-O, the authors utilized Foxp3-cre and Foxp3-cre; ST2^{fl/fl} mice as recipient mice to investigate MLL-AF9 AML development and corresponding immune profiles. Did the recipient mice receive lethal irradiation before MLL-AF9 cells were transplanted? If so, how did such an irradiation affect ST2+Tregs and other immune cells described in the experiment?

Reviewer #3

(Remarks to the Author)

I co-reviewed this manuscript with Professor Chengcheng(Alec) Zhang who provided the listed reports.

Reviewer #4

(Remarks to the Author)

In this manuscript Jiang, et al demonstrated that a subset of Treg cells expressing IL-33 Receptor (ST2) is enriched in the AML bone marrow (BM) microenvironment and promotes CD8+ T cell depletion and exhaustion, while ST2 deficiency in Treg cells restores CD8+ T cell function. The authors suggest that ST2+ Treg cells do not express Tbet, IFN γ , and Bcl6, and exhibit increased granzyme B-mediated cytotoxicity, leading to the enhanced killing of intratumoral CD8+ T cells. Using engineered anti-ST2 antibodies, they finally show induction of ST2+ Treg cell apoptosis and extended survival in aggressive AML models.

Although the work is interesting, suggesting that targeted blockade of ST2 signaling in Treg cells may be useful for novel immune-based therapies in AML, the current version of the manuscript appears to be premature and is not based on a solid mechanistic basis. Additional experimentation, taking advantage of single-cell based technologies, such as scRNA-seq and T-cell Receptor repertoire sequencing (TCR-seq), would be fundamental for such studies. This approach would shed light on the transcriptomic changes within the BM microenvironment, unravel cell-cell interactions between ST2+Treg cells and BM niche subtypes, and delineate the TcR-antigen recognition mechanism. The authors should further explore the underlying mechanisms, and better characterize the changes in activated ST2+ Treg cells, and how those cells differ from rest ST2+ Tregs, or total Treg populations in AML conditions VS steady-state conditions. Analysis of Tregs from BM of AML patients in different stages of disease including Diagnosis, Remission, Relapse would be very important, and validation studies on additional mouse models of AML by coupling flow cytometry data with single-cell technology and imaging approaches to visualize the histological findings would strengthen the manuscript. The blocking experiments shown in Fig. 6, should be repeated in additional mouse models of AML, other than the MLL-AF9 model, and/or in AML Patient-derived xenografts (PDX) to confirm that the findings are not specific to the aggressive models analyzed. Finally, the authors should consider simplifying the manuscript and better highlighting their findings in the main figures.

Version 1:

Reviewer comments:

Reviewer #1

(Remarks to the Author)

The authors have addressed all my comments well; however, I still have one minor comment:

Line 390: The authors suggest a negative effect on NK cell survival via bystander secretion of TGF β and IL-10 by ST2+ Treg cells. However, Fig. S17 indicated that this effect is abrogated using the Z-AAD-CMK GZMB inhibitor, suggesting rather an active killing mechanism.

Reviewer #2

(Remarks to the Author)

The authors have addressed my concerns.

Reviewer #3

(Remarks to the Author)

Reviewer #4

(Remarks to the Author)

The revised manuscript is an improvement over the original version. The authors have included additional models of AML in their analysis. They have also performed histological analysis of mouse spleens, which further supports their findings histologically. However, something that is missing is the histology analysis of the bone marrow from their mice. While deparaffinization and Immunofluorescence staining can be tricky, they are feasible on paraffin sections from bone tissues.

Point-by-point responses to reviewers (indicated by blue font in the responses and highlighted in yellow in the manuscript).

Reviewer 1:

Major comments

1) The reading of the manuscript is somewhat difficult: the authors shall be more precise in i) describing exactly which type of cellular populations are displayed in each FACS blot (% of cells among what population?); ii) clearly indicating which cells are ST2^{-/-} (i.e. ST2 KO), WT, ST2⁺ or ST2⁻ (among WT Tregs); iii) providing information on the origin of the cells shown in the different figures (for instance, BM would be more precise than TME as a tissue of origin, also considering my next comment on extramedullary hematopoiesis in AML). The type of AML model used should be clearly indicated in all legends.

The manuscript has been streamlined.

2) Is the reported effect of ST2 limited to BM-derived Tregs in the context of their AML model? The authors should test the suppressive capacity of such non-BM Tregs and compare it to the one of BM Tregs in AML mice. From the description, it is not clear whether this point has been addressed in the current manuscript. Note that models of MLLAF9- driven leukemia may display the formation of extramedullary tumors, including in the spleen (PMID: 27344946). Although such TME is different from the BM, Treg in these locations may be similarly affected in an ST2-dependent manner.

We appreciate the reviewer's comment, and we agree with the reviewer that extramedullary tumors, including in the spleen and the liver are seen in the MLLAF9- driven leukemia. In the MLLAF9- driven leukemia mice, spleen ST2⁺ Tregs display the same phenotype and killing profile as BM ST2⁺ Tregs from these mice. Killing activity of Spleen and liver ST2⁺ Tregs were previously described in Fig. S28A, text line 321. We have now added the phenotype of leukemic spleen and liver ST2⁺ Tregs earlier in Fig. 1F (see also below). In addition, we used the Dnmt3afl/+ Cre+/Flt3ITD/+ AML transplantation model, which produces

extramedullary malignancies in the spleen (PMID: 27016502, 36073548) (see Fig. 1C). We have also changed the subtitle to "Activated ST2⁺ Treg cells are specifically enriched in malignant leukemic niches".

3) Lines 106 and thereafter:

it should be better described in the manuscript text that the authors started from bulk RNA seq. data and

applied deconvolution for these human data. The current presentation of the data suggests that they had access to single-cell data.

Sorry for the confusion, we have clarified this point.

4) Lines 210-214: the text does not correspond to the fig. that is called. The rationale for assessing central memory T cells and effector memory T cells (CD62L Low CD44 high) in their model should be indicated in the text. This description of the data should be re-checked. Line 216: CD62L Low CD44 high cells are effector memory T cells.

Sorry for the oversight. This has been called for; our focus was on CD44^{high} and CD62L^{high} which is the central memory subset and Fig S20B has been updated and is now Figure 3Q (see below). The rationale is now in the text "Since antigen-specific central memory T cells provide a better antitumoral immunity compared to effector memory T cells due to their ability to rapidly recirculate through secondary lymphoid organs, allowing for sustained antigen presentation and a more robust immune response against cancer cells when re-exposed to tumor antigens (PMID: 15980149), we explored the frequency of central memory CD8⁺ T cells within WT1-tetramer⁺ CD8⁺ T cells."

5) Line 215: this statement is far-fetched; this is not exactly what the data show.

This statement has been modified.

6) Fig. 3A: since in their model ST2 expression determines the trafficking of Treg from lymph nodes to the BM, have the authors assessed the expression of corresponding homing / migration markers? Figure 3 is now Figure 4. We have now assessed several migration markers, added in Figure 4C and in the text lines 285-295. "Since our model suggested ST2 expression determines the trafficking of Treg cells from lymph nodes to the BM, we next assessed the expression of corresponding migration markers on ST2⁺ Treg cells compared to ST2⁻ Treg cells. There was an increased expression of CCR2, CCR5, CCR6, CCR7, CCR8, CCR9, CXCR3, and CXCR6 in on ST2⁺ Treg cells versus ST2⁻ Treg cells in both the healthy and leukemic BM niche (Fig. 4C, also below). This chemokine receptors analysis infers that the ST2⁺ Treg cells has a high migratory potential. Importantly, CXCR4 expression was significantly increased in ST2⁺ Treg cells compared to ST2⁻ Treg cells solely in the leukemic niche (Fig. 4C)." Discussion has also been updated accordingly.

7) Lines 244-246: even though ST2 and Tbet expressions are negatively correlated, it would be better to compare all combinations of ST2⁺/⁻ and Tbet⁺/⁻ as the current comparison Tbet neg. Tregs vs (Tbet pos) ST2⁺ Treg is not directly logical as currently presented and feels a bit forced. The same applies further to the data presented in Lines 264-265.

We appreciate the reviewer's comment. We have now streamlined the comparisons. The experiment in the naïve mice was a hypothesis-driven discovery experiment based on the facts that IFN- γ in T_{reg} cells has been shown to promote anti-tumor immunity¹, and Tbet (encoded by Tbx21 gene) is required for optimal IFN- γ production in CD4⁺ T cells². While Tbet can bind to the ST2 promoter in the context of Th1 cells³, the relationship of Tbet and ST2 in T_{reg} cells is still unknown. Therefore, we investigated whether ST2⁺ T_{reg} cells are correlated with Tbet expression in both steady and pathogenic states. We hypothesized that BM-WT (ST2⁺) T_{reg} cells will be inversely correlated with Tbet expression in healthy and leukemic mice.

We are now showing the data related to the steady state BM in supplementary Fig S14 while the data of leukemic BM niche comparing the 4 groups are shown in new Fig. 5D-E.

8) Figure 4:

Note previous Figure 4 is now Figure 5.

9) Lines 247-248 / Fig. 4A: the description of the data in the text is partly incorrect. Cross-check.

This was edited for accuracy.

10) Fig. 4F: the PCA data are not helpful compared to Fig. 4G, as they show that the non-tumor group is closer to the WT group.

We agree with the reviewer, and we removed it.

11) Line 266 / Fig 4H: the text in the manuscript does not fit with the data shown (up versus down-regulation).

Sorry for this oversight. To allow for more streamlining, we have now removed all pathways analysis.

12) Are the Treg functional phenotypes shown in Fig. 4 and SFig. 23 recapitulated when distinguishing, among WT Tregs, ST2-expressing versus ST2-negative cells (assuming that in Fig4. J,-L, the population designated as “ST2+ TME” represents *all* the WT FoxP3+ Tregs, as shown in the scheme in Fig. 4C)?

The reviewer is correct, the schema now in Fig. 5A (previously 4C) summarizes the experiment and “ST2+ TME” represents *all* the WT FoxP3+ Tregs. We have now added Fig. S15 that summarizes BM-derived Tregs functional phenotypes in this group distinguishing among ST2-expressing versus ST2-negative cells.

13) At least for GZMB, MFI values should be shown for the different groups in Fig. 4J.

GZMA MFI has been added in new Fig.5F (former Fig 4J) and GZMB MFI in new Fig. 5G (previous Fig4H)

14) Lines 281-286: rephrase: the results describe a ChIP-qPCR and not a co-immunoprecipitation assay (no testing of binding of ST2 protein to Tbet). In general, these data on the ChIP-qPCR assay should be more precisely described throughout the manuscript (e.g. also at Line 493).

This has been edited.

15) Line 350: which hypothesis?

At your suggestion below, experiments related to the metabolism of intratumoral CD8⁺ T cells have been removed and the related text as well.

16) Fig 5A: upper left panel: why does the green staining for Treg appear either as a dot or as a ring (also indicating cells of different sizes)?

Note that previous Fig 5 is now Fig 6. ST2+ and ST2- Treg cells were sorted from the malignant leukemic BM niche at day 21 post-MLL-AF9 injection, using a Miltenyi Treg kit combined with ST2-PE Ab/PE beads. On the right is a representation of images obtained from the Imagestream for ST2+ Treg cells. They have different size and shape that we may attribute to cytoskeletal changes seen with AML antigen-triggering (reference PMID: 31894880). This is beyond our expertise and not studied here.

17) Right panel: which Tregs are shown in this graph? ST2-/- Tregs versus WT Tregs or WT Tregs that are either expressing ST2 or negative for ST2? This needs to be clearly indicated. The same applies to the different panels in this figure.

This has been clarified with schemas for the different panels in new Fig 6.

18) What type of Tregs are used for Fig. 5B?

Fig 5B. is now Fig 6 E. Ex vivo Tregs from the BM of mice with leukemia were sorted at day 21 (like what was done in schema Fig. 6 D).

19) Fig. 5D-J (mechanisms of CD8+ T cell killing) may appear not directly relevant to the core message of the study. The authors may consider removing these data.

We agree with the reviewer and have removed this part of the data which helps with the streamlining of the manuscript as well.

20) Fig S24: the data in panel A are not convincing. It is not clear why panel B does not directly show Blimp-1 versus ST2 expression. The data shown are misleading. Are there data with an isotype or FMO control available for these results?

Fig. S24 is now Fig. 5 K and L. We are now showing FMOs as well as BLIMP-1 versus ST2.

21) There should be a concluding alternative explanation (also if putative) on the mechanisms underlying the negative correlation between ST2 and Tbet expressions. Is Tbet (and Blimp-1 and Bcl-6) predicted to bind the ST2 promoter? Is ST2 expression in Treg regulated by multiple transcription factors? Note that in Th1 CD4+ T cells, T-bet regulates ST2 expression (acting together with STAT4 (PMID: 25829541)). In general, the discussion part on the different transcription factors studied in the frame of this study is too limited.

We agree with the reviewer and thank him for the useful reference. We have expanded the rationale and discussed in details the different transcription factors studied.

22) AML hematopoietic stem and progenitor cells (may) upregulate ST2 (PMID 37643244; 33324412), and nonhematopoietic stromal cells in the BM may engage ST2 to support hematopoiesis (PMID: 26011644). These cell types, in addition to ST2+ Treg, should be discussed as further targets of anti-ST2 AML immune therapy.

We agree and have now discussed these other ST2+ targets lines 655-665. See also response to reviewer 2 regarding other immune cells expressing ST2.

23) Line 495: the rationale for this statement should be better explained. This paragraph has been updated.

Minor comments

- The last sentence of the abstract appears convoluted and should be rephrased.

This sentence has been changed.

- Wherever it applies, the fig. legends should contain information on the number of independent experiments the data represent. Or whether they include pooled data from multiple independent experiments. This information is currently not provided for all results where it may apply.

This has been updated.

- Lines 77-78: «their AML antigen specificity»: rephrase/use more precise wording.

Rephrased, Thanks.

- Line 118: check Fig. number. Now line 121 should be Fig S2.
- Figure S1D/line 104: human or murine? Fig S1 A and B = human and Fig S1 C and D = murine.
- The graphs in Fig. S2 are too small to allow for a proper comparison of the group. Size and quality have been increased.
- Fig S3: panel A: an isotype or FMO control should be provided for ST2 expression. Now Figure S4. We now provide a FMO in Fig. S4C.
- Line: provide information in the text on the rationale to assess KLRG1 positivity on Tregs. Provided in lines 133-137 of the revised manuscript.

- Line 138 / Fig. 1D: KLRG1 is not indicated in this fig. or its legend. Same for Fig. S8. We have shown that in leukemia as was shown previously in homeostasis, and other diseases that a large percentage of tissue ST2⁺ Treg cells express KLRG1 and showed that frequencies of KLRG1⁺ST2⁺ Treg cells were correlated with total ST2⁺ Treg cells although slightly smaller. Starting from New Fig. 1G (old Fig.1D) we continued using a gating on the total ST2⁺ Treg cells.
- Lines 144-146: rephrase; the summarizing description of the findings should be more precise. Agreed, this has been rephrased.
- Line 156: check Fig. number. These have been reordered and checked. It should be new Fig. S4D.
- Fig2D/line 181: these are not Ki67⁺ cells, Ki67⁺ cells are only shown in the supplementary data. There has been a significant streamlining and data presenting in main figures at the request of several reviewers. New Fig. 3D shows frequencies and numbers of total CD8⁺ T cells. New Fig. S12 shows frequencies of proliferating Ki67⁺CD8⁺ T cells.
- Line 237: rephrase; the current wording is not logical. See new line 291.
- Line 261: what data are referred to (“as was found in naïve mice”)? Indicate Fig. number. This was removed and clarified in the paragraph above.
- Lines 265-266: define precisely what is “activation” and “precursor-like” as these terms do not appear in the figures that are called. We have removed the section on the GO pathways.
- Lines 281/284: Fig. 4L and Fig. 4M are mixed up. These have been reordered and are now new Fig. 5I (Tbet expression in ST2⁻ Treg cells as compared to ST2⁺ Treg cells) and Fig. 5J (ChiP-qPCR).
- Line 312: provide a rationale for the suggested bystander mechanism. See new line 390.
- Lines 315-316: provide more precision on what exactly is non-tumoral versus naïve. Naïve CD8⁺ T cells were CD45RA sorted.
- Lines 319-321: discrepancy between the description in the text versus the data in Fig. S28A. Fig. S28A: provide more precision on what exactly is normal versus non-leukemic. The tissues of origin of effector Treg and targets CD8 T cells should be indicated. This is now new Fig. 6F and line 397; details have been provided.
- Lines 328-329: discrepancy between the description in the text versus the data in Fig. S29. Check for the correctness of the labeling of the experimental group in the middle panel of Fig. S29 (e.g. typo in the red histogram plot: TME ST2⁺Tregs. Fig. S29 is new Fig. 6I. The red histograms represent the in vivo proportion of non-TME CD8⁺ T cells (CFSE^{low}) and TME CD8⁺ T cells (CFSE^{high}) after adoptive transfer at baseline (Hr 0; equal proportion) and at Hr 24 (decreased proportion in the population that has been killed). The legend and text were correct, but additional details should help clarify the experiment.
- Line 368: which Fig. is called there? Fig. S31c, which is otherwise not called in the text? Fig. S31C, now new Fig. S17C is called in the text line 427.
- Discussion/Line 471: is IL-33 KO instead of ST2 KO meant? Otherwise, this sentence is not fully logical. The last part of this sentence also appears to be incomplete. This paragraph has been edited (lines 669-679). We have performed our experiment in ST2 KO which eliminates all potential signaling through IL-33 which has been more reliable than IL33-KO in our hands.
- Lines 484-485: how does this previous publication of the group relate to the current findings? It does not directly relate to the current study and was removed.

- Line 513: provide more precision on what exactly are naive versus non-TME T cells. The same applies to Fig. 5B. New line 622 and Fig. 6E include detailed description of the populations.
- Fig. S3, middle panel; the data presented are cropped along the Y axis and do not seem to entirely represent the cells gated in the first panel. Thanks for noticing. This is now Fig. S4, and the cropping issue has been fixed.
- Fig. S8, panel B; have bar graphs showing different mice per group (as in Fig. 8C) been omitted? Representative plots were shown in the supplementary material while the bar graphs were shown in the main figure. Data for these experiments are now shown together and presented in Fig. 1G (MLL-AF9) and 1H (DNMT3A/FLT3^{TD}).
- Fig. S23: the AML model indicated in the figure does not match the one described in the legend. Sorry for the oversight both models were shown. This is now Fig. 5H and I and the legend has been fixed.
- Fig. S31: panel A: do the peaks at 25 correspond to the diluent that was used? New Fig. S17. This is correct.
- Fig. S33E is not called in the text. Figure S33 has now been added in new Fig. 7I-L, old Fig S33E is in panel Fig. 7L with the GI toxicity.
- Fig. S34, legend of panel B: first or third administration of the therapeutic antibodies? Now Fig. S19, collection of the samples for analysis were done 21 days after the first administration of the antibodies which happened at D0.
- Fig. S38: panel A: indicate the gate used for the selection of HLA-A2+ MOLM14 cells. Now Fig. S20. This was gated on total cells.
- Fig. S39: are similar data available for Tbet, which was also investigated? These data are now in new main Fig. 7N.
- Lines 711/712: what was the concentration of IL-33 or GZMB inhibitor used? IL-33 (R&D Systems #3626-ML) at 20 ng/ml or GZMB inhibitor Z-AAD-CMK (Millipore-Sigma #368050) at 50 ng/ml. This has been added in the legend and methods.
- Line 738, 308, 304: hrs instead of hours as before. These have been changed.
- Fig. 6B: is there an unspecific effect of the antibody on ST2- Tregs at higher concentrations of antibodies? The results suggest it. We used a Miltenyi kit to sort ST2+ Tregs and ST2- Tregs. The purity is between 85 and 95%. Therefore, they could be up to 15% of contaminating ST2+ Tregs cells explaining killing that might still be specific. The antibody sequence that has been developed by Centocor is highly specific and does not have a target other than ST2. In addition, the sorting procedure takes a few hrs, and the cells may have a low level of unspecific apoptosis.

References

- 1 Overacre-Delgoffe, A. E. *et al.* Interferon-gamma Drives T(reg) Fragility to Promote Anti-tumor Immunity. *Cell* **169**, 1130-1141 e1111, doi:10.1016/j.cell.2017.05.005 (2017).
- 2 Lugo-Villarino, G., Maldonado-Lopez, R., Possemato, R., Penaranda, C. & Glimcher, L. H. T-bet is required for optimal production of IFN-gamma and antigen-specific T cell activation by dendritic cells. *Proc Natl Acad Sci U S A* **100**, 7749-7754, doi:10.1073/pnas.1332767100 (2003).
- 3 Baumann, C. *et al.* T-bet- and STAT4-dependent IL-33 receptor expression directly promotes antiviral Th1 cell responses. *Proc Natl Acad Sci U S A* **112**, 4056-4061, doi:10.1073/pnas.1418549112 (2015).

Reviewer 2:

1. ST2 is expressed by basophils, eosinophils, mast cells, and their progenitors (PMID: 27568595 and ref 42). The authors may show whether ST2 expression is higher on Tregs than those cells. What happened to ST2+ cells other than Tregs (including the mentioned granulocytes) after anti-ST2 antibody treatment? Did those non-Treg ST2+ cells contribute to the AML phenotype described in this study? We thank the reviewer for his critique. We agree that ST2 can be expressed by other immune cells including granulocytes. To address the role and weight of ST2+ T_{reg} cells versus non-Treg immune ST2+ cells in AML development we performed several experiments, some of which were already presented but now have been experimentally clarified. Some experiments are novel.

- 1) New Fig.3A shows the results of an adoptive transfer in a lethally irradiated mice that contained only AML, BM cells, T cells and Tregs cells and does not contain persistent immune cells or mature myeloid cells. Mice transplanted with ST2^{-/-}Foxp3^{eGFP+} T_{reg} cells vs WT Foxp3eGFP+ Treg cells had an extended survival, less AML growth, and an increase in CD8⁺Tbet⁺ and CD8⁺IFN γ ⁺ T cells and less CD8+ PD1+ exhaustion (Fig. 3B-F).
- 2) To further establish the weight of leukemic ST2+ T_{reg} cells in the overall role of ST2 in the BM niche, we used a non-irradiated leukemic model and injected MLL-AF9 leukemia cells into WT or ST2^{-/-} mice and additionally depleted T_{reg} cells with anti-CD25 antibody (Clone No.: PC61) on day 1 and day 7. Approximately, half of the survival increase was due to T_{reg} cells (anti-CD25 in WT mice) and half to other non-Treg ST2 expressing cells (isotype in ST2^{-/-} mice) (Fig. 3G).
- 3) To confirm the specific and unique role of ST2 in leukemic T_{reg} cells, we generated *Foxp3^{Cre}ST2^{fl/fl}* and used these mice with the non-irradiated MLL-AF9 leukemic model (Fig. 3H). In this model where ST2 is deleted in all T_{reg} cells, we observed that compared with *Foxp3^{Cre}* leukemic controls, *Foxp3^{Cre}ST2^{fl/fl}* leukemic mice had longer survival and a lower leukemia burden (Fig. 3I). In this model, the frequencies of either myeloid cells or macrophages, granulocytes, monocytes, and B cells did not differ between groups. Therefore, the sole deficiency of ST2 in T_{reg} cells suffice to inhibit MLL-AF9 AML growth and prolong survival.
- 4) Furthermore, we used the epigenetic *Dnmt3a^{fl/+} Cre+/Flt3^{ITD/+}* AML model. In the context of this *Dnmt3a^{fl/+} Cre+/Flt3^{ITD/+}* AML model monocytes, and neutrophils are well-recognized for their roles in AML progression. The treatment with anti-murine ST2 Ab in this model led to a significant reduction in the tumor burden in the spleen and liver (Fig. 8A-B). Additionally, frequencies of monocytes (CD11b+Gr1intF4/80+), and granulocytes, including neutrophils (Gr1+CD11b+) in both the peripheral blood (not shown) and spleen were decreased in the anti- ST2 Ab treated group compared to the control antibody group. These findings reveal that non-Treg ST2+ cells contribute to the *Dnmt3a^{fl/+} Cre+/Flt3^{ITD/+}* AML.

2. In vitro co-culture results showed that ST2+Treg cells can directly kill TME-derived CD8+ T cells in a GzmB-dependent manner. Thanks for these comments. We realized that some of these experiments needed more details and we have added in new Fig.6, schema explaining the different experiments.

Did GzmB contribute to ST2+Treg cell-mediated suppression on CD8+ T cells in the in vivo system? Pharmacological inhibition of GzmB, which is a serine protease Inhibitor, in vivo has been attempted only topically in autoimmune blistering diseases where GzmB is expressed in lesional skin (PMID: 33436591). Therefore, a VTI-1002 topical gel was used on the skin of these mice. Systemic administration of pharmacological GzmB inhibitor has not been attempted due to potential high toxicity. Furthermore, in cancer, GzmB is secreted as a cytolytic molecule only when it recognizes its target. It has been show previously that adoptive transfer of granzyme B-deficient Treg cells versus WT Treg cells extended survival of mice with AML (PMID: 17919943).

In addition to GzmB, RNAseq data showed significant enrichment of multiple immunoregulatory molecules, such as IL-10 and Areg in ST2+ Treg cells. Did these molecules contribute to the suppression of CD8+ T cells by ST2+Treg cells? We have performed coculture showing that AML-primed ST2+Treg cells killed solely TME CD8+ T cells and solely through a contact dependent mechanism (Fig. 6B). Thus, TGFβ and IL-10 which are secreted cannot be responsible for the killing effect of ST2+Treg cells. Areg seen at the surface of ST2+Treg cells need to be recognized by the EGFR receptor and since CD8+ T cells do not express EGFR we did not pursue this avenue for the cause of the killing.

Apart from inducing cell death, did ST2+Treg cells suppress the activation or proliferation of CD8+ T cells in TME?

ST2+Treg cells suppress the proliferation of CD8+ T cells *in vitro* (Fig. 1J) and *in vivo* (Fig 2A and 2B (3rd panel)). ST2+Treg cells suppress the activation of CD8+ T cells *in vivo* (Fig 2E).

3. ST2+ Treg cells selectively targeted and killed TME-derived CD8+ effector T cells, suggesting that the cytotoxicity exerted by ST2+ Treg cells operates in an antigen-restricted manner. Typically, cytotoxic T cells or CD8+ T cells eliminate intracellular antigen-infected host cells by releasing GzmA/B and perforin. Therefore, ST2+ Treg cells exhibit certain characteristics of CD8+ T cells. We agree with this statement.

Do ST2+ Tregs kill AML cells? ST2+ Tregs do not kill AML cells (see Fig. 6C) but only AML primed CD8+ T cells as well as CD4+ T cells.

What is the overall function of GzmB in the AML niche? Could GzmB inhibitor decrease AML development?

See also answer to your comment # 2. GzmB overall function in AML has been previously studied in the Cao's et al. manuscript (PMID: 17919943) and the survival extension was attributed to GzmB expression by Treg cells. Due to toxicity concerns of the use of a systemic GzmB pharmacological inhibition we did not attempt *in vivo* GzmB inhibition. That said, our *in vitro* cocultures with or without transwell convincingly show that the killing of TME CD8+ T cells by TME ST2+ Tregs is contact dependent and inhibited by the GZMB-specific inhibitor Z-AAD-CMK at 50 ng/ml.

4. Do ST2+ Treg localize closely to CD8+T cells within the AML BM niche?

Using *in vitro* coculture with or without transwell, we have shown that TME ST2+ Tregs kill TME CD8+ T cells in a contact dependent manner (Fig. 6B).

Since decalcification methods of BM biopsy specimens are complex, we used the epigenetic *Dnmt3a^{fl/+} Cre+/Flt3^{ITD/+}* that develop frequent extramedullary leukemia in the spleen and liver and performed confocal imaging to analyze the colocalization of ST2+ Treg and CD8+T cells within the AML spleen niche (Fig.8E).

5. In the AML transplantation models, AML cells tend to engraft the liver and spleen besides BM. What are the roles of ST2+ Tregs in spleen and liver in which the transplanted mouse AML cells are engrafted? In particular, did ST2+ Tregs inhibit CD8+ T cells in liver, spleen, and peripheral blood in the experiment of Fig 2D? What happened to ST2+ Tregs in these organs after ST2 blockade in Fig 6?

We agree with the reviewer, MLLAF9- driven leukemia may display the formation of extramedullary tumors, including in the spleen (PMID: 27344946). As mentioned above the epigenetic *Dnmt3a^{fl/+} Cre+/Flt3^{ITD/+}* develop even more frequent extramedullary leukemia. Thus, we have now tested that spleen ST2+ Tregs from MLLAF9- and *Dnmt3a^{fl/+} Cre+/Flt3^{ITD/+}* driven leukemia display the same phenotype and killing profile as BM ST2+ Tregs (see Fig. 1F, and 6F). Of note, killing activity of these cells were previously described in Fig. S28A, text line 321.

6. In Fig 2H-O, the authors utilized Foxp3-cre and Foxp3-cre; ST2fl/fl mice as recipient mice to investigate MLL-AF9 AML development and corresponding immune profiles. Did the recipient mice receive lethal irradiation before MLL-AF9 cells were transplanted? If so, how did such an irradiation affect ST2+Tregs and other immune cells described in the experiment?

Fig 2H-O is now Fig. 3H-Q. The model with Foxp3-cre and Foxp3-cre ST2fl/fl mice as recipient mice did not receive irradiation (see new Fig. 3H schema of the model). Indeed, MLL-AF9 can engraft without irradiation, and this is the model we used for the Foxp3-cre and Foxp3-cre ST2fl/fl mice.

Reviewers 3 and 4:

Additional experimentation, taking advantage of single-cell based technologies, such as scRNAseq and T-cell Receptor repertoire sequencing (TCR-seq), would be fundamental for such studies. This approach would shed light on the transcriptomic changes within the BM microenvironment, unravel cell-cell interactions between ST2+Treg cells and BM niche subtypes, and delineate the TcR-antigen recognition mechanism.

We thank the reviewer for these suggestions. It is to note that in a previous study that explored AML patients' samples at diagnosis using single cell RNA sequencing with 10X genomics method (PMID: 30827681), the investigators were not able to see an increase of total FOXP3⁺ T_{reg} cells even though it was visible at the protein level using immunohistochemistry. We interpreted these data as the limitation of the current 10X genomics method that capture less than 10,000 cells at best and the fact that the relative frequencies of FOXP3⁺ Treg cells captured is extremely low. To circumvent this limitation and attempt to identify the rare immune populations in BM samples, we selected patients with BM samples after chemotherapy induction with no more than 20% blasts and performed both single cell RNA sequencing with 10X genomics method and flow cytometry. We compared AML from complete responders (CR) vs. refractory patients (Ref.). The data are now shown in Fig. S3. Single cell RNA sequencing showed on the Uniform Manifold Approximation and Projection (UMAP), several subsets of T cells including FOXP3⁺ Treg cells (Fig. S3A) with an overrepresentation of FOXP3⁺ Treg cells, exhausted CD8⁺ T cells and dysfunctional CD8⁺ T cells in the Ref group vs CR group (Fig. S3B). However, a distinct subset of ST2(IL1RL1)⁺ Treg cells was not seen. We agree that single cell RNA sequencing including T-cell Receptor repertoire sequencing (TCR-seq) could help delineate the TcR-antigen recognition mechanism. However, since the patients' samples do not contain enough BM cells to be able to sort Treg cells, we are holding on doing further single cell RNA sequencing and TCR-seq until we have access to a technology able to capture more cells or samples with enough cells to sort enough Treg cells. This latest point will require an IRB-approved study since BM aspirates are painful and will require prolonged anesthesia. We have been able to characterize quite well the ST2⁺ T_{reg} cells at the protein level and through a series of adoptive transfer and knockouts. We focused on the ST2⁺ T_{reg} cells as there are a predominant BM and AML Tregs populations and because specific therapeutic targeting is available to us.

The authors should further explore the underlying mechanisms, and better characterize the changes in activated ST2⁺ Treg cells, and how those cells differ from rest ST2⁺ Tregs, or total Treg populations in AML conditions VS steady-state conditions.

New Fig. 1 C-J shows characteristics of the activated ST2⁺ Treg cells and ST2⁺ Treg cells as well as compared it to ST2⁻ Treg cells (total Tregs that are not ST2⁺). Fig. 1J assessed the immune suppressive capacity of TME BM ST2⁺ Treg cells as compared to ST2⁻ Treg cells with CD8⁺ T cells immune suppression assays and showed that ST2⁺ Treg cells are significantly more suppressive than ST2⁻ Treg cells. New Figure 4C characterizes the high migratory potential of TME BM ST2⁺ Treg cells as compared to ST2⁻ Treg cells as well as non-TME ST2⁺ Treg cells.

Analysis of Tregs from BM of AML patients in different stages of disease including Diagnosis, Remission, Relapse would be very important, and validation studies on additional mouse models of AML by coupling flow cytometry data with single-cell technology and imaging approaches to visualize the histological findings would strengthen the manuscript.

As mentioned above, using the best conditions for 10X genomics method which were BM samples after chemotherapy induction with no more than 20% blasts, a distinct subset of ST2(IL1RL1)+ Treg cells was not visible. We have also analyzed the data from the manuscript PMID: 30827681 and could not detect ST2(IL1RL1)+ Treg cells. The investigators of the manuscript were not able to see an increase of total FOXP3⁺ T_{reg} cells and had to perform immunohistochemistry to see FOXP3⁺ T_{reg} cells, population that was increased in AML vs Healthy Donor but not enough to be captured on 10X genomics sequencing.

Bone marrow biopsies in AML patients are rarely performed (usually only if myelofibrosis is suspected) and if biopsies are performed it is at diagnostic and serial biopsies are not standard of care for AML and will require an IRB-specific study that is not available to us. Furthermore, decalcification methods of BM biopsy specimens are complex. Therefore, to try to address the histology images, we used the epigenetic *Dnmt3a*^{fl/+} *Cre*+/*Flt3*^{ITD/+} that develop frequent extramedullary leukemia in the spleen and liver and performed confocal imaging. Data are shown in new Fig.8E.

The blocking experiments shown in Fig. 6, should be repeated in additional mouse models of AML, other than the MLL-AF9 model, and/or in AML Patient-derived xenografts (PDX) to confirm that the findings are not specific to the aggressive models analyzed.

Fig. 6 is now Fig. 7. We have now performed the ST2 blocking experiments in the epigenetic *Dnmt3a*^{fl/+} *Cre*+/*Flt3*^{ITD/+} as well. Data are shown in new Fig. 8.

We have used a PDX model with MOLM 14 cells established from the peripheral blood of a 20-year-old man with acute myeloid leukemia AML FAB M5a at relapse in 1995 after initial myelodysplastic syndrome. In order to humanize our model and to make it immunologically relevant to our findings, we adoptively transferred both 10⁶ human CD8⁺ T cells and 2*10⁵ human ST2⁺ T_{reg} cells. That said, these “humanized immune” models are suboptimal as compared to non-irradiated immunocompetent MLL-AF9 and epigenetic *Dnmt3a*^{fl/+} *Cre*+/*Flt3*^{ITD/+} models that contain all immune and non-immune cells possibly involved.

Finally, the authors should consider simplifying the manuscript and better highlighting their findings in the main figures.

We agree with the reviewer and the paper has been streamlined and key findings are presented in the main figures.

Reviewer #1 (Remarks to the Author)

The authors have addressed all my comments well; however, I still have one minor comment:

Line 390: The authors suggest a negative effect on NK cell survival via bystander secretion of TGF β and IL-10 by ST2⁺ Treg cells. However, Fig. S17 indicated that this effect is abrogated using the Z-AAD-CMK GZMB inhibitor, suggesting rather an active killing mechanism.

Response: Thank you for pointing this out. We agree that the use of the GZMB inhibitor Z-AAD-CMK in Fig. S17 suggests that ST2⁺ Treg cells may induce NK cell death through a granzyme B-dependent cytotoxic mechanism, rather than solely through the bystander secretion of IL-10 or TGF- β . We have revised the relevant section of the manuscript to clarify this distinction and acknowledge both possibilities while emphasizing the role of active killing.

Reviewer #2 (Remarks to the Author)

The authors have addressed my concerns.

Response: Thank you for your helpful comments and suggestions. We're glad to hear that the revisions and additional data have addressed your concerns. We appreciate your positive feedback and support for the manuscript.

Reviewer #3 (Remarks to the Author)

I co-reviewed this manuscript with one of the reviewers who provided the listed reports. This is part of the 2 Nature Communications initiative to facilitate training in peer review and to provide appropriate recognition for Early Career Researchers who co-review manuscripts.

Response: Thank you for your helpful comments and suggestions and participating to this training program that is a great opportunity for early career researchers.

Reviewer #4 (Remarks to the Author)

The revised manuscript is an improvement over the original version. The authors have included additional models of AML in their analysis. They have also performed histological analysis of mouse spleens, which further supports their findings histologically. However, something that is missing is the histology analysis of the bone marrow from their mice. While deparaffinization and Immunofluorescence staining can be tricky, they are feasible on paraffin sections from bone tissues.

Response: We thank the reviewer for the insightful suggestion and greatly appreciate the overall positive evaluation of our revised manuscript. We have carefully discussed the proposed bone marrow histological analysis with our team and considered both its technical feasibility and scientific necessity.

We respectfully believe that such staining would not provide additional mechanistic insight or alter our conclusions. In our study, we have performed extensive immunofluorescence staining of the spleen—an extramedullary hematopoietic organ—which has been shown to recapitulate the immune microenvironment of the bone marrow in our AML models. The phenotypic and functional data from spleen-resident immune cells in our model are consistent with what has previously been reported in the bone marrow.

Furthermore, the technical and logistical limitations are considerable. In our center, human AML samples rarely include bone marrow core biopsies—we have not had such a sample available in over five years. On the murine side, our genetic, immunocompetent AML models require over six months to develop significant leukemia, making the addition of a new histological series a major undertaking not feasible within the revision timeframe.